# Radiosity from Individual Urban Landscape Elements Measured Using a Modified Low-Cost Temperature Sensor

**Jane Loveday** [1,*], **Grant K. Loveday** [2], **Joshua J. Byrne** [1], **Boon-lay Ong** [3] and **Gregory M. Morrison** [1]

1   School of Design and the Built Environment, Curtin University Sustainability Policy Institute, Curtin University, Bentley 6102, Australia; josh@joshbyrne.com.au (J.J.B.); greg.morrison@curtin.edu.au (G.M.M.)
2   Independent Researcher, Booragoon 6154, Australia; grant.k.loveday@gmail.com
3   School of Design and the Built Environment, Curtin University, Bentley 6102, Australia; boon.ong@curtin.edu.au
*   Correspondence: jane.loveday@postgrad.curtin.edu.au; Tel.: +61-8-9266-9030

**Abstract:** Loss of green space in our suburban environment is contributing to increased urban heat. The material properties of surface treatments or landscape elements (LEs) are a determining factor in the amount, timing, and type of radiation present in the local environment. Landscape designers can use this information to better design for urban heat management, as emitted and reflected radiation (radiosity) from LEs can affect pedestrians via heat stress and glare and affect energy usage in buildings and houses if the landscape sky view factor is low. Low-cost black painted iButton temperature sensors were successfully used as radiometers to concurrently measure the daytime radiosity from 19 LEs samples located on an oval in the warm temperate climate of Perth, (Australia). Normalisation against gloss white paint on polystyrene removed the effect of varying weather conditions. Each LE had the same normalised average radiosity ($DR_{av}$) between seasons (within ±5%), meaning the relative radiosity of new LEs can be measured on any day. White and lighter coloured LEs had the highest $DR_{av}$ and would have the most detrimental effect on nearby objects. Plants and moist LEs had the least $DR_{av}$ and would be most beneficial for managing local daytime urban heat. Measuring relative radiosity with iButtons presents a new way to examine the effect of LEs on the urban environment.

**Keywords:** radiant energy; iButtons; thermochrons; urban heat; landscape design

---

## 1. Introduction

The urban heat island (UHI) effect is the increased night−time ambient temperatures in urban areas when compared with nearby rural areas [1]. Superimposed on this are the increasing ambient temperatures due to global warming, causing particular concern in urban areas. One of the main drivers of the UHI is land use change and the replacement of vegetation with hardscape such as concrete, asphalt and pavers [2–5]. The landscapes surrounding buildings and houses in the urban and suburban environment may thus play an important role in subsequently managing this accumulated heat.

Landscapes can consist of a number of different individual surface treatments defined here as landscape elements (LEs), such as turf grass, concrete paving, crushed rock, mulch, trees, and shrubs. The amount and timing of urban heat is influenced by the material properties of these LEs, in particular albedo, moisture availability, surface roughness, and thermal inertia [1,6]. On reaching the surface of the LE, a portion of the solar radiation is reflected (albedo) and the remainder is absorbed. The surface

temperature rises until there is a balance between heat coming in and heat which is convected, conducted (thermal inertia), emitted and evaporated (if moisture is present) (Figure 1). These properties affect urban heat through the parameters of surface temperature, sensible heat flux (convective heat), and total radiant energy (radiosity). Each of these properties and parameters are relevant to urban heat in different ways.

The albedo of a surface is the fraction of solar radiation which is reflected. White and/or reflective materials generally have lower surface temperatures due to greater reflection of radiation [7–9]. However, studies have shown that increased reflected radiation can significantly affect human thermal comfort within that landscape [4,9–12].

The presence of water at the surface of a LE will lead to some thermal energy being lost through evaporation. Wet permeable pavement was studied by Li [13] and the increased rate of evaporation was found to decrease surface temperatures. The caveat to this was that evaporation rates were only high whilst water was available close to the surface. The surface temperature of plants has been found to be lower than other materials by many researchers [14–17], and this has been associated with higher rates of evapotranspiration [18,19]. Water availability is crucial for this process to occur properly, which may pose problems in drier climates.

The heat storage of a LE is affected by both its heat capacity and thermal conductivity, together called the thermal admittance or thermal inertia [1]. The amount of heat absorbed is a function of its coefficient of thermal conductivity and the temperature difference between the surface and sub-surface layer [20]. Higher thermal inertia of pavements has been found to reduce daytime heat through lower surface temperatures [4], but found to increase heat at night−time [6].

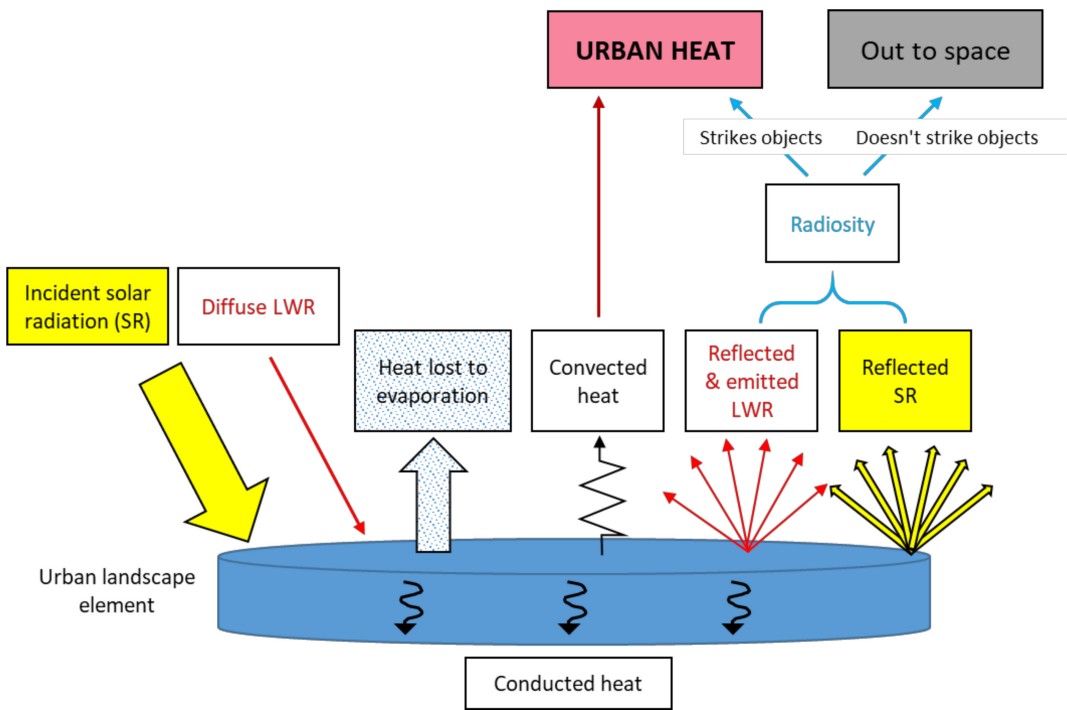

**Figure 1.** Energy balance of an urban landscape element. The range of longwave radiation (LWR) is defined here as 2.5 μm–100 μm. Solar radiation, also termed shortwave radiation (SWR), ranges from 0.3 μm–2.5 μm.

Hence, as well as being a measure of how uncomfortable a surface may be for pedestrians, surface temperature also affects the amount of heat available for storage in the LE. Many studies of urban LE surface temperatures have been undertaken [4,13,15,21–28]. In general, these studies found that manufactured, low albedo and dry materials were hotter than plants or than LEs that were moist or had higher albedos.

Surface temperature also directly affects the amount of heat convected into the air from that surface. From Newton's law of cooling for convection, the amount of convected heat is proportional to the difference between the surface and ambient temperatures, and is also proportional to the coefficient of convection for that surface [29]. The coefficient of convection comprises natural and forced components. When wind speed increases, the convection increases due to the forced convection component. Large areas of a LE with a high surface temperature in urban spaces thus have the potential to significantly increase urban heat.

Longwave radiation (LWR) is emitted from all bodies at temperatures above absolute zero. The higher the surface temperature of a LE, the greater the amount of emitted LWR through the application of Stefan–Boltzmann's law [20]. The amount of emitted LWR is also dependent on the emissivity ($\varepsilon$) of the body, a measure of how effective that body is at emitting heat (0 for a perfect reflector, 1 for a perfect emitter). The emissivity of most LEs is very high, ranging from ~0.76 to ~0.96, with the exception of shiny metals, which can be as low as 0.04 [30], and it varies depending on material surface properties [31].

The radiosity from a LE is the combination of this emitted LWR as well as the reflected solar radiation (Figure 1). Whilst radiosity from LEs does not directly heat the air, it can increase urban heat by striking nearby objects, causing them to increase in temperature [12]. The radiosity of LEs has previously been measured using commercial net radiometers over in situ LEs [12,32], or over specifically constructed LE samples [33]. Other researchers have used pyranometers in conjunction with surface temperature sensors to effectively determine the radiosity over constructed samples [15,34]. As these instruments are relatively expensive, measurements were usually made of one or two LEs at a time, by either alternately facing the instrument towards the LE and then to the sky in the case of a pyranometer, or by leaving the net radiometer above one or two LEs for a period of time, before moving on to the next. This process drastically reduces the number of measurable LEs. Furthermore, the radiosity of smaller LEs cannot be measured using these instruments, as with a diameter of ~ 0.1 m, the shadow cast by the instrument becomes significant, especially as the height above the LE needs to be lowered to increase the amount of radiation received from the LE compared with the surroundings.

Diurnal variations in radiosity are important, as the thermal properties of LEs lead to energy being released back to the surroundings at different times. The timing of this energy release could impact how and when landscapes are used by people. Seasonal variation of radiosity could also be important but does not appear to have been well studied. Li [13], however, measured the albedo and surface temperature of concrete, asphalt and pavers across seasons. Whilst these factors were not combined into a single radiosity measurement, the surface temperature was found to be higher in summer, but the albedo did not change with season.

To address the glare of high albedo materials whilst retaining the benefits of cooler surface temperatures, new materials named cool-coloured materials have been engineered. These increase the reflectance of solar radiation in the near-infrared (NIR) band, offering non-white colours which reflect a significant amount of solar radiation. A laboratory study by Xie et al. [35] found that increased NIR reflectance reduced surface temperatures, while Kyriakodis and Santamouris [36] found the application of these materials to roofs, roads and pavements also reduced surface temperatures when compared with conventional materials (although cool asphalt reflectance halved after deposition of tyre rubber over time). However, although reduced glare for pedestrians is a major benefit of these materials, the effect of the reflected NIR on people and buildings has not been investigated. Hence, as new materials are developed, radiosity testing will be required in the laboratory, but more importantly in the field, where other objects and weathering will influence their radiosity.

Measurement of the radiosity from different LEs will help identify those which would have the most impact on heat management. To reduce the influence of fluctuations in weather conditions on the relative radiosity of different LEs in the field, they should be measured concurrently. Taking a thermal image to find surface temperature is a fairly simple process; however, measuring, and then re-locating, a single radiometer above each LE for each measurement is neither practical, nor time efficient. Furthermore, the high cost of a radiometer and its associated data logger is a barrier to the

purchase of multiple units. Unlike other studies of radiosity, a reference material of white painted polystyrene was used in this research to take into account the effects of changing weather conditions (ambient temperature, wind speed and sky temperature).

Akbari et al. [37] developed a standardised procedure for ranking roofing materials according to both their emitted heat and their solar reflectivity, called the Solar Reflectance Index (SRI). It is generated by measuring a material under two controlled laboratory measurement procedures of reflectance [38] and emittance [39], and calculates SRI based on one assumed set of standard ambient conditions and a convective coefficient for the material. This method could be used for LEs; however, it does not take into account diurnally and seasonally varying ambient temperatures and solar radiation. The assumed convective coefficient would not be applicable to more massive or rough surfaces as are found in LEs as opposed to roofing materials. However, if the reflected and emitted radiant energy from LEs could be measured concurrently and diurnally using a single instrument in the field, then this may provide a simpler and more efficient way of assessing LEs under real world conditions.

Aoki and Mizutani [40] used a low-cost instrument to investigate the vertical distribution of reflected SWR from three different LEs. They constructed a polystyrene post consisting of numerous thin matte black painted aluminium plates mounted at set intervals and monitored the plates' surface temperatures using thermocouples. A similar principle to this has been explored in this paper by using small modified temperature sensors or thermochrons called iButtons [41] to detect radiation across both the visible and thermal spectra. The purpose of this paper was to investigate a low-cost (~AU$50) temperature sensor for use as a radiometer, and to use multiple units concurrently to measure the radiosity from a number of different LEs typically used in Perth urban and suburban areas. Small LEs were tested across the four seasons, whilst larger, ground coupled LEs were tested in summer. LEs were ranked in order of summer radiosity and the implications of the LEs material properties on this ranking are discussed.

## 2. Materials and Methods

Obtaining sufficient measurement accuracy in outdoor urban environments is challenging due to variable weather conditions and interactions between intended measurement materials and surrounding objects such as buildings, vehicles and pedestrians. However, the benefit of measuring in the outdoor environment is the ability to capture the response of the tested LEs to the natural varying diurnal solar radiation (SR) across seasons. Using samples of LEs is beneficial, as they can be placed in an open area with a high sky view factor (SVF) [42] in order to minimise the radiative interference from surrounding objects. The LEs would be exposed to full SR throughout the day. In this study, measurements were performed on samples of LEs located outdoors under a high SVF.

This study consisted of two phases. The first study (phase 1) tested LEs which were smaller in size and placed directly on to the turf grass on the oval. These were measured across each of the four seasons. The second study (phase 2) used larger LEs which were coupled to the ground through a bed of sand placed on the turf grass. Due to their size and subsequent increase in cost, only a subset of the phase 1 LEs were used in phase 2. These were measured in summer only. Loveday et al. [28] indicated that the size and coupling of the LEs to the ground affected the quantity of thermal radiation. Hence, the purpose of phase 2 was to investigate the effect of these factors on radiosity. LEs were compared with each other to determine a relative radiosity ranking.

### 2.1. Study Design Phase 1 (Small LEs)

#### 2.1.1. Materials

Aluminium foil, white painted polystyrene (WPP), black painted cardboard on polystyrene and 19 different LEs (see Table 1) were positioned in the centre of a turf grassed oval at the Booragoon Primary School in Perth, Western Australia (coordinates: −32.035918°, 115.826655°). Perth has a temperate climate with a dry summer, classified as Csa under the Köppen–Geiger climate classification

system [43]. The foil was first crumpled and then straightened out (but not smoothed) and mounted shiny side up on a flat wooden board. This provided a diffuse reflector with a reflectivity as close to 1.0 as possible and was used to measure reflected sky temperature (as per the reflector method used in [44]). The black and white painted polystyrene were used as reference materials, with the polystyrene insulating these from the ground. Most LEs were placed directly onto the turf grass (Figure 2). These were the red, sandstone, and grey pavers, both concrete slabs, artificial turf grass, all of the shade cloths, the decking, and the limestone block. Some LEs were placed onto a white polypropylene (PP) bag on the turf grass. These were the white and polished black stones, and the moist soil. The remaining LEs (soil, asphalt, mulch, and white and grey sand), were also placed onto a PP bag but were then placed within a low-sided plastic crate which allowed the samples to be transported between seasons whilst retaining the same surface area and depth of the LE. The artificial grass was set up in two ways to observe if there were any differences related to the installation method. One sample of artificial grass was placed directly onto the oval grass, while a second sample was first laid onto a bed of crushed rock and the surface sprinkled with white sand, as per a typical installation procedure, and then placed on to a PP bag within a plastic crate. To prevent them from blowing away, all light mass LEs (artificial grass, shade cloths, WPP and BPP), as well as the edges of any PP (including the PP in the plastic crates), were pinned to the ground using four small metal tent pegs. The pegs were hammered straight into the ground next to the LEs, with the short end of the peg unobtrusively holding down each of the four corners of the LE or PP. The turf grass on the oval was *Pennisetum clandestinum*, (the common name being kikuyu), and was managed and irrigated. LEs were spaced between 200 and 600 mm apart to reduce their effect on each other. To prevent any interference from the public, safety tape was erected around the site.

Ideally, all the LEs would have the same surface area in order to minimise confounding factors. In reality, both the standard sizes of the each LE type, and the way they were constructed for the test work, varied (Table 1). The different shapes were accounted for by calculation of view factors, described in Section 2.3.4.

Small wireless temperature sensors called Themochron iButtons (Maxim Integrated, San Jose, CA, USA) [42], capable of logging data over long time periods, were used as low-cost radiometers by painting one surface with matt black paint [45]. These painted sensors measure 17 mm in diameter and are 6 mm high (Figure 3a). The undercoat was a spray of matt black (brand Fiddly Bits), with a topcoat of Rust Guard satin black spray (brand White Knight), applied to give a visibly darker finish. The types of iButtons used were the DS1921G low temperature (−40 to +85 °C, accuracy ±1.0 °C), and the DS1922T, high temperature (0 to +125 °C, accuracy ±0.5 °C) thermochrons. The response time of the iButtons is given as 130 s, although validation studies performed by van Marken Lichtenbelt et al. [46], albeit on the DS1291H (range 15–45 °C), indicated a better accuracy and response time in practice than given by the manufacturer (on average 0.09 °C less than a standard thermometer, and a response of only 19 s). The black paint of the iButton is expected to absorb both longwave radiation (LWR) and short-wave radiation (SWR), with absorptivities of 0.88 [30] and 0.95 [47], respectively. It will also respond to convected heat from the LE, although this depends on its proximity to the LE and the strength of the wind currents.

**Table 1.** Description of landscape elements (LEs) used in the phase 1 seasonal test.

| LE Name | Description | Overall Dimensions of LE (mm) |
|---|---|---|
| Artificial turf grass | Tuff Turf Grass Mat 20 mm pile, green/natural brown curl | 1000 × 1000, height 20 |
| Artificial turf grass (laid) | As above but laid on crushed rock ~50 mm thick. Top surface was then sprinkled with white sand. | 410 × 560 × 50 |
| Asphalt | Westbuild asphalt pack (half a 20 kg bag,) | 290 × 340 × 50 |
| Concrete slab (×40 mm) | Grey concrete slab | 460 × 350 × 38 |
| Concrete slab (×80 mm) | 2 × Grey concrete slab 38 mm thick | 450 × 550 × 76 |

**Table 1.** *Cont.*

| LE Name | Description | Overall Dimensions of LE (mm) |
|---|---|---|
| Crushed rock (white) | Tuscan path 20 mm–25 mm snow white quartz pebbles 20 kg | 440 × 440 × 60 |
| Decking | Timberlink Australia 70 mm × 19 mm treated pine, no finish, (5 lengths mounted on 2 pine joists, height 70 mm) | 360 × 1070 × 19 |
| Limestone block | Flat edge | 500 × 250 × 150 |
| Pine bark mulch | Richgro pinebark (half a 40 L bag) | 580 × 550 × 80 |
| Pavers (grey) | Tuscan Path bluestone paver | 400 × 400 × 40 |
| Pavers (red) | 6 × Ezi-Pave Burnished Red (Midland Brick) 232 mm × 153 mm × 50 mm | 450 × 460 × 50 |
| Pavers (sandstone) | Tuscan Path sandstone paver | 400 × 400 × 40 |
| Polished stones (black) | Tuscan path polished black pebbles stone (half a 15 kg bag) | 550 × 500 × 75 |
| Sand (grey) | Residential garden bed sample | 450 diameter, depth 50 |
| Sand (white) | Westbuild white, washed sand, (half of a 20 kg bag) | 450 diameter, depth 50 |
| Shade cloth (black) | Coolaroo black 70%, folded into four layers | 500 × 560 |
| Shade cloth (cream) | Coolaroo sandstone 70%, folded into four layers | 470 × 530 |
| Shade cloth (white) | Hortshade white 50%, folded into four layers | 470 × 530 |
| Soil (dry) | Hortico garden soil (half a 25 L bag) | 500 × 500 × 50 |
| Soil (moist) | Hortico garden soil (half a 25 L bag) | 410 × 780 × 50 |
| Turf grass (kikuyu) | *Pennisetum clandestinum*, managed and irrigated | ~1000 × ~2500 |
| White painted polystyrene | Gloss white, exterior water based, 4 seasons, (British Paints) on a polystyrene base | 560 × 460 × 35 |
| Black painted card on polystyrene | Flat black, exterior water based, 4 seasons, (British Paints) painted black cardboard on polystyrene | 460 × 410 × 35 |

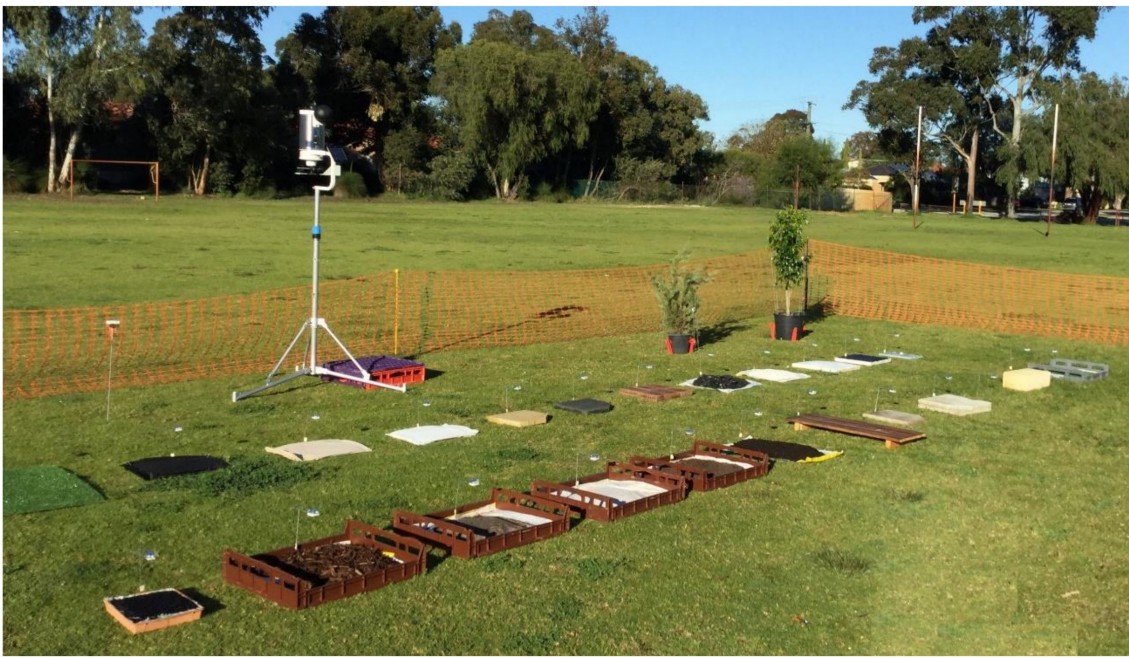

**Figure 2.** Phase 1 LEs showing the low-sided plastic crates and the polypropylene bags used to contain some LEs.

Each LE had a black painted iButton mounted in a polystyrene block (described in [45]), suspended 0.2 m above it, protruding 0.15 m towards the centre of the LE from the middle of one edge of the LE (Figure 3b). The top side of the polystyrene was covered in aluminium foil to reduce solar heat gain to the iButton. A thermal imaging camera (Testo 876, Testo Pty. Ltd, Croydon South, Australia) with range of 8–14 μm and accuracy ±2 °C was used to take images from directly above each LE from a height of approximately 1 m.

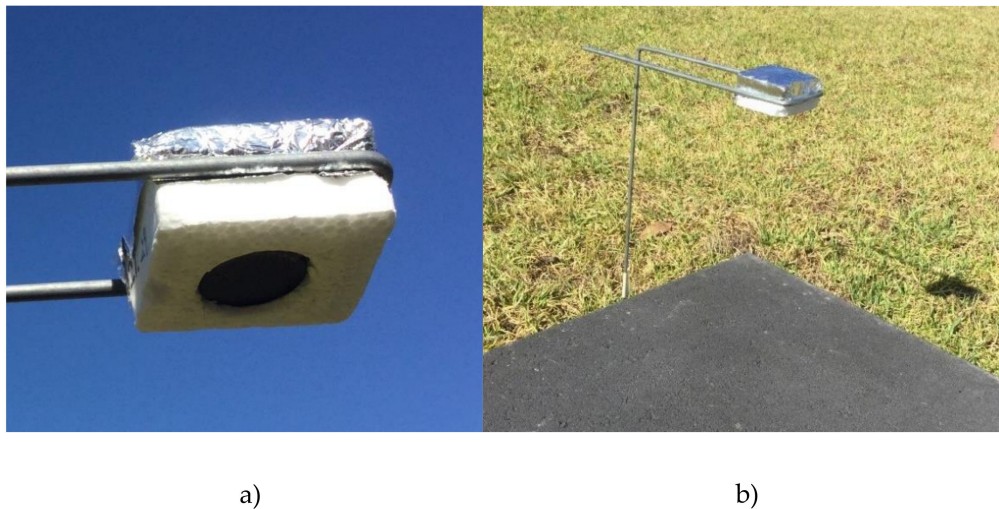

a)                                                                                                           b)

**Figure 3.** iButtons during measurements: (**a**) black painted iButton in a polystyrene block; and (**b**) iButton in a polystyrene block covered in aluminium foil, measuring above the grey pavers LE.

### 2.1.2. Field Measurement Conditions

Measurements were performed over one day during each of the four seasons: 9th April 2017 (autumn); 24th August 2017 (winter); 5th October 2017 (spring); and 9th January 2018 (summer). Measurements commenced a few hours before sunrise and continued until a few hours after sunset. The chosen days had no visible cloud cover and had average wind speed for that time of year. Weather parameters logged every 600 s were accessed via a nearby weather station at Murdoch University [48], 3.6 km away. Conditions were close to the Bureau of Meteorology averages for the measured months. LEs were placed on the oval the day previous to each test day in order to equilibrate to outdoor conditions.

The sampling rate of each iButton was set to 600 s by connecting it to a computer through a USB interface (LinkUSBi, Maxim Integrated, San Jose, CA, USA) and using the manufacturer's software (OneWireViewer, Maxim Integrated, San Jose, CA, USA). Thermal images were taken every half an hour.

### 2.2. Study Design Phase 2 (Larger, Ground-Coupled LEs)

### 2.2.1. Materials

The LEs used for phase 2 are described in Table 2. They were all the same product type as those used in the seasonal measurements, except for decking and artificial turf grass, where the original types were unavailable. Although the same product types were purchased for the other LEs, there were some dissimilarities in colour and size between some of the phase 1 and phase 2 LEs, particularly for the more natural materials such as crushed white rocks, pine bark mulch and the soils. The new artificial grass was a slightly darker colour, whilst the new decking was a different shade of brown (Figure 4). In addition, an annual non-native ground cover of petunias (*Petunias × hybrida*) and some native saltbush seedlings (*Atriplex nummularia*) were tested as a comparison to turf grass. The ground cover (petunias) were in 100 mm pots closely packed together to give nearly 100% coverage of the sand. The saltbush seedlings were in 50 × 50 mm containers within seedling trays and similarly closely

packed together. The planting densities are given in Table 2. The LEs used in this test were all 1200 mm × 1200 mm square, and each were separated by 1 m of managed and irrigated turf grass (kikuyu) (Figure 5). As per phase 1, reference materials of WPP and black painted polystyrene (BPP) were included in these tests. Note that the BPP did not include cardboard as per phase 1, and, instead, the black paint was applied directly onto the polystyrene. Crumpled aluminium foil was again used to measure the reflected sky temperature. Ten of the LEs were laid onto a bed of a 1200 mm × 1200 mm square of yellow sand with a thickness of 50 mm, (seen around the edges of the plants in Figure 6). This sand provided a good thermal coupling between the LE and the ground, similar to how the LE would be installed in situ. The reasons for not placing the remaining LEs on the sand were as follows: the decking simulated typical decking, which is not directly coupled to the ground; the reference LEs (BPP and WPP) were required to be insulated from the turf grass; the soils were naturally effectively coupled to the ground through the infiltration of their particles through the turf grass to the soil below; and the turf grass LE was simply the existing turf grass on the oval.

**Table 2.** Installation details of the larger landscape elements used in the phase 2 study design during summer 2019.

| Landscape Elements | Test Date 21st January 2019, Total LE Surface Area 1200 mm × 1200 mm | | |
| --- | --- | --- | --- |
| | Surface Area of Individual LE Components (mm) | Thickness (mm) | Sub-Layer * |
| Artificial turf grass (Tuff Turf Multi) | 1200 × 1200 | 12 | sand |
| Asphalt | 1200 × 1200 | 50 | sand |
| Concrete slab (×40 mm) | 600 × 600 | 38 | sand |
| Crushed rock (white) | 20–25 | 50 | sand |
| Decking (hardwood, Merbau) | 1200 × 90 | 19 | turf grass |
| Pine bark mulch | ~10–50 | 50 | sand |
| Pavers (grey) | 400 × 400 | 40 | sand |
| Pavers (red) | 232 × 153 | 50 | sand |
| Pavers (sandstone) | 400 × 400 | 40 | sand |
| Seedlings (native, Old man saltbush) (*Atriplex nummularia*) | ~50 diameter within seedling tray (plant density 711 plants/m$^2$) | 100–150 (plant height) | sand |
| Ground cover (annual) (non-native, petunias) (*Petunia × hybrid*) | ~150 diameter pots (plant density 34 plants/m$^2$) | 100–150 (plant height) | sand |
| Soil (dry) | 1200 × 1200 | 50 | turf grass |
| Soil (moist) | 1200 × 1200 | 50 | turf grass |
| Turf grass (kikuyu) | 1200 × 1200 | | |
| White painted polystyrene (two coats of gloss white, exterior water based, 4 Seasons, British Paints) | 1200 × 1200 | ~0 | 40 mm polystyrene |
| Black painted polystyrene (undercoat was exterior low sheen black, 4 Seasons, British Paints; top coat was flat black enamel spray paint, Rust guard, White Knight) | 1200 × 1200 | ~0 | 40 mm polystyrene |

* sand = 50 mm thick yellow sand on top of turf grass.

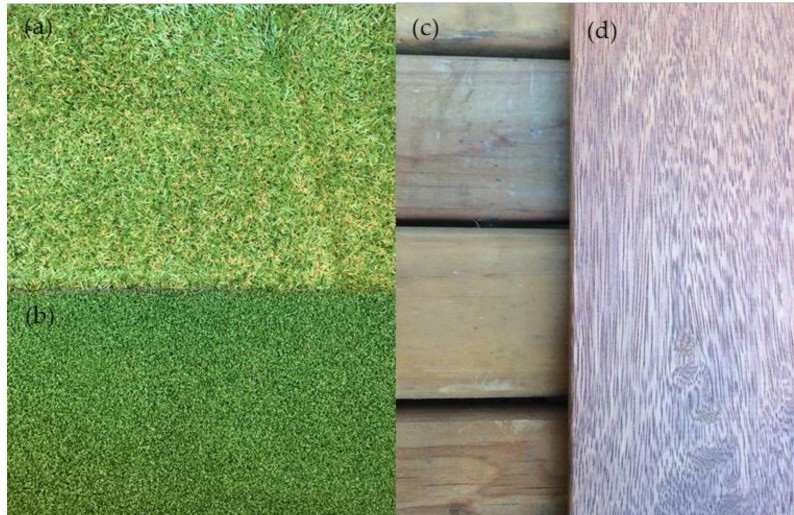

**Figure 4.** Different materials used in phase 1 and phase 2: (**a**) phase 1 artificial grass; (**b**) phase 2 artificial grass; (**c**) phase 1 decking; (**d**) phase 2 decking.

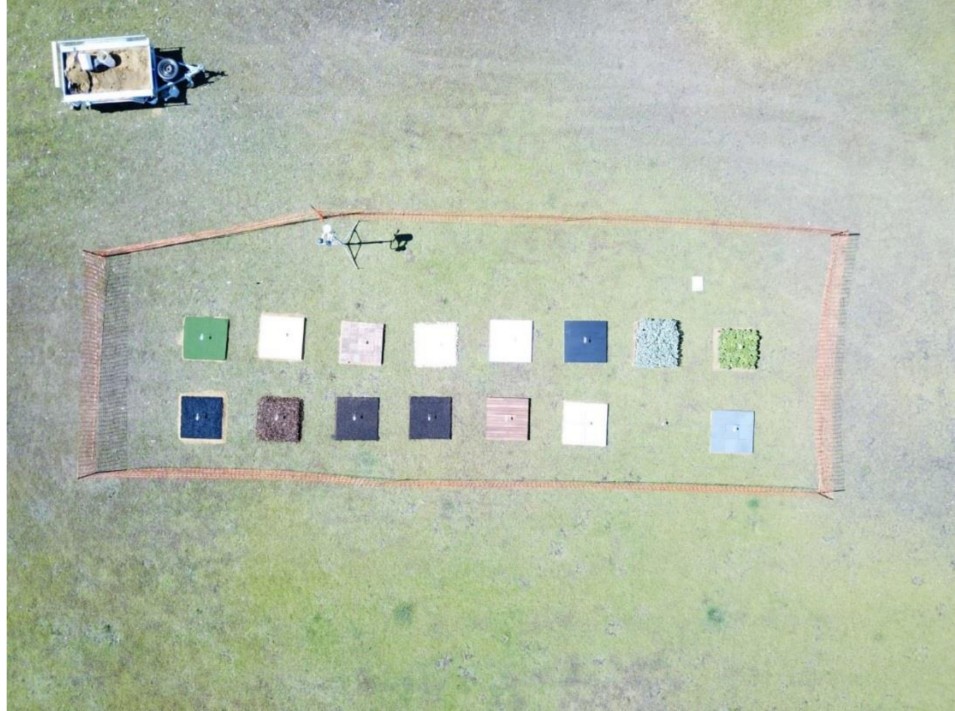

**Figure 5.** Aerial view of landscape elements on the oval. From top left: artificial grass, pavers (sandstone), pavers (red), crushed rock (white), white painted polystyrene (WPP), black painted polystyrene (BPP), saltbush (native), and petunia (non-native). From bottom left: asphalt, pine bark mulch, soil (dry), soil (wet), decking, concrete, turf grass, and pavers (grey). (Photo courtesy of Grant Bayne).

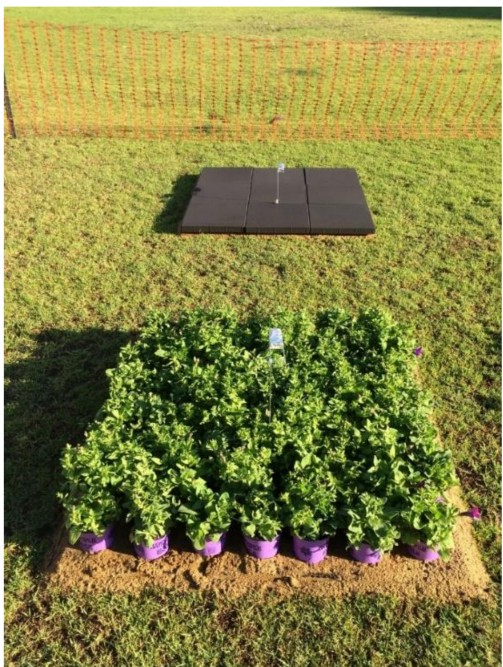

**Figure 6.** Ground cover (non-native, petunias) (*Petunia × hybrid*) and pavers (grey), on a bed of yellow sand.

### 2.2.2. Field Measurement Conditions

Phase 2 was performed on the same oval as the phase 1 tests. The LEs were set up on the oval on the 19th January 2019 to allow them to equilibrate with the conditions. The test day had clear skies only. Thermal images were taken every 30 min and iButton data were logged every 10 min from 06:00 to 21:00 on the 21st January 2019 (summer).

### *2.3. Data Analysis*

### 2.3.1. Energy Balance

The energy balance between the iButtons and the LEs was modelled to determine the unknown parameters of emissivity (described below) and coefficients of convection for the iButtons and the LEs. This would allow for the calculation of the radiosity for each LE. However, the model was not sufficiently robust to provide consistent convective coefficients, possibly due to unaccounted for local weather fluctuations, which varied the amount of convected heat reaching the iButtons. A simpler method was developed which involved using the iButton temperatures ($T_{iB}$) alone, and which provided a relative, rather than absolute, measure of the radiosity from the LEs.

### 2.3.2. iButton Temperatures and Emissivity

The paint on the iButtons was applied manually to the surface of each iButton, resulting in a natural variation in quality and thickness. This variation would result in a slightly different emissivity for each iButton ($\varepsilon_{iB}$), ultimately affecting the $T_{iB}$. The purpose of this section is to explain why the variation in emissivity was not significant.

Approximate values of the convective coefficients were used from the previous modelling work to determine how much the variation in the black iButton paint would affect $T_{iB}$. It was found that changing $T_{iB}$ by up to ±1.0 °C changed $\varepsilon_{iB}$ by a maximum of ±0.2. As the emissivities are unlikely to vary by this large amount, the effect of emissivity variation could be ignored, as it was negligible when compared with the uncertainty in the iButton temperature loggers themselves (given by the manufacturer as ±0.5 °C and ±1.0 °C for high and low temperature iButtons, respectively). The fact that

$T_{iB}$ was so insensitive to the $\varepsilon_{iB}$ was perhaps another reason why the initial energy balance modelling described above was unsuccessful.

### 2.3.3. iButton Temperatures and Convection

The resultant iButton temperatures $T_{iB}$ are likely to be more sensitive to the convective heat transfer from their surface than to the emissivity. The iButtons were used in the downward facing position, and because of this, the heat loss due to forced convection (wind speed), was a more significant factor than the heat loss due to natural convection, which is of the order of $1 \text{ W·m}^{-2}\text{·K}^{-2}$ under calm conditions [49]. However, as the LEs were all measured concurrently, each iButton would have responded similarly to the changes in wind speed (and in fact, to all weather conditions). The relative iButton temperatures, and hence the relative radiosity measured by the iButtons, would thus be comparable.

### 2.3.4. View Factors

The height of the iButton above the LE was chosen so that its shadow did not impact significantly on the LE, but such that the majority of the radiation it received would be from the LE rather than from the surrounding grass. The optimisation of height was discussed in terms of the view factor (VF) by Cueto et al. [33], who measured net radiation from various LEs using a net radiometer. The VF of the LE to the measurement instrument is defined as the fraction of radiation received by the instrument from the LE as a percentage of the total radiation received. In this case, the temperature of each iButton was affected by the field of view (FOV) of its painted face which was pointing directly down at the LE and estimated to be ~150°. The assumption was that the sides and back of the iButton were well insulated within the polystyrene block.

The initial experimental design using phase 1 LEs was developed for measuring surface temperature, where interference from nearby LEs was not considered to be significant as long as they were all low-set and located a reasonable distance apart on the ground. However, when using iButtons, this set up causes the iButton to receive radiosity not only from the LE plus a smaller fraction from the surrounding turf grass, but also from the PP bag, the plastic crate, and the nearby LEs. Hence, more complex analyses are required to determine the radiosity purely from the LE alone.

To calculate the approximate VF of each iButton/LE configuration, each LE was assessed to be either a circular disc, or a rectangle. The iButtons above circular LEs were treated as a circle of radius 8 mm; however, for ease of calculation, the iButtons above the rectangular and square LEs were considered to be square with side lengths 7 mm. An online calculator by Howell [50] was used to find the VFs for each iButton/LE pair plus their surrounding grass, and, where required, the edges of the PP bag. The rectangle-to-rectangle in a parallel plane function was used to determine the VFs for the rectangular LEs, and the disk-to-parallel coaxial disk of unequal radius function was used for the circular LEs. Anything other than LE, grass, and PP, was termed Other, and the VF was denoted as $VF_{Other}$. Equation (1) relates the combination of iButton temperatures and VFs.

$$T_{iB,LE+grass+PP+Other} = VF_{LE}T_{iB,LE} + VF_{grass}T_{iB,grass} + VF_{PP}T_{iB,PP} + VF_{other}T_{iB,Other} \tag{1}$$

where $T_{iB,LE+grass+PP+Other}$ is the iButton temperature, $T_{iB,LE}$ is the iButton temperature due to the LE only, $VF_{LE}$ is the VF of the LE only, with similar nomenclature applying to grass, PP, and Other.

Equation (1) was used to recalculate $T_{iB,LE}$ for each LE, by removing the fraction of energy from the visible turf grass (using the $T_{iB,grass}$ measured at the same time), and, for some LEs, by also removing the $T_{iB,PP}$, which was assumed to have a similar $T_{iB}$ as white shade cloth. $VF_{Other}$ was the remaining fraction of the total FOV once the VFs of the LE, grass and PP were summed. As there were no data for $T_{iB,Other}$, in these analyses it was assumed to be equal to $T_{iB,grass}$, i.e., everything other than LE and PP was assumed to be turf grass. Considering this, Equation (1) became Equation (2):

$$T_{iB,LE} = \frac{T_{iB,LE+grass+PP+Other} - (VF_{grass} + VF_{other})T_{iB,grass} - VF_{PP}T_{iB,PP}}{VF_{LE}} \tag{2}$$

### 2.3.5. Material Properties Affecting iButton Temperatures

The expected values of each $T_{iB,LE}$ can be estimated based on the material properties of thermal inertia, colour (albedo), roughness, and moisture content. As discussed in the introduction, darker but higher thermal inertia LEs (such as asphalt and black polished stones) have the capacity to absorb more incident energy during the day, hence their $T_{iB,LE}$ would be cooler than that of dark LEs with lower thermal inertia (such as black shade cloth, BPP and artificial grass). Convected heat from the LEs can be removed by wind currents [29], hence the convected heat received by the iButtons will vary with local wind conditions. In comparison, the surface of a white LE reflects most of the incident radiation onto the iButton and is less affected by wind conditions. Hence, the $T_{iB,LE}$ of dark LEs would tend to be lower, whilst that of white LEs would tend to be higher. LEs with rough surfaces have a higher specific surface area, i.e., more surface area available for cooling through forced convection, and would thus have a lower $T_{iB,LE}$ than smooth surfaces under windier conditions due to less radiant LWR. A higher specific surface area also has a lower albedo, as light has more chances to be internally reflected before escaping from the surface. LEs with a high moisture content are able to use that moisture for cooling through evaporation (or evapotranspiration in the case of plants), and would have a lower $T_{iB,LE}$.

### 2.3.6. Relative Radiosity Ranking

Each LE was compared with the reference material of WPP in order to find its relative ranking with respect to the other LEs. The temperature of the iButton over the WPP ($T_{iB,WPP}$,) is always going to be the hottest during the daytime as it will have the most reflected incident radiation (due to WPP having the highest albedo), and because the WPP is insulated, no heat will be conducted away into the ground. The other LEs absorb more radiation (than the WPP), which heats their surface. Some of this is re-emitted as LWR, some is convected away, and some is conducted into the body of the LE and then into the ground. Consequently, the black iButton above these LEs ($T_{iB,LE}$) will never be as hot as that above the WPP. Thus, WPP was an ideal material to be used as an upper reference point and $T_{iB,WPP}$ was used to normalise each $T_{iB,LE}$.

A direct ratio of $T_{iB,LE}$ over the black iButton temperature of the WPP ($T_{iB,WPP}$) would not be consistent across the day, because the iButtons are also affected by the varying ambient temperature. Removing $T_{amb}$ from each iButton temperature removes this diurnal variability from the data. In the evening, the WPP temperature drops below ambient temperature because it has no thermal inertia and is radiating heat to the cold night sky. This method was therefore not used after sunset.

All $T_{iB}$ were first corrected for VFs. Equation (3) was used to determine the daytime ratio (*DR*) for ranking the radiosity of the LEs relative to the WPP.

$$DR = \frac{(T_{iB,LE} - T_{amb})}{(T_{iB,WPP} - T_{amb})} \tag{3}$$

The uncertainties in $T_{iB,WPP}$ were ±1.0 °C, whilst the uncertainties in the other iButtons were either ±1.0 or ±0.5 °C, depending on their type. The uncertainty in $T_{amb}$ was estimated to be ±0.5 °C. The standard errors (SE) were calculated assuming random errors and a normal distribution [51]. *DR* was averaged (*DR*$_{av}$) over the time periods where the SR was greater than 70% of that occurring at solar noon on that day, in line with the standard for measuring reflected radiation from objects in the field [38]. For autumn this was 09:30–15:00; for winter, 09:20–15:00; for spring, 09:00–15:00; and for both summer measurements, 09:00–15:40.

Whilst the iButton and $T_{amb}$ errors are relatively small, the uncertainty in the mean of the data due to the methodology used to derive the day–time ranking was expected to be higher than that calculated from the instrumentation alone. This is because the factors affecting the measurement and calculation of $DR_{av}$ relied on physical properties of materials, convection coefficients, and on varying weather parameters. Hence, the $DR_{av}$ uncertainty was found by calculating the standard error (SE)

of the actual $DR_{av}$ data for each LE during the daytime hours, where the SE is equal to the standard deviation divided by the square root of the number of sampling points.

Microsoft Excel's Data Analysis package (Microsoft Corporation, Makati, WA, USA) was used to perform multiple two-sample t-tests (assuming equal variances) on both the seasonal $DR$ data from phase 1, and on the summer $DR$ data from phase 2. The summer 2018 and summer 2019 $DR$ data was also compared using this t-test. This analysis tested the null hypothesis that the means between each LE whose rankings placed them next to each other, were the same (at a significance level $\alpha = 0.05$).

### 2.4. Delimitations

This study was performed under clear sky conditions with a high SVF and with minimal wind. The results may not be applicable to measurements made under cloudy skies or where wind gusts are strong enough to significantly cool LEs and iButtons. Ambient conditions would be best measured using a local weather station; however, the local weather station malfunctioned during some of the test work and data was used from the nearby meteorological station. It is recognized that in reality, some of the hardscape LEs are coupled to the ground using more complex substrates than the sand used in this study. Furthermore, mechanical compaction of the asphalt LE was not performed according to standard road surface preparation. The number and type of LEs tested in phase 2 was limited by the time available for both construction and measurement (LEs were on a school oval), as well as by the cost of these LEs. Hence, replicate LEs were not tested. The moisture content of the LEs and of the turf grass surrounding them was not measured as it was outside the scope of this study; however, moisture content contributes to surface temperature through evaporative cooling.

## 3. Results and Discussion

### 3.1. Weather Conditions

Apart from 21st January, 2019, which had slightly cooler ambient temperatures and a higher relative humidity than typical for this season, the weather conditions for each of the days measured were close to typical for that season, according to data from the Bureau of Meteorology [52]. The ranges and conditions for both studies are given in Table 3, and shown in Figure 7. Smoke haze from a bush fire was present from around 13:30 to the early evening on the spring day (5th October, 2017) and this can be seen in Figure 7 as a reduction in the SR at these times. This may have led to lower than expected LE temperatures for this season. In summer 2018 at ~13:30, the ambient temperature dropped whilst the relative humidity and the wind speed increased suddenly, possibly due to a weather front moving in. The effects of this weather change are seen in results presented later.

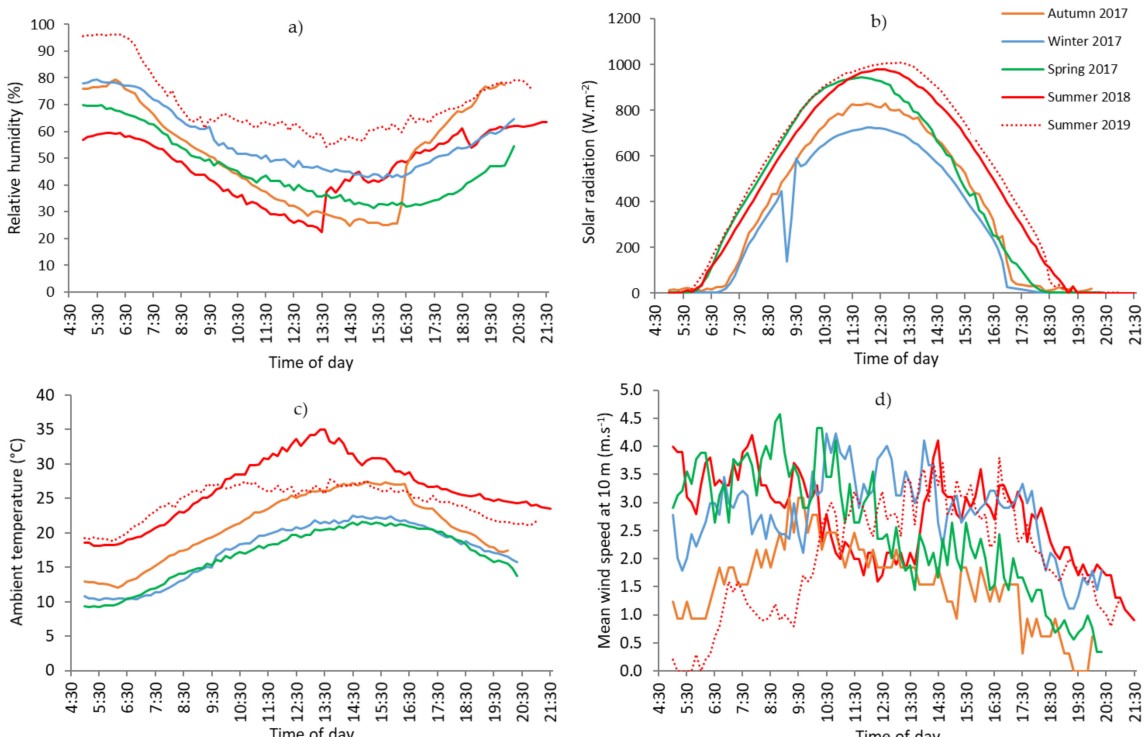

**Figure 7.** Summary of the weather conditions during the measurement days; (**a**) relative humidity (%); (**b**) solar radiation (W·m$^{-2}$); (**c**) ambient temperature (°C); and (**d**) mean wind speed at 10 m (m·s$^{-1}$). Dates were: 9th April 2017 (autumn 2017); 24th August 2017 (winter 2017); 5th October 2017 (spring 2017); 9th January 2018 (summer 2018); and 21st January 2019 (summer 2019).

## 3.2. View Factors

For phase 1 LEs, $VF_{Other}$ ranged from ~1% up to ~8% of the total FOV of the iButton depending on the LE. Hence the assumption made that $T_{iB,Other}$ was equal to $T_{iB,grass}$, introduced an uncertainty in the calculation of $T_{iB,LE}$, which was dependent on the difference between the radiosity of turf grass and the radiosity of Other. As the VFs of Other were relatively small, this uncertainty was also expected to be small.

**Table 3.** Weather conditions during day within the specified time of measurements (from Murdoch University).

| Date & Season | Time Range | Air Temp. Range (°C) | Relative Humidity Range (%) | Max Solar Radiation (W·m$^{-2}$) | Wind Speed (m·s$^{-1}$) Average (Standard Deviation) |
|---|---|---|---|---|---|
| **9th April 2017** **Autumn 2017** | 09:30–15:00 | 19.4–27.4 | 25–51 | 831 | 2.05 (0.47) |
| **24th August 2017** **Winter 2017** | 09:20–15:00 | 15.0–22.5 | 43–62 | 724 | 3.34 (0.55) |
| **5th October 2017** **Spring 2017** | 09:00–15:00 | 14.9–21.6 | 33–51 | 945 | 2.79 (0.76) |
| **9th January 2018** **Summer 2018** | 09:00–15:40 | 24.5–35.0 | 23–45 | 1049 | 2.61 (0.66) |
| **21st January 2019** **Summer 2019** | 09:00–15:40 | 25.4–27.8 | 54–67 | 1009 | 2.67 (0.75) |

The VFs of the phase 1 LEs, where most LEs had an approximate surface area of ~0.13 m$^2$, were between 0.43 and 0.75 (Table 4). VFs varied slightly with season as the LEs were re-installed each season

and, whilst the order of installation was kept the same, the distances between LEs varied. The large $VF_{grass}$ for these LEs (Table 4) introduces further uncertainty due to the natural variability of the turf grass across the oval and hence around each LE. Whilst the variation in VFs between phase 1 LEs was not ideal, a more significant factor was the variation in the surrounding turf grass quality and the distance to the nearest LEs. To reduce these uncertainties, the height of the iButton over the smaller LEs would have to be reduced in order to increase the $VF_{LE}$. However, this would introduce new uncertainties as the size of the shadow of the iButton would increase, thus reducing the radiosity from the LE.

For the larger phase 2 LEs, $VF_{LE}$ were all 0.92, as these LEs all had the same surface area of 1.44 m$^2$. Calculations from [53] estimated $VF_{Other}$ to be ~1% of the FOV of the iButton. Hence, everything other than the LE was assumed to be turf grass, with a VF of 0.08.

In summary, the uncertainty in the $T_{iB,LE}$ from the phase 1 LEs is much greater than for the phase 2 LEs due to the assumptions made regarding the VFs and the variability in the turf grass.

**Table 4.** Seasonal radiosity view factors to the iButton surfaces from the small phase 1 landscape elements (LEs), and from the surrounding polypropylene (PP) and turf grass.

| Landscape Element | View Factors | | | | | | | | | | | |
| | Autumn | | | Winter | | | Spring | | | Summer 2018 | | |
| | LE | PP | Grass | LE | PP | Grass | LE | PP | Grass | LE | PP | Grass |
|---|---|---|---|---|---|---|---|---|---|---|---|---|
| Artificial grass | 0.75 | 0 | 0.25 | 0.71 | 0 | 0.29 | 0.71 | 0 | 0.29 | 0.71 | 0 | 0.29 |
| Artificial grass (laid) | 0.00 | 0 | 0 | 0.58 | 0.1 | 0.32 | 0.61 | 0.04 | 0.35 | 0.61 | 0.04 | 0.35 |
| Asphalt | 0.43 | 0 | 0.57 | 0.43 | 0 | 0.57 | 0.43 | 0 | 0.57 | 0.43 | 0 | 0.57 |
| Black painted polystyrene | 0.56 | 0 | 0.44 | 0.56 | 0 | 0.44 | 0.56 | 0 | 0.44 | 0.56 | 0 | 0.44 |
| Concrete ($\times$ 40 mm) | 0.54 | 0 | 0.46 | 0.52 | 0 | 0.48 | 0.52 | 0 | 0.48 | 0.52 | 0 | 0.48 |
| Concrete ($\times$ 80 mm) | 0.58 | 0 | 0.42 | 0.58 | 0 | 0.42 | 0.58 | 0 | 0.42 | 0.58 | 0 | 0.42 |
| Crushed rock (white) | 0.58 | 0.06 | 0.36 | 0.58 | 0.06 | 0.36 | 0.58 | 0.06 | 0.36 | 0.58 | 0.06 | 0.36 |
| Decking | 0.60 | 0 | 0.40 | 0.56 | 0 | 0.44 | 0.56 | 0 | 0.44 | 0.64 | 0 | 0.36 |
| Limestone block | 0.45 | 0 | 0.55 | 0.45 | 0 | 0.55 | 0.45 | 0 | 0.55 | 0.45 | 0 | 0.55 |
| Pavers (grey) | 0.54 | 0 | 0.46 | 0.54 | 0 | 0.46 | 0.54 | 0 | 0.46 | 0.54 | 0 | 0.46 |
| Pavers (red) | 0.59 | 0 | 0.41 | 0.59 | 0 | 0.41 | 0.59 | 0 | 0.41 | 0.59 | 0 | 0.41 |
| Pavers (sandstone) | 0.54 | 0 | 0.46 | 0.54 | 0 | 0.46 | 0.54 | 0 | 0.46 | 0.54 | 0 | 0.46 |
| Pine bark mulch | 0.66 | 0.01 | 0.33 | 0.65 | 0.04 | 0.31 | 0.66 | 0.01 | 0.33 | 0.66 | 0.01 | 0.33 |
| Polished stones (black) | 0.62 | 0.07 | 0.31 | 0.62 | 0.07 | 0.31 | 0.62 | 0.07 | 0.31 | 0.62 | 0.07 | 0.31 |
| Sand (grey) | 0.56 | 0.16 | 0.28 | 0.56 | 0.13 | 0.31 | 0.56 | 0.16 | 0.28 | 0.56 | 0.16 | 0.28 |
| Sand (white) | 0.56 | 0.13 | 0.31 | 0.56 | 0.13 | 0.31 | 0.56 | 0.13 | 0.31 | 0.56 | 0.13 | 0.31 |
| Shade cloth (black) | 0.63 | 0 | 0.37 | 0.63 | 0 | 0.37 | 0.63 | 0 | 0.37 | 0.63 | 0 | 0.37 |
| Shade cloth (cream) | 0.61 | 0 | 0.39 | 0.61 | 0 | 0.39 | 0.61 | 0 | 0.39 | 0.61 | 0 | 0.39 |
| Shade cloth (white) | 0.61 | 0 | 0.39 | 0.61 | 0 | 0.39 | 0.61 | 0 | 0.39 | 0.61 | 0 | 0.39 |
| Soil (dry) | 0.62 | 0.05 | 0.33 | 0.62 | 0.07 | 0.31 | 0.62 | 0.05 | 0.33 | 0.62 | 0.05 | 0.33 |
| Soil (moist) | 0.59 | 0.05 | 0.36 | 0.59 | 0.05 | 0.36 | 0.59 | 0.05 | 0.36 | 0.59 | 0 | 0.41 |
| Turf grass | 0.95 | 0 | 0.05 | 0.95 | 0 | 0.05 | 0.92 | 0 | 0.08 | 0.98 | 0 | 0.02 |
| White painted polystyrene | 0.61 | 0 | 0.39 | 0.61 | 0 | 0.39 | 0.61 | 0 | 0.39 | 0.61 | 0 | 0.39 |

*3.3. Seasonal Comparison—Phase 1*

Equation (2) was used to calculate $T_{iB,LE}$ for each LE (data given in Appendix A) based on the VFs given in Table 4. Seasonal data were calculated by normalising all of the $T_{iB,LE}$ with $T_{iB,WPP}$ as per Equation (3). Both seasonal data and the summer 2019 data were analysed using this method. Examples of the temporal daytime ratio data from summer 2018 are shown below, with data separated for clarity (Figures 8–10). The standard errors (SE) given in these figures were calculated assuming random errors and a normal distribution [51]. Temporal data from other seasons were similar but are not presented here. The tails of these figures are extremely unstable due to the nature of the ratio calculation. For example, as the SR decreases to zero at around 18:30, $T_{iB,WPP}$ approaches $T_{amb}$, so the denominator ($T_{iB,WPP} - T_{amb}$) approaches zero, and then becomes negative as $T_{iB,WPP}$ begins to reflect the cold sky temperature. This results in a very large positive ratio just on sunset, followed by a very large negative ratio, as seen in the edges of the graph. Hence this method is only valid during the day.

Figures 8–10 illustrate that the normalisation process is able to take account of the weather variability and varying solar radiation as seen by the reasonably consistent ratio during the daytime from 09:00 to

15:40 (standard errors ≤0.02). This is despite the larger uncertainties in this data due to the experimental design and the corresponding assumptions made around the VFs of these phase 1 LEs.

However, some generalisations can be made. The *DR*s of some of the low albedo, higher thermal inertia LEs (red and grey pavers, and the polished black stones) (Figures 9a and 10a, respectively) have a slight general trend upwards across the day. However, the high albedo, higher thermal inertia LEs (concrete, sandstone pavers, and limestone) (Figures 8b, 9a and 10a, respectively) have a fairly consistent *DR* through the day. It seems the daytime radiosity may be driven more by albedo than by LWR. This is consistent with results by [12] who found that increases in albedo significantly increased the upwards SWR whilst delivering only a minimal difference in the upwards LWR. In the later afternoon, the accumulated heat from the day is released at a slower rate in the higher thermal inertia LEs when compared with WPP, and hence the *DR* increases slowly for these LEs.

In general, the *DR* of the low thermal inertia LEs and natural materials (Figures 8a, 9b and 10b) are more consistent across the day. The sudden weather change at ~13:30, however, affected the *DR* of some LEs (in particular sandstone pavers (Figure 9a), white shade cloth (Figure 9b), and dry soil (Figure 10b), but not others, and affected them in different ways. This requires investigation in future work.

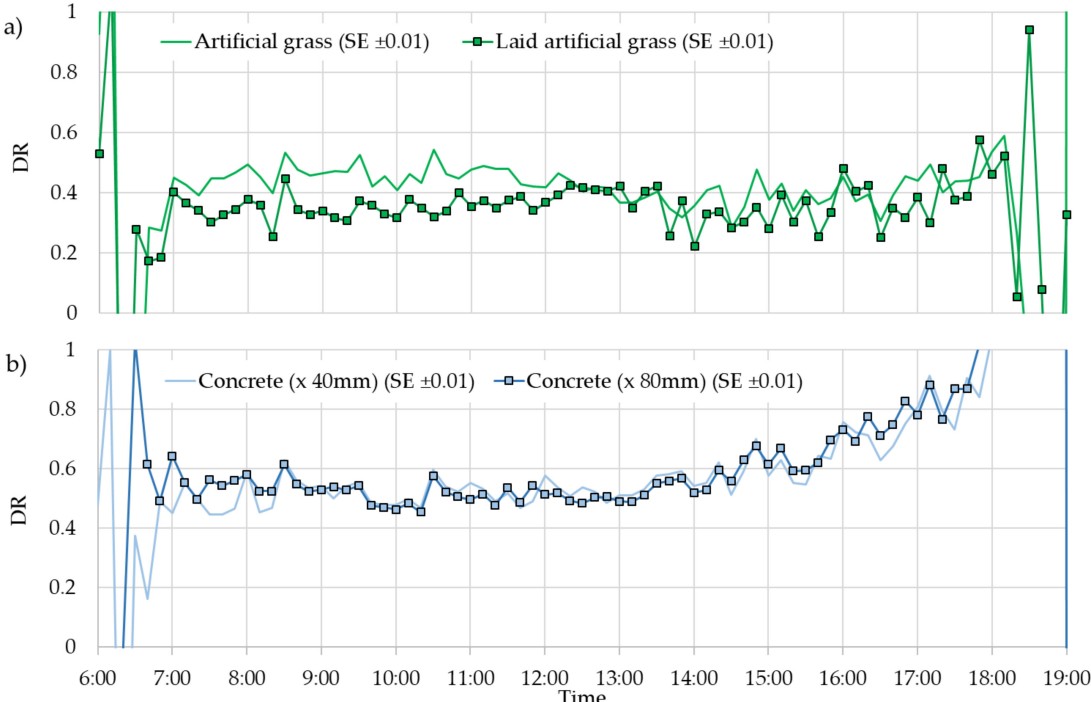

**Figure 8.** Daytime ratio *($T_{iB,LE}-T_{amb}$)* over *($T_{iB,WPP}-T_{amb}$)* for LEs, summer 2018. Comparison of: (**a**) low thermal inertia artificial grass with higher thermal inertia laid artificial grass; and (**b**) lower thermal inertia concrete × 40 mm with higher thermal inertia concrete × 80 mm. Data from 09:00 to 15:40 were averaged. Standard errors (SE) were calculated assuming random errors and a normal distribution [51].

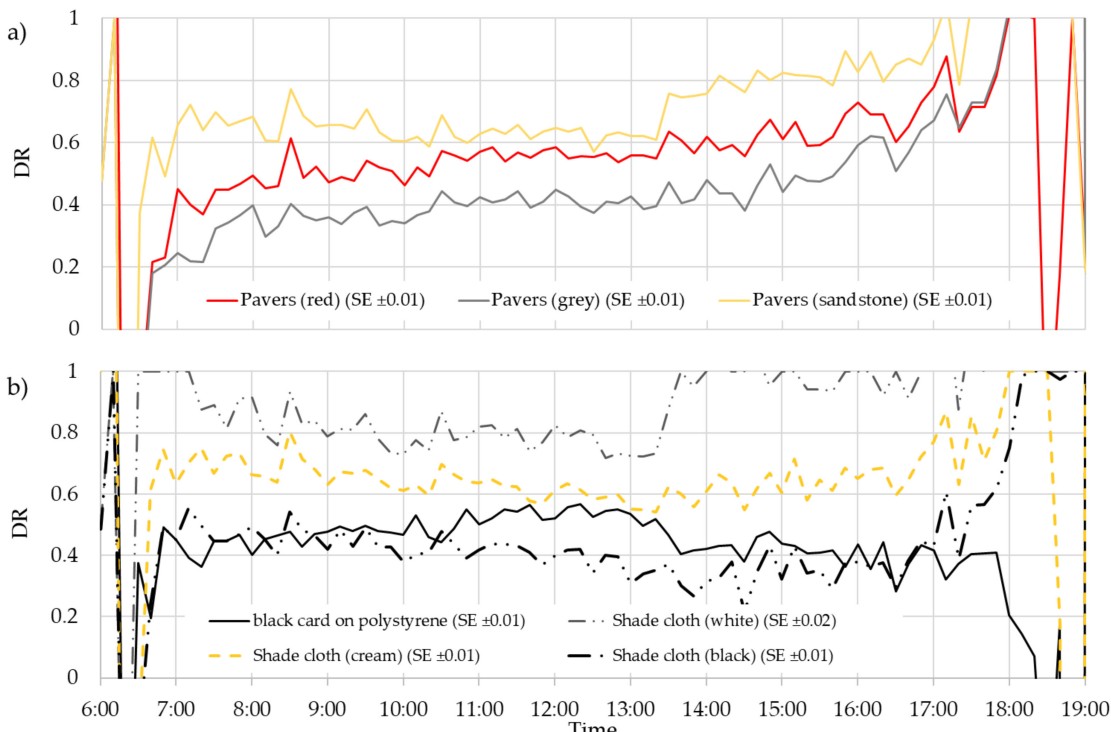

**Figure 9.** Daytime ratio $(T_{iB,LE}-T_{amb})$ over $(T_{iB,WPP}-T_{amb})$ for LEs, summer 2018. Comparison of: (**a**) different coloured pavers; and (**b**) low thermal inertia LEs. Data from 09:00 to 15:40 were averaged. Standard errors (SE) were calculated assuming random errors and a normal distribution [51].

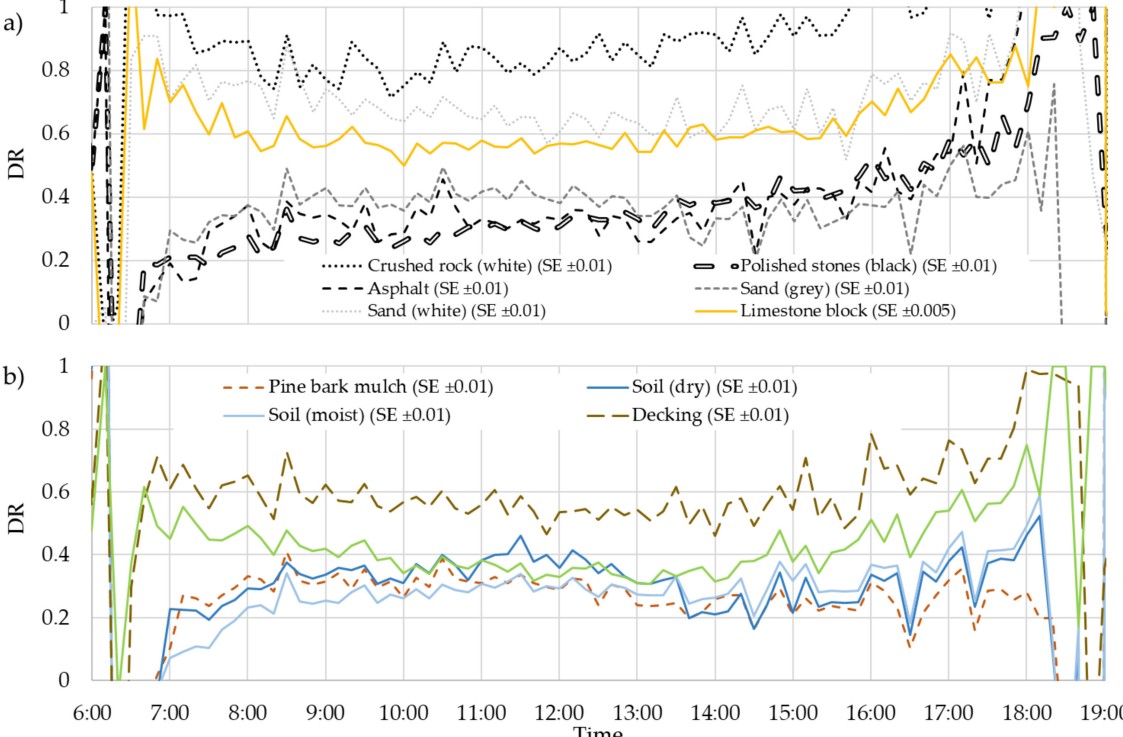

**Figure 10.** Daytime ratio $(T_{iB,LE}-T_{amb})$ over $(T_{iB,WPP}-T_{amb})$ for LEs, summer 2018. Comparison of: (**a**) rocks, sand, and high thermal inertia LEs; and (**b**) softscape or natural LEs. Data from 09:00 to 15:40 were averaged. Standard errors (SE) were calculated assuming random errors and a normal distribution [51].

Relative Average Radiosity

The *DR* data from phase 1 were averaged for each season over the times given in Table 3, to give $DR_{av}$ (data are given in Appendix B). $DR_{av}$ gave a relative ranking for each LE for each season, based on its radiosity (Figure 11). The cross-seasonal average $DR_{av}$ was calculated and is shown in Figure 11 as the thick black line, with the data sorted from lowest to highest according to this average. The standard errors of the measured data for each season and LE were small (around 1%) and are shown as error bars in Figure 11. Hence, the differences seen between the seasonal $DR_{av}$ of each LE are likely to be significant. However, the uncertainties due to the experimental design are likely to be more significant, as discussed previously. On inspection of the $DR_{av}$ data, each LE was observed to lie within a band of around ±0.05 across all seasons (an uncertainty of ±5%). As 5% is not a large uncertainty for real world data, the seasonal differences would not be very significant for all practical purposes. Hence, the relative radiosity from the LEs could be considered to be similar across seasons.

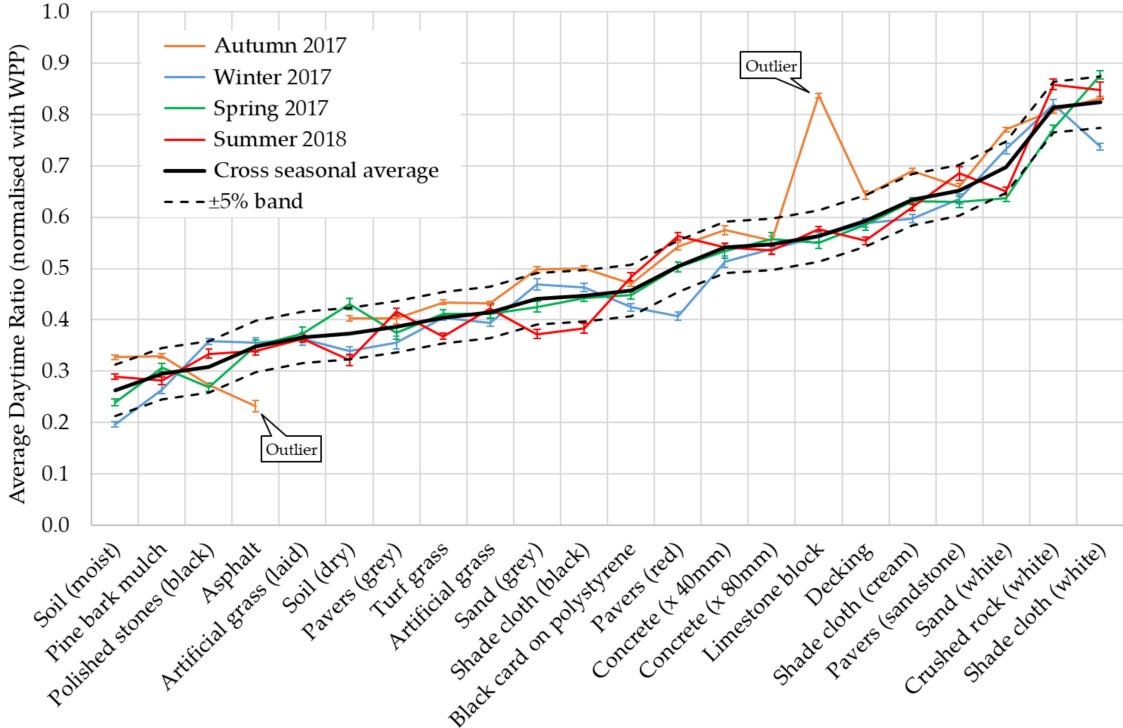

**Figure 11.** Cross-seasonal comparison of the average daytime ratio ($DR_{av}$) of landscape element radiosity normalised with white painted polystyrene (WPP). Data were averaged from ~09:00 to 15:40 depending on the season. Uncertainties are standard errors (SEs) of the averaged data for each LE. Ranked in order of cross-seasonal average. The outliers were not included in the averages.

When calculating the cross-seasonal average for each LE, the autumn data points for limestone block and for asphalt were not included. Both of these LEs have the same $DR_{av}$ in summer, winter, and spring, but their autumn values are quite different and appear to be outliers. The possible reasons for this were that the limestone block was bought new for autumn but then stored outside between seasons. Aging of the new surface is suspected to have occurred, reducing the albedo for subsequent seasons and causing $DR_{av}$ to reduce. Conversely, asphalt had a lower $DR_{av}$ in autumn, as it was newer in this season and was therefore a darker black, becoming lighter in colour after aging outside between seasons. The winter $DR_{av}$ of red pavers and white shade cloth are outside the 5% band, but these data points were retained for the cross-seasonal average calculation as they were not significantly outside the band and their $DR_{av}$ from other seasons were also not consistent. The variance in the red pavers data may have been due to different arrangements of the six unevenly speckled pavers between

seasons. Within the 5% band, the dry and moist soils are also slightly variable across seasons due to their moisture content, which is assumed to have been different, although this was not measured.

As discussed in Section 2.3.5, the effects of the material properties of the LEs can be seen in Figure 11, where white and lighter coloured LEs cause the highest $DR_{av}$ (white shade cloth, white crushed rock, white sand, sandstone pavers, and cream shade cloth). The lowest $DR_{av}$ were from moist soil, pine bark mulch, polished black stones, asphalt, and laid artificial grass, all of which could be classified as having higher surface roughness and/or moisture, and/or having a darker colour.

In general, white and lighter LEs had a higher radiosity than the darker LEs, indicating that during the daytime, SR is the dominant driver of the radiosity from the LEs. Based on the results shown in Figure 11, LEs can be measured at any time of the year using WPP as a reference, and the $DR_{av}$ will provide the relative radiosity of that LE to within ±5%.

Table 5 shows the results from the two-sample t-tests on the LEs ranked next to each other in Figure 11. All LEs, apart from six pairs (*p*-values given in bold), show a statistically significant difference in their means. The negative value of t-Stat indicates that, moving down the table, each mean is larger than the next. Where the *p*-value >0.05, further t-tests were performed, comparing the LE with the two rows beneath it in the table, e.g., asphalt was compared with soil (dry), and soil (dry) was compared with turf grass. These further t-tests all had *p*-values <0.01, indicating that these means were all significantly different, and that the position of each LE in the ranking table would only vary by one place at most.

**Table 5.** Phase 1 seasonal *DR* data two sample t-test (assuming equal variances, α=0.05). Results are a comparison of each landscape element (LE) with the LE in the row beneath it in the table. The bold *p*-values indicate the means of these LEs are not statistically different.

| Phase 1 LEs | $DR_{av}$ (Mean) | Variance | Number of Observations | t-Stat | *p*-Value (Two-Tail) |
|---|---|---|---|---|---|
| Soil (moist) | 0.26 | 0.00 | 147 | −5.00 | <0.001 |
| Pine bark mulch | 0.30 | 0.00 | 147 | −2.27 | 0.02 |
| Polished stones (black) | 0.31 | 0.00 | 147 | −5.26 | <0.001 |
| Asphalt | 0.35 | 0.00 | 113 | −0.36 | **0.72** |
| Artificial grass (laid) | 0.35 | 0.00 | 113 | −2.40 | 0.02 |
| Soil (dry) | 0.37 | 0.01 | 147 | −1.93 | **0.05** |
| Pavers (grey) | 0.39 | 0.00 | 147 | −2.34 | 0.02 |
| Turf grass | 0.40 | 0.00 | 147 | −2.42 | 0.02 |
| Artificial grass | 0.42 | 0.00 | 147 | −3.17 | <0.01 |
| Sand (grey) | 0.44 | 0.01 | 147 | −0.85 | **0.39** |
| Shade cloth (black) | 0.44 | 0.00 | 147 | −1.99 | **0.05** |
| Black card on polystyrene | 0.46 | 0.00 | 147 | −6.46 | <0.001 |
| Pavers (red) | 0.51 | 0.01 | 147 | −4.11 | <0.001 |
| Concrete (× 40 mm) | 0.54 | 0.00 | 147 | −0.75 | **0.45** |
| Concrete (× 80 mm) | 0.55 | 0.00 | 147 | −2.44 | 0.02 |
| Limestone block | 0.56 | 0.00 | 113 | −3.81 | <0.001 |
| Decking | 0.59 | 0.00 | 147 | −6.45 | <0.001 |
| Shade cloth (cream) | 0.63 | 0.00 | 147 | −2.91 | <0.01 |
| Pavers (sandstone) | 0.65 | 0.00 | 147 | −5.14 | <0.001 |
| Sand (white) | 0.69 | 0.01 | 147 | −15.75 | <0.001 |
| Crushed rock (white) | 0.82 | 0.00 | 147 | −1.13 | **0.26** |
| Shade cloth (white) | 0.83 | 0.01 | 147 | −25.82 | <0.001 |
| White painted polystyrene | 1.00 | 0.00 | 147 | | |

Note: All LE t-Critical two-tail = 1.97.

Whilst the focus of the phase 1 test work was a cross-seasonal comparison, the absolute ranking position of each LE was potentially affected by the different ways these LEs were placed in contact with the ground (Section 2.1.1), and the consequential variation in their effective thermal inertia. Although all LEs were buffered from the underlying soil by the turf grass, it is assumed that the air volume underneath each LE would be dependent on its mass, e.g., heavier LEs such as concrete and pavers

would have had the least air volume, whilst lighter LEs such as artificial grass and the shade cloths would have had a greater air volume. Similarly, LEs mounted on the PP alone would also compress the turf grass by an amount dependent on their mass, but with the added thin layer of PP. LEs on PP and then on plastic crates had the greatest air volume (apart from decking). The effect of this changing air volume beneath the LEs was not able to be quantified.

As each LE was in contact with the ground in the same manner across each season, this varying air volume does not affect the cross-seasonal result that the relative radiosity of a particular LE will be the same within ±5% regardless of season. However, the ranking order in Figure 11 may be slightly different than if all of the LEs were coupled in the same manner. To remove the variability associated with ground coupling and to provide a more accurate ranking, the phase 2 investigation was undertaken.

### 3.4. Summer Comparison of LE Size and Ground coupling—Phase 2

Figure 12 shows the temporal daytime ratio data for the larger, ground-coupled LEs measured in summer 2019. Similarly to the higher thermal inertia LEs for the phase 1 LEs, the *DR* of the pavers, asphalt, and concrete showed a slight upwards trend throughout the time period 09:00 to 15:40 (Figure 12a). The slope of this trend appeared to be lower for the phase 2 LEs; however, the increased ground coupling of these LEs meant they were warmer than the smaller LEs at the start of the day. This was confirmed from surface temperature measurements, which showed that these phase 2 LEs were around 4 °C warmer than the phase 1 equivalent LEs just before sunrise (relative to ambient temperature). In contrast, the softscape LEs (Figure 12b) have a fairly consistent *DR* across the day, similar to the equivalent phase 1 LEs.

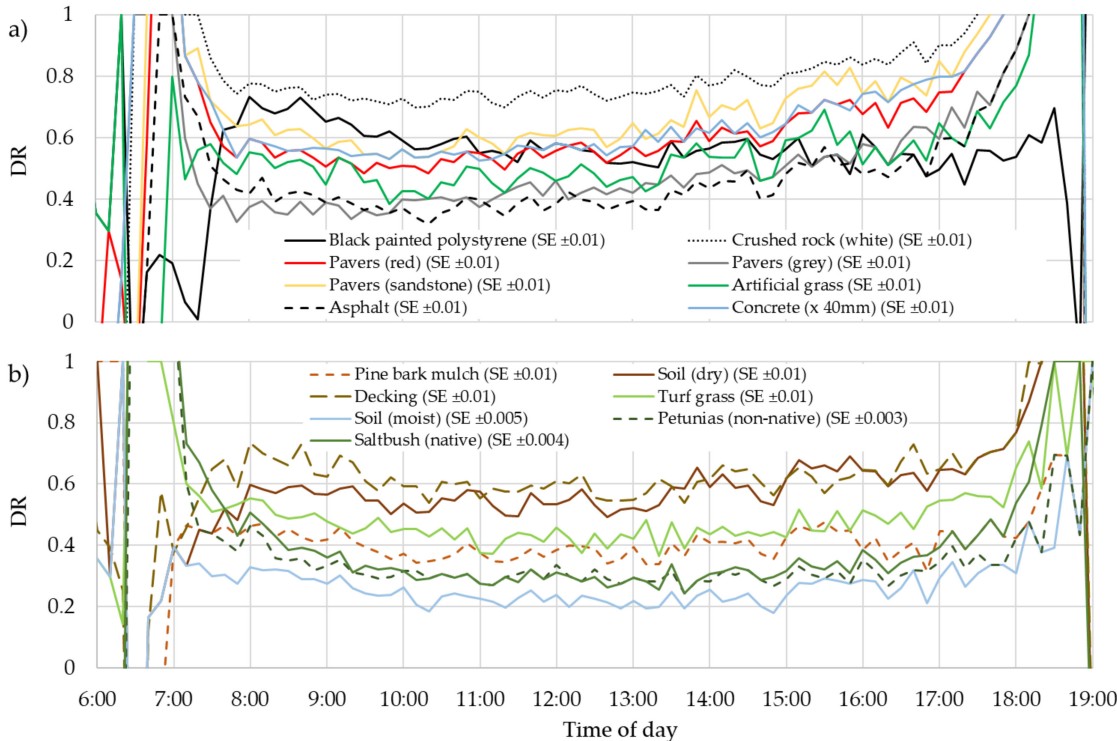

**Figure 12.** Daytime ratio $(T_{iB,LE}-T_{amb})$ over $(T_{iB,WPP}-T_{amb})$ for LEs, summer 2019 (21st January 2019). Comparison of: a) hardscape LEs; and b) softscape or natural LEs. Data from 09:00 to 15:40 were averaged. Standard errors (SE) were calculated assuming random errors and a normal distribution [51].

Figure 13 shows the summer 2019 $DR_{av}$ from phase 2 LEs ordered from lowest to highest. Although the LEs from phase 1 are not all identical to those used in phase 2, they are shown in this figure to give some idea of how they compared. The uncertainties shown are the standard errors for each LE, based

on the measured $DR_{av}$ data. However, the uncertainties in the phase 1 LEs are likely to be higher, as discussed previously in the View Factors section.

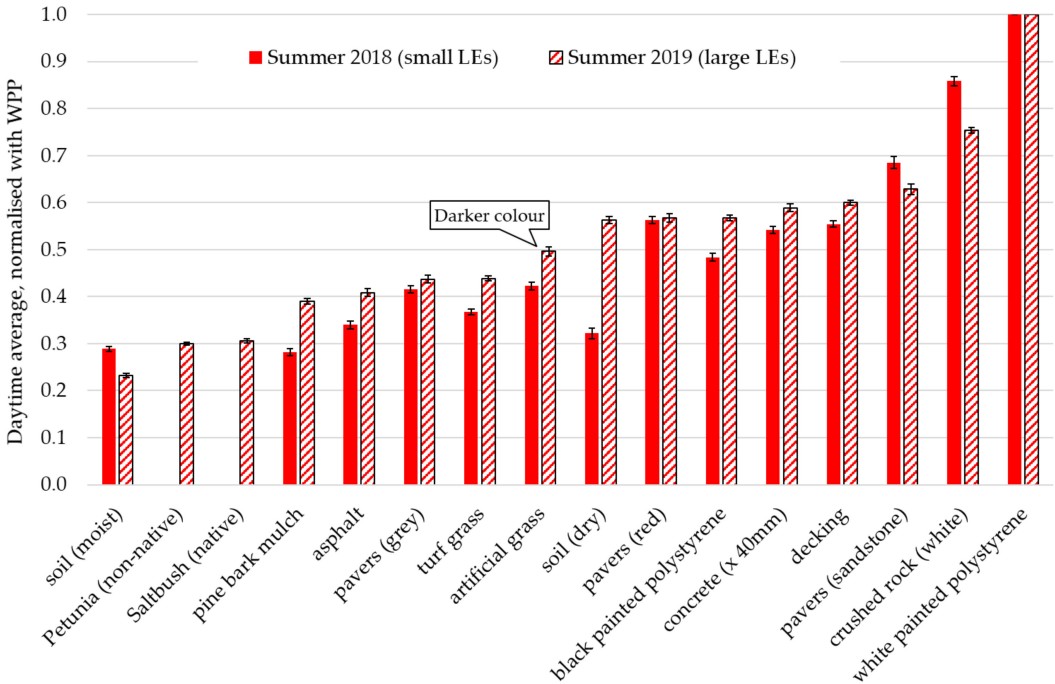

**Figure 13.** Comparison of relative total radiation of LEs when normalised with white painted polystyrene. Summer 2018 and summer (21st January) 2019 $DR_{av}$ (day−time) comparison of size and ground coupling of similar LEs. Uncertainties are the standard errors (SEs).

Although the data are difficult to compare given the variability in product types, the phase 1 LEs generally seem to have a lower $DR_{av}$ than the corresponding similar phase 2 LEs. The *p*-values for this data are given in Table 6. Only the means of grey and red pavers were the same (*p*-values > 0.05). The positive values of t-Stat for moist soil, sandstone pavers, and white rocks indicated that for these LEs, the summer 2018 means were higher than the summer 2019 means. One reason why the 2018 $DR_{av}$ data for most of the LEs are generally lower than 2019 data is that the smaller LEs will have less thermal inertia than the ground-coupled LEs. This means that their response to changes in ambient conditions will be faster and they will follow the WPP changes more closely than the larger LEs will. Hence, the $DR_{av}$ for the smaller LEs would be lower. Another reason is that the LEs are warmer in their central area, but become cooler towards their edges [28]. The surface area of the cooler edges makes up a larger proportion of the total surface area of the small LEs than it does of the larger LEs. The iButtons used in this study measured the whole LE, including the edges. Thus, the thermal radiation received by the iButtons from the smaller LEs will be less than that received by the larger LEs, resulting in a lower $DR_{av}$ of smaller LEs.

The extremes of the phase 2 LEs presented in Figure 13 show that crushed white rock had the highest $DR_{av}$, followed by sandstone pavers and decking, whilst the moist soil and the plants had the lowest. Plants use the incident radiation to evaporate water, which reduces the amount of radiation available for heating, hence the plants and similarly the moist soil are ranked lowest. Apart from moisture content, the $DR_{av}$ seems to be largely dominated by the albedo of the LEs rather than other material properties.

It is believed that the radiosity in the evenings, where no SR is present, would be dominated by the thermal inertia. Evening measurements were taken for all of the LEs across all seasons for this study; however, the analysis was more complex, and initial results indicate that data should be collected over at least three or more evenings in order to account for the faster changing evening ambient temperature, sky temperature, and wind speed. Hence, this data has not been presented here.

In summary, the larger, ground-coupled LEs provided a larger VF of the LE for the iButtons than did the small LEs, and thus assumptions about the background material had an insignificant effect on the accuracy of phase 2 $DR_{av}$. For this reason, the larger LEs are recommended for future radiosity tests, given an iButton height of 200 mm.

**Table 6.** Summer 2018 and summer 2019 *DR* data, two-sample t-test (assuming equal variances, α = 0.05). A negative t-Stat indicates the 2018 mean is less than the 2019 mean. The bold *p*-values indicate the means of these LEs are not statistically different.

| LE | t-Stat | *p*-Value (Two-Tail) |
| --- | --- | --- |
| Soil (moist) | 8.15 | <0.001 |
| Pine bark mulch | −11.48 | <0.001 |
| Asphalt | −5.68 | <0.001 |
| Pavers (grey) | −1.94 | 0.06 |
| Turf grass | −8.75 | <0.001 |
| Artificial grass | −5.50 | <0.001 |
| Soil (dry) | −18.23 | <0.001 |
| Pavers (red) | −0.39 | 0.70 |
| black card on polystyrene | −8.32 | <0.001 |
| Concrete (x 40mm) | −4.13 | <0.001 |
| Decking | −4.87 | <0.001 |
| Pavers (sandstone) | 3.30 | <0.01 |
| Crushed rock (white) | 8.71 | <0.001 |

Note that for all LEs: Number of observations = 41, t Critical two-tail = 1.99.

## 3.5. Ranking LEs

The summer 2019 values of $DR_{av}$ are given in Table 7 and indicate the relative radiosity ranking of the LEs. The results from the two-sample t-tests on the LEs ranked next to each other in Figure 13 are also shown. Six pairs of LEs do not have statistically different means (*p*-values given in bold). However, similar to the phase 1 seasonal tests, the negative value of t Stat indicates that, moving down the table, each mean is larger than the next. Further t-tests were performed on the six LEs, comparing each of these with the LE two rows beneath it in the table. These t-tests all had *p*-values <0.01, apart from dry soil and red pavers. These LEs were further tested against LEs lower in the table. The mean of the dry soil was not significantly different to red pavers or BPP; however, it was significantly different to concrete (x40mm), the next LE in the table (*p*-value = 0.03). Similarly, the result for red pavers was not significantly different to BPP or concrete (x40mm), but was significantly different to decking (*p*-value <0.01). These results indicate that there is some overlap in the ranking order of the LEs in Table 7, particularly in the mid-range of relative radiosities.

The ranking order of the LEs in Table 7 is found to be the same as similar LEs tested by other authors. Cueto et al. [33] found day-time upwelling radiation from six surfaces were ordered from least to most as: grass, asphalt, concrete, clay, and a white painted polystyrene. Similarly, Montague and Kjelgren [15] found their six surfaces to be ordered from least to most as: turf, lava rock mulch, asphalt, pine bark mulch, gravel, and concrete.

Table 7 does not give absolute values for radiosity, hence it is unable to be used for calculating human thermal comfort (HTC) levels or indexes such as the physiological equivalent temperature (PET), which are also affected by other parameters such as wind speed, air temperature, age, and clothing type [54]. However, $DR_{av}$ does provide guidance for choosing a more comfortable LE for a particular location or situation. For example, in areas of high daytime pedestrian use, high radiosity LEs would decrease HTC levels [10,12], and LEs such as plants, mulch or asphalt would be better (depending on the type of usage the surface is required to withstand). Similarly, LEs with high radiosities would not be suitable for use close to buildings, where the higher radiant energy would cause increased building envelope temperatures during the time of occupation [40].

**Table 7.** Daytime relative average radiosity ($DR_{av}$) of larger, ground-coupled LEs from summer 2019, normalised with WPP. Results of the two sample t-test (assuming equal variances, $\alpha$=0.05) are a comparison of each LE with the LE in the row beneath it in the table. The bold *p*-values indicate the means of these LEs are not statistically different.

| Relative Average Radiosity | Daytime (09:00–15:40) | | | | |
|---|---|---|---|---|---|
| | Landscape Element | $DR_{av}$ | SE | t Stat | *p*-Value (Two-Tail) |
| least | Soil (moist) | 0.23 | 0.00 | −11.79 | <0.001 |
| | Petunia (non−native) | 0.30 | 0.00 | −1.04 | **0.30** |
| | Saltbush (native) | 0.31 | 0.00 | −11.46 | <0.001 |
| | Pine bark mulch | 0.39 | 0.01 | −1.85 | **0.07** |
| | Asphalt | 0.41 | 0.01 | −2.39 | 0.02 |
| | Pavers (grey) | 0.44 | 0.01 | −0.18 | **0.86** |
| | Turf grass | 0.44 | 0.01 | −4.95 | <0.001 |
| | Artificial grass | 0.50 | 0.01 | −5.24 | <0.001 |
| | Soil (dry) | 0.56 | 0.01 | −0.36 | **0.72** |
| | Pavers (red) | 0.57 | 0.01 | −0.02 | **0.98** |
| | Black painted polystyrene | 0.57 | 0.01 | −2.05 | 0.04 |
| | Concrete ($\times$ 40 mm) | 0.59 | 0.01 | −1.12 | **0.27** |
| | Decking | 0.60 | 0.01 | −2.23 | 0.03 |
| | Pavers (sandstone) | 0.63 | 0.01 | −9.65 | <0.001 |
| | Crushed rock (white) | 0.75 | 0.01 | −40.77 | <0.001 |
| most | White painted polystyrene | 1.00 | 0.00 | | |

Note: For all LEs: t Critical two-tail = 1.99, Observations = 41, Variance $\leq$ 0.01

Decking seems to have an unusually high $DR_{av}$, perhaps because the surface of the decking, although unfinished, appears to be quite reflective. As the surface ages, or with application of a surface finish, the reflectivity of decking would change, indicating the importance of being able to easily measure $DR_{av}$ at any time. Although moist soil would not be recommended for use as a LE as such, it does provide a good contrast with the dry soil, indicating the important role of evaporative cooling in reducing radiosity. This supports findings by Lindberg et al. [55] that mean radiant temperature in full sun was less over moist grass than over asphalt, likely caused by the presence of moisture, as the albedos were similar. Table 7 shows that plants are the most suitable LEs to use if radiosity is to be minimised in a landscape, with the assumption that they are watered as required.

*3.6. Limitations*

The repeatability of the results would be improved by measurements over longer time periods during each season. Similarly, analysis of evening radiosity measurements indicated that more data should be collected over a number of evenings to improve accuracy. The lower VFs of the phase 1 LEs compared with their VF of the surrounding grass, PP, plastic crate, and nearby LEs, led to greater uncertainties in their $DR_{av}$, uncertainties which were decreased when using the larger LEs.

**4. Conclusions**

Black painted iButton temperature sensors, along with the normalised methodology, performed well as relative low-cost radiometers for measuring multiple LEs concurrently. Between seasons, the ange in values of the normalised average radiosity varied by up to 10% absolutely. Hence, LEs can be measured at any time of the year using WPP as a reference, and the $DR_{av}$ will provide the relative radiosity of that LE to within ±5%. Future work could involve exploring sources of the slight variations between seasons to determine if they are caused by the method (e.g., the size of the LEs), the weather conditions, or by actual physical differences in the LEs between seasons.

When comparing similar material type phase 1 (small) with phase 2 (larger, ground-coupled) LEs, most of the small LEs had lower relative radiosities than the large LEs.

Accuracy of the measured radiosity was improved when using larger, ground-coupled LEs due to the increased $VF_{LE}$ (0.92). It is therefore recommended that larger, ground-coupled LEs be used in future measurements.

Daytime radiosity appears to be driven more by albedo than by LWR, consistent with results by [12]. During the daytime, white and lighter coloured LEs caused the highest iButton temperatures and would therefore have the most detrimental effect on nearby objects such as buildings and/or people in hotter periods, potentially increasing energy costs [56,57], and/or heat stress issues, respectively [10,12,55]. Conversely, white and lighter coloured LEs would reduce urban heat if they were positioned in open spaces or on rooftops, with a high SVF [58,59]. Plants and moist LEs caused the smallest rise in daytime iButton temperature and would be most beneficial for managing daytime urban heat [59].

The use of low-cost iButtons to measure relative radiosity from numerous LEs concurrently presents a new way to examine the relative effect of different LEs on the urban environment. They may be a useful tool when using landscape design for urban heat mitigation.

**Author Contributions:** Conceptualization, J.L., G.K.L., J.J.B. and B.-l.O.; Data curation, J.L.; Formal analysis, J.L. and G.K.L.; Funding acquisition, J.L. and J.J.B.; Investigation, J.L.; Methodology, J.L. and G.K.L.; Project administration, J.L. and J.J.B.; Supervision, J.J.B., B.-l.O. and G.M.M.; Writing—original draft, J.L.; Writing—review & editing, J.L., G.K.L. and G.M.M. All authors have read and agreed to the published version of the manuscript.

**Funding:** This research was supported by an Australian Government Research Training Program Scholarship and a Curtin University Postgraduate Scholarship top up. This research was also partly funded by the CRC for Low Carbon Living Ltd supported by the Cooperative Research Centres program, an Australian Government initiative.

**Acknowledgments:** Appreciation to Bruce Ivers for his assistance in purchasing some LEs and for the use of his vehicle and trailer. Thanks to Grant Bayne for his aerial photography, and to the Booragoon Primary school for the use of their oval.

**Conflicts of Interest:** The authors declare no conflict of interest.

## Nomenclature

| | |
|---|---|
| $DR_{av}$ | Average daytime ratio of the LE's iButton temperature minus $T_{amb}$ over the WPP's iButton temperature minus $T_{amb}$ during the hours where SR > 70% of that occurring at solar noon on that day |
| $\varepsilon_{iB}$ | Emissivity of the painted iButton |
| FOV | Field of view |
| LE | Landscape element; a surface treatment found in a domestic garden or urban landscape |
| LWR | Long-wave radiation; electromagnetic radiation from 2.5 μm–100 μm (W·m$^{-2}$) |
| NIR | Near infra-red radiation, electromagnetic radiation from 0.7 μm–2.5 μm (W·m$^{-2}$) |
| PP | Polypropylene (white Polypropylene bags) |
| Radiosity | total radiant energy from a LE, comprising reflected SR and emitted LWR (W·m$^{-2}$) |
| SE | Standard error |
| SR | Solar radiation (W·m$^{-2}$) |
| SVF | Sky view factor; a measure of the degree of site sky visibility |
| SWR | Short-wave radiation, electromagnetic radiation consisting of ultraviolet, visible and near infra-red radiation ranging from 0.3 μm–2.5 μm (W·m$^{-2}$) |
| $T_{amb}$ | Temperature of ambient air (°C) |
| $T_{iB,Other}$ | Temperature of the iButton due to objects other than the LE, surrounding turf grass and PP bag (°C) |
| $T_{iB,grass}$ | Temperature of the iButton due to the surrounding turf grass (°C) |
| $T_{iB,LE}$ | Temperature of the iButton due to the landscape element (°C) |
| $T_{iB,PP}$ | Temperature of the iButton due to the white polypropylene (°C) |
| $T_{iB,WPP}$ | Temperature of the iButton due to the white painted polystyrene (°C) |
| VF | View factor (subscripts are as for $T_{iB}$) |
| WPP | White painted polystyrene |

## Appendix A

**Table A1.** iButton temperatures due to the landscape element (LE) only (after correction for VFs of surroundings).

| | 09/04/2017, autumn | | | $T_{iB,LE}$ has been corrected for View Factors over turf grass and PP | | | | | | | | | | | | | | | | | | | | |
| --- | --- | --- | --- | --- | --- | --- | --- | --- | --- | --- | --- | --- | --- | --- | --- | --- | --- | --- | --- | --- | --- | --- | --- | --- |
| $T_{iB,LE}$ (°C) | VF of LE | 0.75 | | 0.43 | 0.56 | 0.54 | 0.58 | 0.58 | 0.60 | 0.45 | 0.54 | 0.58 | 0.54 | 0.66 | 0.62 | 0.56 | 0.56 | 0.63 | 0.61 | 0.61 | 0.62 | 0.59 | 0.95 | 0.61 |
| Time | Murdoch $T_{ambient}$ | A | B | C | D | E | F | G | H | I | J | K | L | M | N | O | P | Q | R | S | T | U | V | W |
| 6:00 | 12.3 | 12.3 | | 14.5 | 12.8 | 11.9 | 11.9 | 13.4 | 12.5 | 13.2 | 12.9 | 12.7 | 12.9 | 11.7 | 13.2 | 11.4 | 12.2 | 12.6 | 11.8 | 12.6 | 11.7 | 11.7 | 11.0 | 12.6 |
| 6:10 | 12.0 | 12.0 | | 13.2 | 12.0 | 12.0 | 12.9 | 12.9 | 12.6 | 13.1 | 12.0 | 12.0 | 12.0 | 12.0 | 12.8 | 11.1 | 12.7 | 12.8 | 11.2 | 12.0 | 12.0 | 12.0 | 12.0 | 12.0 |
| 6:20 | 12.4 | 11.5 | | 12.7 | 11.5 | 11.5 | 12.4 | 12.4 | 12.1 | 12.6 | 11.5 | 11.5 | 11.5 | 11.5 | 12.3 | 10.6 | 12.2 | 11.5 | 10.7 | 11.5 | 10.7 | 11.5 | 11.5 | 10.7 |
| 6:30 | 13.0 | 11.0 | | 12.2 | 11.0 | 11.0 | 11.9 | 11.9 | 11.6 | 13.2 | 11.0 | 11.0 | 11.0 | 11.0 | 11.8 | 11.0 | 11.6 | 11.8 | 10.2 | 11.0 | 11.0 | 11.0 | 11.0 | 11.0 |
| 6:40 | 13.2 | 10.3 | | 12.2 | 10.1 | 11.0 | 11.9 | 11.9 | 11.6 | 12.1 | 11.0 | 11.0 | 11.0 | 11.0 | 11.8 | 10.1 | 11.6 | 11.0 | 10.2 | 11.0 | 10.2 | 11.0 | 11.0 | 10.2 |
| 6:50 | 13.4 | 11.0 | | 12.2 | 11.0 | 11.0 | 11.9 | 11.9 | 11.6 | 12.1 | 11.0 | 11.0 | 11.0 | 11.0 | 11.8 | 11.0 | 11.6 | 11.8 | 11.0 | 11.0 | 11.0 | 11.0 | 11.0 | 11.0 |
| 7:00 | 14.0 | 11.5 | | 12.7 | 11.5 | 11.5 | 12.4 | 12.4 | 12.1 | 12.6 | 11.5 | 11.5 | 11.5 | 11.5 | 11.5 | 10.6 | 12.2 | 12.3 | 11.5 | 11.5 | 11.5 | 11.5 | 11.5 | 11.5 |
| 7:10 | 14.4 | 11.5 | | 13.8 | 11.5 | 11.5 | 12.4 | 13.1 | 12.1 | 13.7 | 11.5 | 11.5 | 11.5 | 12.2 | 12.2 | 11.3 | 12.0 | 12.3 | 11.5 | 12.3 | 11.4 | 11.4 | 11.5 | 12.3 |
| 7:20 | 14.8 | 12.7 | | 13.2 | 12.0 | 12.0 | 12.9 | 13.6 | 12.6 | 14.2 | 12.0 | 12.0 | 12.0 | 12.0 | 12.7 | 11.8 | 13.4 | 12.8 | 12.0 | 12.8 | 11.9 | 11.9 | 12.0 | 12.8 |
| 7:30 | 15.3 | 12.2 | | 12.3 | 11.7 | 12.6 | 13.5 | 12.7 | 12.5 | 14.6 | 11.6 | 11.8 | 11.6 | 12.0 | 12.0 | 11.1 | 12.6 | 12.7 | 11.9 | 12.7 | 12.0 | 11.9 | 13.5 | 11.9 |
| 7:40 | 15.7 | 13.0 | | 13.0 | 13.0 | 13.0 | 13.9 | 13.8 | 13.6 | 15.2 | 13.0 | 13.0 | 13.0 | 13.0 | 12.9 | 11.9 | 13.5 | 13.8 | 13.0 | 13.8 | 12.1 | 12.9 | 13.0 | 13.8 |
| 7:50 | 16.1 | 12.5 | | 9.5 | 15.6 | 13.7 | 15.6 | 18.6 | 15.5 | 19.8 | 10.9 | 13.1 | 10.9 | 12.0 | 14.5 | 12.3 | 13.7 | 11.7 | 11.6 | 12.4 | 12.8 | 13.5 | 16.5 | 18.1 |
| 8:00 | 16.6 | 15.2 | | 13.8 | 17.6 | 16.6 | 17.6 | 21.9 | 18.4 | 21.8 | 12.9 | 15.9 | 16.6 | 15.5 | 16.0 | 15.6 | 18.2 | 14.5 | 16.9 | 19.3 | 16.0 | 15.9 | 18.5 | 21.8 |
| 8:10 | 16.9 | 17.3 | | 15.3 | 18.2 | 18.1 | 19.1 | 23.2 | 21.6 | 23.3 | 16.3 | 17.4 | 19.1 | 16.9 | 16.5 | 17.5 | 21.1 | 16.8 | 20.0 | 22.5 | 17.4 | 17.2 | 20.0 | 23.3 |
| 8:20 | 17.3 | 19.0 | | 16.3 | 20.1 | 19.1 | 21.0 | 24.9 | 23.4 | 25.4 | 17.3 | 19.3 | 21.0 | 18.7 | 18.1 | 18.9 | 22.7 | 18.6 | 21.8 | 25.1 | 19.1 | 18.1 | 21.0 | 25.1 |
| 8:30 | 17.5 | 19.8 | | 16.7 | 21.6 | 20.6 | 21.6 | 26.4 | 25.0 | 25.8 | 18.8 | 20.8 | 21.6 | 19.4 | 18.8 | 20.4 | 24.2 | 19.3 | 23.3 | 26.6 | 19.8 | 18.8 | 22.5 | 26.6 |
| 8:40 | 17.6 | 20.8 | | 17.7 | 22.6 | 21.6 | 22.6 | 27.4 | 26.0 | 27.9 | 19.8 | 21.8 | 23.5 | 20.4 | 19.8 | 21.4 | 26.1 | 20.3 | 24.3 | 27.6 | 20.8 | 19.8 | 23.5 | 28.4 |
| 8:50 | 18.2 | 22.0 | | 19.3 | 24.0 | 24.0 | 24.0 | 29.4 | 28.1 | 29.6 | 21.2 | 23.1 | 25.9 | 21.6 | 20.9 | 23.3 | 27.1 | 22.4 | 26.5 | 29.7 | 21.9 | 21.0 | 24.0 | 30.6 |
| 9:00 | 18.5 | 23.0 | | 20.3 | 25.0 | 25.0 | 25.0 | 30.4 | 29.2 | 30.6 | 22.2 | 24.1 | 26.9 | 22.6 | 21.9 | 24.3 | 28.1 | 23.4 | 26.6 | 30.7 | 22.9 | 22.0 | 25.0 | 32.4 |
| 9:10 | 18.8 | 24.7 | | 20.2 | 26.0 | 26.0 | 26.0 | 31.4 | 30.2 | 31.6 | 23.2 | 25.1 | 27.9 | 23.6 | 22.9 | 25.3 | 30.1 | 26.0 | 27.6 | 31.7 | 23.9 | 22.1 | 26.0 | 33.4 |
| 9:20 | 19.1 | 24.5 | | 20.7 | 25.6 | 26.5 | 26.5 | 32.0 | 30.7 | 32.1 | 23.7 | 25.6 | 28.4 | 23.4 | 22.7 | 26.0 | 30.8 | 25.7 | 28.1 | 31.4 | 24.5 | 23.5 | 26.5 | 33.9 |
| 9:30 | 19.4 | 26.2 | | 21.7 | 27.5 | 27.5 | 27.5 | 33.8 | 31.7 | 33.1 | 24.7 | 27.5 | 29.4 | 24.4 | 23.6 | 26.8 | 31.6 | 27.5 | 30.0 | 33.2 | 25.4 | 23.6 | 27.5 | 35.7 |
| 9:40 | 19.8 | 27.3 | | 23.3 | 28.0 | 28.9 | 28.9 | 35.1 | 33.0 | 34.7 | 26.1 | 28.9 | 30.8 | 25.6 | 24.8 | 27.9 | 33.7 | 29.6 | 32.1 | 34.6 | 26.7 | 24.9 | 28.0 | 37.8 |

**Table A1.** *Cont.*

| 09/04/2017, autumn | | | | $T_{iB,LE}$ has been corrected for View Factors over turf grass and PP | | | | | | | | | | | | | | | | | | | |
|---|---|---|---|---|---|---|---|---|---|---|---|---|---|---|---|---|---|---|---|---|---|---|---|---|
| $T_{iB,LE}$ (°C) | VF of LE | 0.75 | | 0.43 | 0.56 | 0.54 | 0.58 | 0.58 | 0.60 | 0.45 | 0.54 | 0.58 | 0.54 | 0.66 | 0.62 | 0.56 | 0.56 | 0.63 | 0.61 | 0.61 | 0.62 | 0.59 | 0.95 | 0.61 |
| Time | Murdoch $T_{ambient}$ | A | B | C | D | E | F | G | H | I | J | K | L | M | N | O | P | Q | R | S | T | U | V | W |
| 9:50 | 20.1 | 28.3 | | 23.2 | 29.0 | 29.9 | 29.9 | 36.1 | 34.0 | 35.7 | 26.2 | 29.0 | 31.8 | 25.9 | 25.0 | 28.9 | 34.7 | 30.6 | 33.1 | 35.6 | 27.7 | 25.9 | 29.0 | 38.8 |
| 10:00 | 20.5 | 29.5 | | 23.7 | 29.5 | 30.4 | 30.4 | 37.4 | 34.5 | 36.2 | 26.7 | 30.4 | 33.2 | 27.1 | 25.4 | 30.1 | 35.0 | 31.1 | 34.4 | 36.9 | 28.1 | 26.3 | 29.5 | 40.2 |
| 10:10 | 20.9 | 29.8 | | 25.8 | 30.5 | 31.4 | 31.4 | 38.4 | 35.5 | 38.3 | 27.7 | 31.4 | 33.3 | 28.1 | 26.4 | 31.1 | 36.9 | 31.3 | 36.2 | 37.9 | 29.1 | 27.3 | 30.5 | 41.2 |
| 10:20 | 21.2 | 29.8 | | 24.7 | 30.5 | 32.4 | 31.4 | 38.4 | 35.5 | 38.3 | 28.6 | 31.4 | 34.2 | 27.4 | 26.4 | 31.1 | 36.9 | 31.3 | 36.2 | 37.9 | 29.1 | 27.3 | 30.5 | 42.0 |
| 10:30 | 21.3 | 31.5 | | 25.7 | 31.5 | 33.4 | 33.2 | 39.3 | 36.6 | 39.3 | 29.6 | 32.4 | 35.2 | 29.1 | 27.3 | 32.7 | 38.6 | 33.1 | 37.2 | 39.7 | 30.0 | 28.3 | 31.5 | 43.0 |
| 10:40 | 22.0 | 30.2 | | 25.7 | 30.6 | 33.4 | 33.2 | 39.4 | 35.7 | 39.3 | 29.6 | 32.4 | 35.2 | 28.4 | 27.4 | 32.1 | 37.9 | 32.3 | 36.4 | 38.9 | 29.3 | 28.3 | 31.5 | 43.0 |
| 10:50 | 22.3 | 31.3 | | 25.0 | 31.1 | 33.9 | 33.7 | 39.8 | 36.2 | 39.8 | 30.1 | 32.9 | 35.7 | 28.8 | 27.8 | 32.3 | 38.2 | 32.8 | 37.7 | 40.2 | 29.7 | 28.8 | 32.0 | 43.5 |
| 11:00 | 23.0 | 31.8 | | 26.7 | 32.5 | 34.4 | 34.2 | 41.1 | 37.6 | 41.4 | 30.6 | 34.2 | 37.1 | 29.3 | 28.3 | 33.5 | 39.5 | 33.3 | 38.2 | 41.5 | 31.0 | 29.2 | 32.5 | 44.8 |
| 11:10 | 23.0 | 32.8 | | 26.5 | 33.5 | 35.4 | 35.2 | 41.2 | 37.7 | 42.4 | 31.6 | 34.4 | 37.2 | 29.6 | 29.3 | 33.6 | 40.5 | 34.3 | 39.2 | 42.5 | 31.2 | 30.2 | 33.5 | 45.8 |
| 11:20 | 23.2 | 32.8 | | 27.7 | 33.5 | 36.3 | 35.2 | 42.0 | 38.6 | 42.4 | 31.6 | 35.2 | 38.1 | 30.3 | 29.2 | 34.3 | 41.2 | 34.3 | 40.1 | 43.3 | 31.9 | 31.0 | 33.5 | 46.6 |
| 11:30 | 23.6 | 33.5 | | 27.7 | 33.5 | 36.3 | 35.2 | 42.0 | 37.7 | 42.4 | 32.6 | 36.1 | 38.1 | 30.3 | 29.2 | 34.3 | 41.2 | 35.1 | 40.1 | 43.3 | 31.9 | 31.0 | 33.5 | 46.6 |
| 11:40 | 24.0 | 34.7 | | 29.3 | 35.8 | 37.7 | 36.6 | 44.1 | 39.1 | 44.0 | 33.1 | 37.4 | 39.6 | 31.6 | 30.4 | 35.4 | 42.4 | 36.4 | 40.6 | 44.7 | 33.1 | 31.4 | 34.0 | 48.8 |
| 11:50 | 24.5 | 35.2 | | 29.8 | 36.3 | 38.2 | 37.1 | 43.7 | 39.6 | 44.5 | 33.6 | 37.1 | 40.1 | 32.1 | 30.9 | 35.9 | 42.9 | 36.9 | 41.9 | 45.2 | 34.4 | 32.7 | 34.5 | 49.3 |
| 12:00 | 24.8 | 35.7 | | 30.3 | 36.8 | 38.7 | 37.6 | 44.2 | 40.1 | 45.0 | 34.1 | 38.4 | 40.6 | 32.6 | 31.4 | 36.4 | 43.4 | 37.4 | 41.6 | 45.7 | 34.9 | 32.4 | 35.0 | 49.8 |
| 12:10 | 24.9 | 35.0 | | 29.2 | 35.0 | 37.8 | 37.6 | 43.5 | 39.2 | 43.9 | 34.1 | 37.6 | 40.6 | 32.6 | 30.7 | 35.8 | 42.7 | 35.8 | 41.6 | 44.8 | 34.2 | 32.5 | 35.0 | 48.9 |
| 12:20 | 25.4 | 36.2 | | 30.8 | 37.3 | 39.2 | 38.9 | 44.7 | 40.6 | 45.5 | 34.6 | 38.9 | 41.1 | 33.8 | 31.9 | 37.8 | 44.8 | 37.1 | 42.1 | 46.2 | 35.4 | 32.9 | 35.5 | 50.3 |
| 12:30 | 25.1 | 36.2 | | 30.8 | 36.4 | 39.2 | 38.9 | 44.8 | 40.6 | 45.5 | 34.6 | 38.1 | 41.1 | 33.1 | 32.0 | 37.2 | 44.1 | 37.9 | 42.1 | 45.3 | 35.5 | 33.8 | 35.5 | 49.4 |
| 12:40 | 25.4 | 35.7 | | 31.5 | 36.8 | 39.6 | 38.4 | 44.2 | 40.1 | 45.0 | 36.9 | 38.4 | 41.5 | 33.3 | 32.2 | 37.3 | 44.3 | 37.4 | 42.4 | 45.7 | 34.9 | 33.2 | 35.0 | 49.8 |
| 12:50 | 25.5 | 36.7 | | 31.3 | 37.8 | 39.7 | 39.4 | 45.3 | 41.1 | 46.0 | 37.9 | 39.4 | 41.6 | 34.3 | 32.5 | 38.5 | 45.5 | 38.4 | 42.6 | 45.8 | 36.0 | 34.3 | 36.0 | 49.9 |
| 13:00 | 26.1 | 36.7 | | 31.3 | 37.8 | 39.7 | 39.4 | 44.5 | 40.3 | 46.0 | 37.9 | 39.4 | 41.6 | 33.6 | 32.5 | 37.7 | 43.7 | 37.6 | 42.6 | 45.8 | 36.0 | 34.3 | 36.0 | 49.9 |
| 13:10 | 26.0 | 36.8 | | 33.2 | 38.2 | 40.1 | 39.8 | 45.6 | 40.6 | 46.6 | 37.4 | 39.8 | 42.0 | 33.8 | 32.7 | 37.8 | 43.9 | 38.7 | 42.9 | 46.2 | 36.3 | 34.6 | 35.5 | 50.3 |
| 13:20 | 26.2 | 36.7 | | 32.5 | 37.8 | 40.6 | 39.4 | 44.5 | 41.1 | 46.0 | 38.8 | 40.3 | 42.5 | 34.3 | 33.3 | 38.5 | 44.6 | 38.4 | 41.7 | 45.8 | 36.8 | 34.3 | 36.0 | 49.9 |
| 13:30 | 26.1 | 36.0 | | 32.5 | 36.9 | 39.7 | 39.4 | 44.6 | 40.3 | 44.9 | 37.9 | 39.4 | 41.6 | 34.3 | 32.6 | 37.9 | 43.9 | 37.6 | 41.7 | 45.0 | 36.1 | 33.5 | 36.0 | 49.1 |
| 13:40 | 26.2 | 36.0 | | 32.5 | 37.8 | 40.6 | 40.3 | 44.5 | 40.3 | 46.0 | 36.0 | 39.4 | 42.5 | 34.3 | 33.3 | 38.5 | 43.7 | 37.6 | 42.6 | 45.8 | 35.2 | 34.3 | 36.0 | 49.1 |
| 13:50 | 26.6 | 35.3 | | 31.3 | 36.0 | 39.7 | 39.4 | 43.8 | 39.4 | 44.9 | 35.1 | 38.6 | 41.6 | 33.6 | 32.7 | 37.2 | 43.2 | 36.8 | 40.9 | 44.2 | 35.3 | 34.5 | 36.0 | 48.3 |
| 14:00 | 26.8 | 36.7 | | 33.7 | 37.8 | 40.6 | 40.3 | 43.7 | 40.3 | 46.0 | 36.0 | 39.4 | 41.6 | 35.1 | 33.4 | 38.8 | 43.9 | 37.6 | 41.7 | 45.0 | 36.9 | 35.2 | 36.0 | 48.3 |

**Table A1.** *Cont.*

| 09/04/2017, autumn | | | | $T_{iB,LE}$ has been corrected for View Factors over turf grass and PP | | | | | | | | | | | | | | | | | | | |
|---|---|---|---|---|---|---|---|---|---|---|---|---|---|---|---|---|---|---|---|---|---|---|---|---|
| $T_{iB,LE}$ (°C) | VF of LE | 0.75 | | 0.43 | 0.56 | 0.54 | 0.58 | 0.58 | 0.60 | 0.45 | 0.54 | 0.58 | 0.54 | 0.66 | 0.62 | 0.56 | 0.56 | 0.63 | 0.61 | 0.61 | 0.62 | 0.59 | 0.95 | 0.61 |
| Time | Murdoch $T_{ambient}$ | A | B | C | D | E | F | G | H | I | J | K | L | M | N | O | P | Q | R | S | T | U | V | W |
| 14:10 | 27.1 | 36.0 | | 33.7 | 36.9 | 39.7 | 39.4 | 43.8 | 40.3 | 44.9 | 35.1 | 38.6 | 41.6 | 35.1 | 32.7 | 38.1 | 43.2 | 37.6 | 40.9 | 44.2 | 36.1 | 34.5 | 36.0 | 48.3 |
| 14:20 | 26.9 | 35.5 | | 33.2 | 36.4 | 40.1 | 39.8 | 43.3 | 39.7 | 44.4 | 35.5 | 38.9 | 41.1 | 33.9 | 33.0 | 37.6 | 42.7 | 36.3 | 40.4 | 43.7 | 35.6 | 34.0 | 35.5 | 47.0 |
| 14:30 | 27.2 | 36.2 | | 34.3 | 37.3 | 40.1 | 39.8 | 42.5 | 39.7 | 44.4 | 35.5 | 38.9 | 41.1 | 34.6 | 33.1 | 37.9 | 42.9 | 37.1 | 40.4 | 42.9 | 35.7 | 34.0 | 35.5 | 46.2 |
| 14:40 | 26.9 | 34.5 | | 32.2 | 35.4 | 39.1 | 37.9 | 41.5 | 37.9 | 42.3 | 34.5 | 37.9 | 40.1 | 32.9 | 32.1 | 36.9 | 41.9 | 35.3 | 38.6 | 41.9 | 34.7 | 33.0 | 34.5 | 45.2 |
| 14:50 | 27.2 | 34.3 | | 32.7 | 35.9 | 38.7 | 38.4 | 41.2 | 37.6 | 42.8 | 35.0 | 37.6 | 40.6 | 32.6 | 32.6 | 36.7 | 40.8 | 35.8 | 38.3 | 41.6 | 34.5 | 33.6 | 35.0 | 44.8 |
| 15:00 | 27.4 | 35.2 | | 33.3 | 36.3 | 38.2 | 37.9 | 40.7 | 37.9 | 42.3 | 35.4 | 37.9 | 40.1 | 33.6 | 32.1 | 37.1 | 41.1 | 36.1 | 38.6 | 41.1 | 34.8 | 33.1 | 34.5 | 44.3 |
| 15:10 | 27.3 | 34.0 | | 32.8 | 34.9 | 37.7 | 37.4 | 39.4 | 36.6 | 40.7 | 34.0 | 36.6 | 38.6 | 32.4 | 31.7 | 35.9 | 39.0 | 34.8 | 37.3 | 39.7 | 33.5 | 32.7 | 34.0 | 42.2 |
| 15:20 | 27.1 | 33.5 | | 32.3 | 33.5 | 37.2 | 36.9 | 38.9 | 36.1 | 40.2 | 33.5 | 36.1 | 38.1 | 31.9 | 31.2 | 35.4 | 38.5 | 34.3 | 36.8 | 39.2 | 32.2 | 32.2 | 33.5 | 41.7 |
| 15:30 | 27.0 | 31.8 | | 31.3 | 32.5 | 35.3 | 35.9 | 37.2 | 34.2 | 38.1 | 32.5 | 35.1 | 37.1 | 30.9 | 30.3 | 33.8 | 36.8 | 32.5 | 34.1 | 37.4 | 30.5 | 30.4 | 32.5 | 39.9 |
| 15:40 | 27.4 | 30.8 | | 30.3 | 30.6 | 34.3 | 34.9 | 35.4 | 33.2 | 37.1 | 31.5 | 33.2 | 35.2 | 29.9 | 29.4 | 33.0 | 36.0 | 30.7 | 32.3 | 35.6 | 30.4 | 29.5 | 31.5 | 38.1 |
| 15:50 | 27.2 | 29.2 | | 29.3 | 29.6 | 33.3 | 33.1 | 34.4 | 31.4 | 34.9 | 31.4 | 32.2 | 34.2 | 28.9 | 29.2 | 31.1 | 34.1 | 29.7 | 31.3 | 34.6 | 28.6 | 28.5 | 30.5 | 36.2 |
| 16:00 | 27.0 | 28.2 | | 28.3 | 28.6 | 32.3 | 32.9 | 33.5 | 30.3 | 35.1 | 30.4 | 31.2 | 33.2 | 27.9 | 28.3 | 30.3 | 33.3 | 28.7 | 30.3 | 32.8 | 27.6 | 27.5 | 29.5 | 35.2 |
| 16:10 | 27.0 | 27.2 | | 27.3 | 27.6 | 30.4 | 31.1 | 31.6 | 29.3 | 31.8 | 29.4 | 30.2 | 32.2 | 26.9 | 27.3 | 29.3 | 31.4 | 26.9 | 28.5 | 31.8 | 26.6 | 26.5 | 28.5 | 32.6 |
| 16:20 | 27.1 | 26.2 | | 26.3 | 25.7 | 29.4 | 30.1 | 29.8 | 27.5 | 30.8 | 27.5 | 28.4 | 30.3 | 25.2 | 26.4 | 27.7 | 29.6 | 25.9 | 26.7 | 30.0 | 25.7 | 25.6 | 27.5 | 30.8 |
| 16:30 | 25.3 | 25.2 | | 25.3 | 25.6 | 28.4 | 29.1 | 28.9 | 26.5 | 29.8 | 27.4 | 27.4 | 29.3 | 25.0 | 25.5 | 26.9 | 28.8 | 25.7 | 26.5 | 28.1 | 24.8 | 24.7 | 26.5 | 29.8 |
| 16:40 | 24.2 | 24.8 | | 25.5 | 24.6 | 27.4 | 28.1 | 27.9 | 26.3 | 28.8 | 26.4 | 27.2 | 28.3 | 24.7 | 25.3 | 25.9 | 27.8 | 24.7 | 25.5 | 27.1 | 24.6 | 24.5 | 25.5 | 28.0 |
| 16:50 | 23.8 | 23.8 | | 24.5 | 23.6 | 26.4 | 27.1 | 26.9 | 25.3 | 27.8 | 26.4 | 26.2 | 27.3 | 23.7 | 24.3 | 24.9 | 26.8 | 24.5 | 24.5 | 26.1 | 23.6 | 23.5 | 24.5 | 27.0 |
| 17:00 | 23.6 | 23.3 | | 25.2 | 23.1 | 25.9 | 26.6 | 25.6 | 24.0 | 26.2 | 24.9 | 24.9 | 26.8 | 23.2 | 23.9 | 24.7 | 25.6 | 24.0 | 23.2 | 24.8 | 23.1 | 23.1 | 24.0 | 25.6 |
| 17:10 | 23.5 | 22.3 | | 24.2 | 22.1 | 24.9 | 25.6 | 24.6 | 22.9 | 25.2 | 24.9 | 23.9 | 25.8 | 22.2 | 22.9 | 23.7 | 24.6 | 23.0 | 22.2 | 23.8 | 22.1 | 22.1 | 23.0 | 23.8 |
| 17:20 | 23.5 | 20.7 | | 24.3 | 20.2 | 24.8 | 25.4 | 22.1 | 22.8 | 24.2 | 22.9 | 22.0 | 23.9 | 22.0 | 22.1 | 23.1 | 23.9 | 21.2 | 20.4 | 21.2 | 22.1 | 22.1 | 22.0 | 20.4 |
| 17:30 | 22.8 | 19.0 | | 23.3 | 19.2 | 23.8 | 24.4 | 21.2 | 21.8 | 23.2 | 21.9 | 21.9 | 22.9 | 21.0 | 22.0 | 22.4 | 23.1 | 20.2 | 18.5 | 19.4 | 21.1 | 21.1 | 21.0 | 19.4 |
| 17:40 | 22.1 | 19.2 | | 22.8 | 19.6 | 22.4 | 23.1 | 21.4 | 21.2 | 22.7 | 22.4 | 21.4 | 22.4 | 20.5 | 21.4 | 20.7 | 21.5 | 19.7 | 18.9 | 19.7 | 20.6 | 20.6 | 20.5 | 19.7 |
| 17:50 | 21.6 | 18.2 | | 21.8 | 18.6 | 21.4 | 22.9 | 20.4 | 20.2 | 21.7 | 21.4 | 20.4 | 21.4 | 19.5 | 21.2 | 19.7 | 20.5 | 18.7 | 17.9 | 18.7 | 19.6 | 19.6 | 19.5 | 17.9 |
| 18:00 | 21.0 | 17.2 | | 20.8 | 17.6 | 20.4 | 21.9 | 19.4 | 19.2 | 21.8 | 20.4 | 19.4 | 20.4 | 18.5 | 20.2 | 18.7 | 19.5 | 17.7 | 16.9 | 17.7 | 18.6 | 18.6 | 18.5 | 16.9 |
| 18:10 | 20.6 | 16.8 | | 21.0 | 17.5 | 20.3 | 21.8 | 18.4 | 18.2 | 20.8 | 20.3 | 19.2 | 20.3 | 18.3 | 19.1 | 18.4 | 19.2 | 17.5 | 16.7 | 17.5 | 18.3 | 18.3 | 17.5 | 16.7 |
| 18:20 | 20.3 | 16.2 | | 19.8 | 16.6 | 19.4 | 20.9 | 18.5 | 18.2 | 20.8 | 19.4 | 18.4 | 19.4 | 16.8 | 19.3 | 17.1 | 18.6 | 16.7 | 15.0 | 15.9 | 17.6 | 17.6 | 17.5 | 15.9 |
| 18:30 | 20.2 | 14.7 | | 19.5 | 16.0 | 19.7 | 21.2 | 17.8 | 17.5 | 20.4 | 18.8 | 17.7 | 18.8 | 16.0 | 18.5 | 17.1 | 17.8 | 16.0 | 14.4 | 15.2 | 16.9 | 16.9 | 16.0 | 15.2 |

**Table A1.** *Cont.*

| 09/04/2017, autumn | | | | | | | | | | | | | | | | | | | | | | | | |
|---|---|---|---|---|---|---|---|---|---|---|---|---|---|---|---|---|---|---|---|---|---|---|---|---|
| $T_{iB,LE}$ (°C) VF of LE | 0.75 | | 0.43 | 0.56 | 0.54 | 0.58 | 0.58 | 0.60 | 0.45 | 0.54 | 0.58 | 0.54 | 0.66 | 0.62 | 0.56 | 0.56 | 0.63 | 0.61 | 0.61 | 0.62 | 0.59 | 0.95 | 0.61 |
| Time | Murdoch $T_{ambient}$ | A | B | C | D | E | F | G | H | I | J | K | L | M | N | O | P | Q | R | S | T | U | V | W |
| 18:40 | 20.0 | 14.7 | | 19.5 | 15.1 | 18.8 | 20.3 | 16.9 | 16.7 | 19.3 | 17.9 | 17.7 | 17.9 | 16.0 | 17.7 | 16.2 | 16.9 | 15.2 | 14.4 | 15.2 | 16.1 | 16.9 | 16.0 | 15.2 |
| 18:50 | 19.6 | 15.0 | | 19.7 | 15.0 | 17.8 | 19.3 | 16.7 | 16.5 | 19.4 | 17.8 | 17.6 | 17.8 | 15.8 | 17.4 | 15.9 | 16.6 | 15.8 | 14.2 | 15.0 | 15.8 | 15.8 | 15.0 | 15.0 |
| 19:00 | 19.0 | 13.8 | | 18.0 | 13.6 | 17.3 | 18.8 | 15.4 | 15.2 | 17.8 | 16.4 | 16.2 | 16.4 | 14.5 | 16.2 | 14.7 | 15.4 | 13.7 | 12.9 | 13.7 | 14.6 | 14.6 | 14.5 | 13.7 |
| 19:10 | 18.5 | 12.8 | | 17.0 | 13.5 | 16.3 | 17.8 | 15.3 | 15.0 | 17.9 | 16.3 | 15.2 | 16.3 | 13.5 | 16.0 | 14.6 | 15.3 | 12.7 | 11.9 | 12.7 | 14.4 | 14.4 | 13.5 | 12.7 |
| 19:20 | 18.0 | 12.3 | | 16.5 | 12.1 | 15.8 | 17.3 | 14.8 | 13.6 | 17.4 | 14.9 | 14.7 | 14.9 | 13.0 | 15.5 | 14.1 | 14.8 | 12.2 | 11.4 | 12.2 | 13.1 | 13.9 | 13.0 | 12.2 |
| 19:30 | 17.9 | 11.7 | | 15.3 | 12.1 | 14.9 | 17.3 | 14.8 | 13.6 | 17.4 | 14.9 | 14.7 | 14.9 | 13.0 | 14.7 | 13.2 | 13.9 | 12.2 | 11.4 | 12.2 | 13.1 | 13.1 | 13.0 | 12.2 |
| 19:40 | 17.3 | 11.8 | | 17.2 | 11.6 | 15.3 | 16.8 | 14.3 | 14.0 | 16.9 | 14.4 | 14.2 | 14.4 | 13.3 | 15.0 | 13.6 | 14.3 | 12.5 | 11.7 | 11.7 | 13.4 | 14.3 | 12.5 | 11.7 |
| 19:50 | 17.2 | 12.5 | | 17.2 | 12.5 | 15.3 | 17.7 | 14.2 | 14.0 | 18.1 | 15.3 | 15.1 | 15.3 | 13.3 | 14.9 | 13.4 | 14.1 | 12.5 | 12.5 | 12.5 | 13.3 | 14.2 | 12.5 | 12.5 |
| 20:00 | 17.4 | 12.5 | | 16.0 | 12.5 | 15.3 | 16.8 | 14.2 | 14.0 | 16.9 | 14.4 | 14.2 | 15.3 | 13.3 | 14.9 | 13.4 | 14.1 | 12.5 | 12.5 | 12.5 | 13.3 | 14.2 | 12.5 | 12.5 |
| 20:10 | 17.4 | 12.3 | | 15.3 | 12.1 | 14.9 | 16.4 | 13.9 | 13.6 | 16.3 | 14.9 | 14.7 | 14.9 | 13.0 | 14.6 | 13.0 | 14.6 | 12.2 | 11.4 | 13.0 | 13.0 | 13.8 | 13.0 | 12.2 |
| 20:20 | 17.0 | 11.3 | | 15.5 | 12.0 | 14.8 | 16.3 | 13.8 | 12.6 | 16.4 | 13.9 | 13.7 | 13.9 | 12.0 | 13.7 | 13.1 | 13.8 | 11.2 | 10.4 | 11.2 | 12.1 | 12.9 | 12.0 | 12.0 |
| 20:30 | 16.3 | 10.8 | | 15.0 | 12.4 | 14.3 | 15.8 | 13.2 | 13.0 | 17.1 | 13.4 | 13.2 | 13.4 | 11.5 | 13.9 | 12.4 | 13.1 | 11.5 | 10.7 | 11.5 | 12.3 | 13.2 | 11.5 | 11.5 |
| 20:40 | 16.3 | 10.8 | | 13.8 | 11.5 | 13.4 | 14.9 | 13.3 | 12.1 | 15.9 | 13.4 | 12.4 | 13.4 | 11.5 | 13.2 | 11.7 | 13.3 | 10.7 | 9.9 | 10.7 | 11.6 | 12.4 | 11.5 | 11.5 |
| 20:50 | 15.7 | 11.3 | | 14.3 | 11.1 | 13.9 | 15.4 | 12.9 | 12.6 | 15.3 | 12.9 | 12.9 | 12.9 | 12.0 | 13.7 | 12.2 | 12.9 | 11.2 | 10.4 | 11.2 | 12.1 | 12.9 | 12.0 | 11.2 |
| 21:00 | 15.9 | 12.5 | | 14.8 | 11.6 | 14.4 | 15.9 | 13.4 | 13.1 | 15.8 | 13.4 | 13.4 | 13.4 | 13.3 | 13.3 | 13.4 | 14.1 | 12.5 | 11.7 | 12.5 | 12.5 | 13.3 | 12.5 | 11.7 |
| 24/08/2017, winter | | | | | | | | | | | | | | | | | | | | | | | | |
| $T_{iB,LE}$ (°C) VF of LE | 0.71 | 0.58 | 0.43 | 0.56 | 0.52 | 0.58 | 0.58 | 0.56 | 0.45 | 0.54 | 0.59 | 0.54 | 0.65 | 0.62 | 0.56 | 0.56 | 0.63 | 0.61 | 0.61 | 0.62 | 0.59 | 0.95 | 0.61 |
| Time | Murdoch $T_{ambient}$ | A | B | C | D | E | F | G | H | I | J | K | L | M | N | O | P | Q | R | S | T | U | V | W |
| 5:30 | 10.2 | 5.8 | 6.5 | 7.7 | 5.6 | 7.5 | 7.4 | 6.6 | 6.2 | 7.6 | 6.5 | 6.5 | 6.5 | 5.8 | 7.4 | 5.8 | 7.2 | 6.5 | 5.7 | 5.7 | 5.8 | 6.6 | 6.5 | 5.7 |
| 5:40 | 10.5 | 6.0 | 5.8 | 8.3 | 6.0 | 7.0 | 6.9 | 6.9 | 6.6 | 8.2 | 6.9 | 6.8 | 6.9 | 6.8 | 6.8 | 6.9 | 7.4 | 7.6 | 6.0 | 6.0 | 6.0 | 6.8 | 6.0 | 5.2 |
| 5:50 | 10.5 | 5.3 | 5.8 | 7.2 | 6.0 | 7.0 | 6.9 | 6.9 | 6.6 | 8.2 | 6.9 | 6.0 | 6.9 | 5.2 | 6.8 | 5.1 | 6.5 | 6.8 | 5.2 | 6.0 | 5.2 | 6.0 | 6.0 | 5.2 |
| 6:00 | 10.4 | 4.8 | 6.2 | 6.7 | 5.5 | 6.5 | 6.4 | 6.4 | 5.1 | 7.7 | 6.4 | 6.3 | 6.4 | 4.7 | 6.3 | 4.6 | 6.0 | 6.3 | 4.7 | 5.5 | 4.7 | 5.5 | 5.5 | 4.7 |
| 6:10 | 10.5 | 5.5 | 5.3 | 6.7 | 5.5 | 6.5 | 6.4 | 6.4 | 5.1 | 7.7 | 6.4 | 6.3 | 6.4 | 4.7 | 6.3 | 4.6 | 6.0 | 6.3 | 5.5 | 5.5 | 4.7 | 5.5 | 5.5 | 4.7 |
| 6:20 | 10.5 | 6.0 | 5.8 | 7.2 | 6.0 | 7.0 | 6.9 | 6.9 | 5.7 | 7.1 | 6.0 | 6.0 | 6.0 | 5.2 | 6.8 | 6.0 | 6.5 | 6.8 | 6.0 | 6.0 | 5.2 | 6.0 | 6.0 | 5.2 |
| 6:30 | 10.4 | 6.3 | 7.0 | 8.2 | 6.1 | 7.0 | 7.9 | 7.1 | 6.7 | 8.1 | 7.0 | 7.0 | 7.0 | 6.3 | 7.9 | 6.3 | 6.9 | 7.0 | 6.2 | 6.2 | 6.3 | 7.1 | 7.0 | 6.2 |
| 6:40 | 10.5 | 6.1 | 6.6 | 7.5 | 6.6 | 7.5 | 7.5 | 7.6 | 6.3 | 8.6 | 6.6 | 6.7 | 6.6 | 6.0 | 7.6 | 5.9 | 6.5 | 7.5 | 6.7 | 6.7 | 6.0 | 6.7 | 7.5 | 6.7 |

$T_{iB,LE}$ has been corrected for View Factors over turf grass and PP

**Table A1.** *Cont.*

| | | | | | | | | | | | | | | | | | | | | | | | | |
|---|---|---|---|---|---|---|---|---|---|---|---|---|---|---|---|---|---|---|---|---|---|---|---|---|
| 24/08/2017, winter | | $T_{iB,LE}$ has been corrected for View Factors over turf grass and PP | | | | | | | | | | | | | | | | | | | | | | |
| $T_{iB,LE}$ (°C) | VF of LE | 0.71 | 0.58 | 0.43 | 0.56 | 0.52 | 0.58 | 0.58 | 0.56 | 0.45 | 0.54 | 0.59 | 0.54 | 0.65 | 0.62 | 0.56 | 0.56 | 0.63 | 0.61 | 0.61 | 0.62 | 0.59 | 0.95 | 0.61 |
| Time | Murdoch $T_{ambient}$ | A | B | C | D | E | F | G | H | I | J | K | L | M | N | O | P | Q | R | S | T | U | V | W |
| 6:50 | 10.4 | 7.3 | 8.0 | 8.0 | 7.1 | 7.0 | 8.0 | 8.1 | 6.8 | 8.0 | 7.1 | 7.2 | 7.1 | 6.5 | 8.1 | 6.4 | 7.0 | 8.0 | 7.2 | 7.2 | 7.3 | 7.2 | 8.0 | 7.2 |
| 7:00 | 10.7 | 7.5 | 7.3 | 8.7 | 7.5 | 7.5 | 7.5 | 7.5 | 7.2 | 8.6 | 7.5 | 7.5 | 7.5 | 6.7 | 8.3 | 6.6 | 8.1 | 8.3 | 7.5 | 7.5 | 7.5 | 7.5 | 7.5 | 6.7 |
| 7:10 | 10.9 | 7.3 | 7.0 | 8.0 | 8.0 | 8.0 | 8.0 | 8.0 | 6.8 | 9.1 | 7.1 | 7.2 | 7.1 | 7.2 | 8.0 | 7.1 | 7.7 | 8.0 | 7.2 | 8.0 | 7.2 | 7.2 | 8.0 | 7.2 |
| 7:20 | 11.0 | 7.8 | 8.3 | 8.5 | 9.4 | 8.5 | 8.5 | 9.4 | 8.2 | 9.6 | 7.6 | 8.5 | 8.5 | 7.7 | 8.5 | 7.6 | 8.2 | 8.5 | 7.7 | 8.5 | 7.7 | 8.5 | 8.5 | 9.3 |
| 7:30 | 11.3 | 10.0 | 9.0 | 10.0 | 10.0 | 9.0 | 10.0 | 10.9 | 9.7 | 11.1 | 9.1 | 8.3 | 9.1 | 8.5 | 9.2 | 9.1 | 9.7 | 10.0 | 9.2 | 10.0 | 9.2 | 9.2 | 10.0 | 10.8 |
| 7:40 | 11.4 | 8.7 | 9.9 | 10.3 | 10.6 | 10.5 | 11.5 | 11.7 | 11.3 | 11.5 | 8.7 | 8.1 | 8.7 | 9.3 | 10.1 | 10.1 | 11.6 | 9.1 | 9.0 | 9.9 | 10.1 | 9.9 | 11.5 | 11.5 |
| 7:50 | 11.7 | 9.7 | 10.6 | 11.3 | 11.6 | 11.5 | 12.5 | 14.2 | 12.3 | 12.5 | 10.6 | 10.8 | 11.6 | 10.2 | 11.7 | 10.7 | 12.3 | 10.9 | 10.9 | 12.5 | 10.9 | 10.8 | 12.5 | 13.3 |
| 8:00 | 12.0 | 11.6 | 11.0 | 11.8 | 12.1 | 12.0 | 13.0 | 14.6 | 11.9 | 13.0 | 11.1 | 11.3 | 12.1 | 10.6 | 12.1 | 11.0 | 11.7 | 12.2 | 12.2 | 13.8 | 10.5 | 10.4 | 13.0 | 13.8 |
| 8:10 | 12.5 | 13.8 | 12.4 | 13.3 | 13.6 | 13.5 | 14.5 | 16.1 | 14.3 | 14.5 | 12.6 | 12.8 | 13.6 | 12.1 | 12.7 | 12.3 | 13.9 | 13.7 | 14.5 | 16.1 | 11.9 | 11.8 | 14.5 | 16.1 |
| 8:20 | 13.0 | 14.3 | 12.7 | 13.8 | 14.1 | 14.0 | 15.0 | 17.3 | 14.8 | 16.1 | 13.1 | 13.3 | 15.0 | 12.5 | 13.9 | 13.5 | 15.2 | 14.2 | 15.8 | 17.5 | 13.1 | 12.2 | 15.0 | 17.5 |
| 8:30 | 13.2 | 15.8 | 14.3 | 15.3 | 14.7 | 15.5 | 16.5 | 19.7 | 16.3 | 17.6 | 14.6 | 14.8 | 16.5 | 14.0 | 14.6 | 15.0 | 16.7 | 15.7 | 17.3 | 19.0 | 14.6 | 13.7 | 16.5 | 19.0 |
| 8:40 | 13.7 | 15.1 | 14.4 | 14.2 | 14.7 | 16.5 | 16.5 | 19.8 | 16.3 | 17.6 | 14.6 | 14.8 | 16.5 | 13.3 | 15.5 | 15.2 | 16.9 | 15.7 | 16.5 | 18.1 | 14.7 | 13.8 | 16.5 | 19.8 |
| 8:50 | 14.4 | 14.1 | 14.3 | 14.3 | 14.6 | 15.5 | 15.5 | 17.9 | 15.3 | 16.6 | 14.6 | 14.7 | 15.5 | 13.1 | 14.5 | 14.2 | 15.9 | 14.7 | 15.5 | 17.1 | 13.7 | 13.7 | 15.5 | 18.0 |
| 9:00 | 14.4 | 14.8 | 14.3 | 14.3 | 14.6 | 15.5 | 15.5 | 17.9 | 15.3 | 16.6 | 14.6 | 14.7 | 15.5 | 13.1 | 14.5 | 14.2 | 15.9 | 14.7 | 15.5 | 17.1 | 13.7 | 13.7 | 15.5 | 18.0 |
| 9:10 | 14.7 | 17.8 | 16.1 | 17.3 | 16.7 | 18.5 | 18.5 | 21.6 | 19.3 | 19.6 | 16.6 | 17.7 | 18.5 | 16.0 | 16.5 | 17.7 | 20.3 | 17.7 | 19.3 | 21.8 | 16.5 | 15.7 | 18.5 | 22.6 |
| 9:20 | 15.1 | 19.3 | 17.6 | 18.8 | 19.1 | 20.0 | 20.9 | 24.0 | 21.7 | 21.1 | 18.1 | 19.2 | 20.9 | 17.5 | 18.8 | 19.2 | 22.7 | 19.2 | 21.6 | 23.3 | 18.0 | 17.2 | 20.0 | 26.6 |
| 9:30 | 15.0 | 19.8 | 18.9 | 19.3 | 20.5 | 20.5 | 21.4 | 24.4 | 22.2 | 21.6 | 18.6 | 19.7 | 22.4 | 17.9 | 19.2 | 19.5 | 23.1 | 20.5 | 22.1 | 24.6 | 19.2 | 17.6 | 20.5 | 27.9 |
| 9:40 | 16.7 | 20.8 | 19.9 | 20.3 | 21.5 | 21.5 | 22.4 | 25.4 | 23.2 | 22.6 | 19.6 | 20.7 | 23.4 | 18.9 | 20.2 | 21.4 | 24.1 | 21.5 | 23.1 | 25.6 | 19.4 | 18.6 | 21.5 | 28.1 |
| 9:50 | 17.1 | 20.3 | 19.4 | 19.8 | 21.0 | 22.0 | 21.9 | 24.9 | 22.7 | 22.1 | 19.1 | 20.2 | 22.9 | 19.2 | 20.5 | 20.9 | 24.5 | 21.8 | 22.6 | 25.1 | 19.7 | 19.0 | 21.0 | 27.6 |
| 10:00 | 17.7 | 21.3 | 20.4 | 20.8 | 22.0 | 23.0 | 22.9 | 25.9 | 24.6 | 23.1 | 21.1 | 21.2 | 24.8 | 20.2 | 21.5 | 21.9 | 26.4 | 22.8 | 23.6 | 26.1 | 20.7 | 20.0 | 22.0 | 29.4 |
| 10:10 | 17.6 | 21.1 | 20.3 | 20.2 | 21.6 | 22.5 | 23.4 | 25.7 | 23.3 | 22.5 | 20.6 | 21.7 | 24.4 | 20.0 | 21.4 | 21.9 | 25.5 | 21.7 | 23.3 | 25.0 | 20.6 | 19.7 | 22.5 | 29.1 |
| 10:20 | 17.9 | 21.8 | 20.1 | 21.3 | 21.6 | 22.5 | 23.4 | 26.5 | 23.3 | 23.6 | 20.6 | 21.7 | 24.4 | 20.0 | 21.3 | 22.6 | 26.2 | 22.5 | 24.1 | 25.8 | 21.3 | 19.7 | 22.5 | 29.9 |
| 10:30 | 18.3 | 23.3 | 21.5 | 21.7 | 24.0 | 24.0 | 24.9 | 28.7 | 25.7 | 25.1 | 23.1 | 23.2 | 26.8 | 20.7 | 22.7 | 23.9 | 27.5 | 24.8 | 25.6 | 28.1 | 22.7 | 21.1 | 24.0 | 32.2 |
| 10:40 | 18.5 | 24.0 | 22.2 | 22.8 | 24.0 | 25.0 | 25.7 | 29.5 | 25.7 | 26.2 | 23.1 | 24.0 | 27.7 | 21.4 | 23.4 | 23.8 | 27.3 | 24.8 | 27.3 | 28.9 | 23.4 | 21.0 | 24.0 | 32.2 |
| 10:50 | 18.4 | 25.0 | 23.4 | 23.8 | 25.0 | 26.0 | 25.9 | 30.6 | 27.7 | 27.2 | 23.1 | 24.2 | 26.9 | 22.4 | 23.7 | 25.8 | 29.4 | 25.8 | 27.5 | 29.1 | 23.7 | 21.3 | 25.0 | 33.2 |
| 11:00 | 19.1 | 24.8 | 23.9 | 24.3 | 25.5 | 25.5 | 26.4 | 31.1 | 27.3 | 27.7 | 23.6 | 24.7 | 28.3 | 22.9 | 24.2 | 25.4 | 29.9 | 26.3 | 28.0 | 29.6 | 24.2 | 21.8 | 25.5 | 34.5 |

**Table A1.** *Cont.*

| | 24/08/2017, winter | | | | | | $T_{iB,LE}$ has been corrected for View Factors over turf grass and PP | | | | | | | | | | | | | | | | | |
|---|---|---|---|---|---|---|---|---|---|---|---|---|---|---|---|---|---|---|---|---|---|---|---|---|
| $T_{iB,LE}$ (°C) | VF of LE | 0.71 | 0.58 | 0.43 | 0.56 | 0.52 | 0.58 | 0.58 | 0.56 | 0.45 | 0.54 | 0.59 | 0.54 | 0.65 | 0.62 | 0.56 | 0.56 | 0.63 | 0.61 | 0.61 | 0.62 | 0.59 | 0.95 | 0.61 |
| Time | Murdoch $T_{ambient}$ | A | B | C | D | E | F | G | H | I | J | K | L | M | N | O | P | Q | R | S | T | U | V | W |
| 11:10 | 19.5 | 26.2 | 24.6 | 25.5 | 26.4 | 27.4 | 27.2 | 31.9 | 29.1 | 27.7 | 24.6 | 25.5 | 29.2 | 23.7 | 24.9 | 27.0 | 30.6 | 27.1 | 28.8 | 30.4 | 24.9 | 22.5 | 25.5 | 35.3 |
| 11:20 | 19.5 | 26.7 | 25.1 | 24.8 | 26.9 | 27.9 | 27.7 | 32.4 | 28.7 | 28.2 | 24.1 | 26.0 | 29.7 | 23.4 | 24.6 | 27.5 | 31.1 | 26.8 | 29.3 | 30.9 | 25.4 | 23.0 | 26.0 | 35.0 |
| 11:30 | 19.9 | 27.0 | 26.1 | 24.7 | 27.9 | 28.0 | 27.9 | 33.4 | 29.7 | 29.2 | 25.1 | 27.0 | 29.8 | 24.4 | 25.6 | 27.6 | 31.3 | 27.8 | 29.5 | 31.9 | 26.4 | 23.2 | 27.0 | 36.8 |
| 11:40 | 20.0 | 26.0 | 25.0 | 26.0 | 26.0 | 27.9 | 27.7 | 33.2 | 29.6 | 28.2 | 26.0 | 26.0 | 29.7 | 24.1 | 25.4 | 27.3 | 31.0 | 27.6 | 29.3 | 31.7 | 25.4 | 23.0 | 26.0 | 35.0 |
| 11:50 | 20.3 | 28.2 | 27.3 | 27.5 | 29.3 | 29.4 | 30.1 | 34.7 | 31.1 | 30.8 | 25.6 | 28.3 | 31.2 | 26.4 | 26.9 | 29.7 | 34.3 | 29.1 | 31.6 | 33.2 | 27.7 | 24.5 | 27.5 | 38.2 |
| 12:00 | 20.6 | 28.0 | 27.8 | 26.8 | 28.9 | 29.0 | 29.7 | 34.3 | 30.7 | 30.2 | 26.1 | 28.0 | 31.7 | 26.1 | 27.4 | 28.5 | 33.9 | 28.8 | 31.3 | 33.7 | 28.2 | 24.1 | 28.0 | 37.8 |
| 12:10 | 20.4 | 27.0 | 27.0 | 27.0 | 27.0 | 28.9 | 28.7 | 34.2 | 29.7 | 29.2 | 26.1 | 27.0 | 30.7 | 25.2 | 26.4 | 28.5 | 32.2 | 27.8 | 30.3 | 31.9 | 25.6 | 23.2 | 27.0 | 36.0 |
| 12:20 | 20.4 | 27.5 | 27.5 | 26.3 | 27.5 | 29.4 | 29.2 | 33.9 | 30.2 | 29.7 | 26.6 | 27.5 | 31.2 | 25.7 | 26.9 | 29.0 | 32.7 | 28.3 | 30.0 | 32.4 | 26.9 | 23.7 | 27.5 | 36.5 |
| 12:30 | 20.9 | 28.2 | 27.3 | 26.3 | 28.4 | 29.4 | 30.1 | 34.7 | 31.1 | 30.8 | 26.6 | 27.5 | 31.2 | 25.6 | 26.9 | 28.8 | 32.5 | 29.1 | 31.6 | 33.2 | 26.9 | 24.5 | 27.5 | 37.3 |
| 12:40 | 20.6 | 29.2 | 28.2 | 28.5 | 29.4 | 30.4 | 31.1 | 35.6 | 32.1 | 31.8 | 27.6 | 29.3 | 33.1 | 26.6 | 27.8 | 30.5 | 34.2 | 30.9 | 32.6 | 35.1 | 28.6 | 25.4 | 28.5 | 38.3 |
| 12:50 | 20.8 | 27.5 | 27.3 | 27.5 | 27.5 | 29.4 | 30.1 | 33.8 | 31.1 | 29.7 | 27.5 | 27.5 | 31.2 | 26.4 | 26.0 | 28.8 | 34.3 | 28.3 | 30.8 | 33.2 | 26.9 | 24.5 | 27.5 | 36.5 |
| 13:00 | 21.2 | 27.2 | 27.2 | 26.5 | 28.3 | 29.4 | 29.1 | 34.5 | 30.1 | 29.8 | 27.4 | 28.2 | 31.1 | 25.4 | 26.7 | 27.8 | 31.5 | 27.3 | 29.8 | 32.2 | 25.0 | 23.5 | 26.5 | 36.3 |
| 13:10 | 21.8 | 27.0 | 26.1 | 25.8 | 27.0 | 28.9 | 29.6 | 34.2 | 29.7 | 29.2 | 27.0 | 27.0 | 29.8 | 25.2 | 26.4 | 28.5 | 32.2 | 27.8 | 29.5 | 31.9 | 25.6 | 23.2 | 27.0 | 35.2 |
| 13:20 | 21.4 | 27.7 | 26.8 | 27.0 | 27.9 | 29.9 | 29.6 | 34.2 | 29.7 | 30.3 | 27.0 | 27.8 | 30.7 | 25.1 | 26.4 | 28.3 | 32.0 | 28.6 | 30.3 | 32.7 | 26.4 | 24.8 | 27.0 | 36.0 |
| 13:30 | 21.6 | 28.5 | 27.6 | 27.3 | 28.5 | 30.4 | 31.1 | 34.9 | 31.2 | 30.7 | 27.6 | 28.5 | 32.2 | 26.7 | 27.9 | 30.0 | 33.7 | 29.3 | 31.8 | 33.4 | 27.9 | 25.5 | 28.5 | 37.5 |
| 13:40 | 21.4 | 26.8 | 27.6 | 27.5 | 27.5 | 29.4 | 30.1 | 33.1 | 31.1 | 29.7 | 27.5 | 27.5 | 31.2 | 26.5 | 27.0 | 29.2 | 33.8 | 28.3 | 30.0 | 31.6 | 26.2 | 24.6 | 27.5 | 35.7 |
| 13:50 | 21.8 | 27.0 | 27.0 | 27.0 | 27.9 | 29.9 | 29.6 | 33.4 | 30.6 | 30.3 | 27.9 | 27.8 | 30.7 | 25.2 | 26.4 | 28.5 | 32.2 | 27.8 | 30.3 | 31.9 | 26.4 | 24.0 | 27.0 | 35.2 |
| 14:00 | 21.3 | 27.0 | 27.0 | 27.0 | 27.9 | 29.9 | 29.6 | 33.4 | 29.7 | 29.2 | 27.9 | 27.8 | 30.7 | 25.2 | 27.3 | 28.5 | 31.3 | 27.8 | 30.3 | 31.9 | 26.4 | 24.0 | 27.0 | 35.2 |
| 14:10 | 21.7 | 27.0 | 26.1 | 27.0 | 27.0 | 28.9 | 29.6 | 33.4 | 29.7 | 29.2 | 27.9 | 27.8 | 31.6 | 25.9 | 27.3 | 28.5 | 32.2 | 28.6 | 30.3 | 31.9 | 26.4 | 24.0 | 27.0 | 35.2 |
| 14:20 | 21.7 | 26.5 | 25.8 | 26.5 | 26.5 | 28.4 | 29.1 | 32.1 | 28.3 | 28.7 | 26.5 | 27.3 | 30.2 | 24.7 | 26.8 | 28.2 | 30.9 | 27.3 | 29.0 | 30.6 | 26.0 | 24.5 | 26.5 | 34.7 |
| 14:30 | 22.5 | 26.5 | 26.5 | 26.5 | 27.4 | 29.4 | 29.1 | 32.9 | 30.1 | 29.8 | 27.4 | 27.3 | 30.2 | 25.4 | 26.8 | 28.0 | 31.7 | 28.1 | 29.8 | 31.4 | 25.9 | 24.4 | 26.5 | 33.9 |
| 14:40 | 22.3 | 27.0 | 27.1 | 27.0 | 27.9 | 29.9 | 29.6 | 32.6 | 30.6 | 30.3 | 27.9 | 27.8 | 30.7 | 26.0 | 26.5 | 28.7 | 32.4 | 27.8 | 30.3 | 31.1 | 26.5 | 25.0 | 27.0 | 34.4 |
| 14:50 | 22.4 | 26.5 | 25.8 | 26.5 | 27.4 | 28.4 | 29.1 | 32.1 | 28.3 | 29.8 | 26.5 | 27.3 | 30.2 | 24.7 | 26.8 | 27.3 | 30.0 | 27.3 | 29.0 | 30.6 | 26.0 | 24.5 | 26.5 | 33.9 |
| 15:00 | 22.1 | 26.5 | 26.6 | 27.7 | 26.5 | 28.4 | 29.1 | 32.1 | 29.2 | 28.7 | 27.4 | 27.3 | 30.2 | 25.5 | 26.0 | 28.2 | 30.9 | 28.1 | 29.0 | 30.6 | 26.0 | 24.5 | 26.5 | 33.1 |
| 15:10 | 22.1 | 26.5 | 26.8 | 26.5 | 26.5 | 28.4 | 28.2 | 31.3 | 29.2 | 28.7 | 27.4 | 26.5 | 30.2 | 25.5 | 26.1 | 27.5 | 31.1 | 27.3 | 29.0 | 29.8 | 25.3 | 24.5 | 26.5 | 32.2 |

**Table A1.** *Cont.*

| 24/08/2017, winter | | $T_{iB,LE}$ has been corrected for View Factors over turf grass and PP | | | | | | | | | | | | | | | | | | | | | | |
|---|---|---|---|---|---|---|---|---|---|---|---|---|---|---|---|---|---|---|---|---|---|---|---|---|
| $T_{iB,LE}$ (°C) | VF of LE | 0.71 | 0.58 | 0.43 | 0.56 | 0.52 | 0.58 | 0.58 | 0.56 | 0.45 | 0.54 | 0.59 | 0.54 | 0.65 | 0.62 | 0.56 | 0.56 | 0.63 | 0.61 | 0.61 | 0.62 | 0.59 | 0.95 | 0.61 |
| Time | Murdoch $T_{ambient}$ | A | B | C | D | E | F | G | H | I | J | K | L | M | N | O | P | Q | R | S | T | U | V | W |
| 15:20 | 22.3 | 25.8 | 26.0 | 26.5 | 25.6 | 27.5 | 28.2 | 30.6 | 28.3 | 28.7 | 26.5 | 26.5 | 29.3 | 24.8 | 26.2 | 27.7 | 30.4 | 27.3 | 28.1 | 29.0 | 25.4 | 23.7 | 26.5 | 32.2 |
| 15:30 | 22.1 | 25.5 | 24.9 | 25.5 | 24.6 | 27.4 | 27.2 | 29.5 | 27.3 | 27.7 | 26.4 | 25.5 | 28.3 | 24.5 | 25.1 | 26.5 | 29.2 | 26.3 | 27.1 | 28.8 | 25.1 | 23.5 | 25.5 | 29.6 |
| 15:40 | 22.1 | 25.5 | 25.8 | 26.7 | 25.5 | 27.4 | 28.1 | 29.5 | 27.3 | 27.7 | 26.4 | 26.3 | 29.2 | 24.5 | 25.9 | 26.5 | 29.2 | 26.3 | 28.0 | 28.8 | 25.1 | 23.5 | 25.5 | 31.2 |
| 15:50 | 22.4 | 25.0 | 25.4 | 26.2 | 25.0 | 26.9 | 27.6 | 28.2 | 26.8 | 27.2 | 25.9 | 25.8 | 28.7 | 24.1 | 25.5 | 26.2 | 28.0 | 25.8 | 26.6 | 27.5 | 24.7 | 23.9 | 25.0 | 29.9 |
| 16:00 | 22.0 | 25.0 | 24.5 | 25.0 | 25.0 | 26.9 | 27.6 | 28.2 | 26.8 | 27.2 | 25.9 | 25.8 | 27.8 | 23.3 | 25.5 | 25.3 | 27.1 | 25.8 | 26.6 | 27.5 | 24.7 | 23.1 | 25.0 | 29.1 |
| 16:10 | 21.7 | 23.3 | 24.4 | 25.2 | 24.0 | 25.9 | 25.7 | 27.2 | 25.7 | 26.2 | 25.9 | 24.8 | 26.8 | 23.1 | 24.5 | 24.3 | 27.0 | 24.8 | 25.6 | 26.5 | 23.7 | 22.9 | 24.0 | 28.1 |
| 16:20 | 21.8 | 23.5 | 23.9 | 24.7 | 23.5 | 25.4 | 26.1 | 26.7 | 25.2 | 25.7 | 24.4 | 24.3 | 26.3 | 22.6 | 24.0 | 23.8 | 25.6 | 24.3 | 25.1 | 26.0 | 23.2 | 22.4 | 23.5 | 26.8 |
| 16:30 | 21.5 | 22.8 | 21.4 | 23.5 | 19.9 | 25.4 | 25.2 | 22.5 | 25.2 | 23.5 | 24.4 | 22.7 | 26.3 | 22.6 | 22.5 | 24.0 | 24.9 | 24.3 | 24.3 | 25.1 | 23.3 | 22.5 | 23.5 | 21.0 |
| 16:40 | 21.4 | 21.9 | 21.2 | 26.3 | 20.5 | 23.4 | 23.1 | 21.3 | 22.2 | 21.6 | 23.3 | 21.3 | 23.3 | 22.8 | 21.2 | 23.9 | 25.6 | 22.9 | 21.3 | 21.3 | 22.8 | 21.3 | 20.5 | 20.5 |
| 16:50 | 21.1 | 19.3 | 20.8 | 23.5 | 20.0 | 21.9 | 21.7 | 20.9 | 20.8 | 21.1 | 21.9 | 20.8 | 21.9 | 20.8 | 20.8 | 20.9 | 21.7 | 20.8 | 19.2 | 20.0 | 20.8 | 20.0 | 20.0 | 20.0 |
| 17:00 | 21.0 | 18.8 | 19.4 | 21.8 | 18.6 | 20.5 | 21.2 | 20.4 | 19.4 | 20.6 | 21.4 | 19.5 | 20.4 | 19.5 | 20.3 | 19.5 | 20.3 | 19.5 | 18.7 | 19.5 | 19.5 | 18.7 | 19.5 | 18.7 |
| 17:10 | 20.8 | 18.8 | 19.6 | 20.7 | 18.6 | 20.5 | 21.2 | 20.4 | 19.4 | 20.6 | 20.4 | 19.5 | 19.5 | 18.8 | 19.6 | 18.8 | 19.6 | 19.5 | 17.9 | 18.7 | 18.8 | 18.7 | 19.5 | 18.7 |
| 17:20 | 20.5 | 18.3 | 19.8 | 20.2 | 19.0 | 20.0 | 20.7 | 19.9 | 18.9 | 20.1 | 19.9 | 19.8 | 19.9 | 18.2 | 19.8 | 18.1 | 19.8 | 19.8 | 19.0 | 19.0 | 19.0 | 18.2 | 19.0 | 19.0 |
| 17:30 | 20.0 | 17.8 | 18.6 | 19.7 | 18.5 | 19.5 | 20.2 | 19.4 | 18.4 | 19.6 | 19.4 | 18.5 | 19.4 | 17.8 | 19.4 | 17.8 | 19.5 | 18.5 | 17.7 | 17.7 | 18.6 | 17.7 | 18.5 | 17.7 |
| 17:40 | 19.7 | 16.8 | 18.3 | 19.8 | 17.5 | 18.5 | 19.2 | 18.4 | 17.4 | 18.6 | 18.4 | 17.5 | 18.4 | 17.5 | 18.3 | 17.5 | 18.3 | 18.3 | 16.7 | 17.5 | 17.5 | 17.5 | 17.5 | 16.7 |
| 17:50 | 19.4 | 16.8 | 18.4 | 18.7 | 17.5 | 18.5 | 19.2 | 18.4 | 17.4 | 18.6 | 18.4 | 17.5 | 18.4 | 17.6 | 18.4 | 16.8 | 18.5 | 17.5 | 16.7 | 16.7 | 17.6 | 16.7 | 17.5 | 16.7 |
| 18:00 | 19.5 | 16.3 | 17.1 | 18.2 | 16.1 | 18.0 | 18.7 | 17.1 | 15.9 | 18.1 | 17.0 | 17.0 | 17.0 | 16.3 | 17.9 | 16.3 | 17.0 | 17.0 | 16.2 | 16.2 | 16.3 | 16.2 | 17.0 | 15.4 |
| 18:10 | 19.0 | 15.3 | 16.0 | 17.2 | 15.1 | 17.0 | 17.7 | 16.1 | 15.8 | 17.1 | 16.9 | 16.0 | 16.9 | 16.1 | 16.9 | 15.3 | 16.0 | 16.0 | 15.2 | 15.2 | 16.1 | 16.1 | 16.0 | 14.4 |
| 18:20 | 18.4 | 15.3 | 16.0 | 17.2 | 15.1 | 17.0 | 17.7 | 16.1 | 15.8 | 18.2 | 16.9 | 16.0 | 16.0 | 15.3 | 16.9 | 15.3 | 16.0 | 16.0 | 14.4 | 15.2 | 15.3 | 15.2 | 16.0 | 14.4 |
| 18:30 | 18.8 | 14.8 | 15.5 | 17.8 | 15.5 | 16.5 | 17.2 | 15.6 | 15.3 | 17.7 | 16.4 | 15.5 | 16.4 | 15.6 | 16.4 | 15.7 | 16.4 | 15.5 | 14.7 | 14.7 | 15.6 | 15.6 | 15.5 | 14.7 |
| 18:40 | 18.3 | 14.8 | 15.5 | 16.7 | 14.6 | 16.5 | 17.2 | 15.6 | 15.3 | 16.6 | 15.5 | 15.5 | 15.5 | 14.8 | 15.6 | 14.8 | 15.5 | 15.5 | 14.7 | 14.7 | 15.6 | 15.6 | 15.5 | 13.9 |
| 18:50 | 18.1 | 14.3 | 15.9 | 16.2 | 15.0 | 16.0 | 16.7 | 15.1 | 14.8 | 17.2 | 15.9 | 15.0 | 15.9 | 14.3 | 15.9 | 14.3 | 15.0 | 15.0 | 14.2 | 14.2 | 15.1 | 15.1 | 15.0 | 14.2 |
| 19:00 | 17.9 | 14.3 | 15.9 | 16.2 | 15.0 | 16.0 | 16.7 | 15.9 | 14.8 | 17.2 | 15.9 | 15.0 | 15.9 | 15.1 | 15.9 | 14.3 | 15.0 | 15.0 | 14.2 | 14.2 | 15.1 | 15.1 | 15.0 | 14.2 |
| 19:10 | 17.4 | 14.3 | 15.9 | 16.2 | 15.0 | 15.0 | 15.9 | 15.1 | 14.8 | 16.1 | 15.0 | 15.0 | 15.0 | 14.3 | 15.9 | 14.3 | 15.0 | 15.0 | 14.2 | 14.2 | 14.3 | 15.1 | 15.0 | 14.2 |
| 19:20 | 17.3 | 13.8 | 14.5 | 15.7 | 13.6 | 14.5 | 16.2 | 14.6 | 14.3 | 16.7 | 15.4 | 14.5 | 14.5 | 14.6 | 15.4 | 13.8 | 15.4 | 14.5 | 13.7 | 13.7 | 14.6 | 14.6 | 14.5 | 13.7 |

**Table A1.** *Cont.*

| 24/08/2017, winter | | \multicolumn | | | | | $T_{iB,LE}$ has been corrected for View Factors over turf grass and PP | | | | | | | | | | | | | | | | |
|---|---|---|---|---|---|---|---|---|---|---|---|---|---|---|---|---|---|---|---|---|---|---|---|---|
| $T_{iB,LE}$ (°C) | VF of LE | 0.71 | 0.58 | 0.43 | 0.56 | 0.52 | 0.58 | 0.58 | 0.56 | 0.45 | 0.54 | 0.59 | 0.54 | 0.65 | 0.62 | 0.56 | 0.56 | 0.63 | 0.61 | 0.61 | 0.62 | 0.59 | 0.95 | 0.61 |
| Time | Murdoch $T_{ambient}$ | A | B | C | D | E | F | G | H | I | J | K | L | M | N | O | P | Q | R | S | T | U | V | W |
| 19:30 | 17.1 | 14.0 | 14.8 | 15.2 | 14.0 | 15.0 | 15.7 | 14.9 | 14.7 | 16.2 | 14.9 | 14.8 | 14.9 | 14.0 | 15.6 | 14.0 | 14.7 | 14.8 | 14.0 | 14.0 | 14.0 | 14.0 | 14.0 | 13.2 |
| 19:40 | 17.0 | 13.8 | 15.4 | 14.5 | 14.5 | 15.5 | 16.2 | 14.6 | 14.3 | 16.7 | 14.5 | 14.5 | 14.5 | 13.8 | 15.4 | 13.8 | 14.5 | 14.5 | 13.7 | 13.7 | 13.8 | 14.6 | 14.5 | 13.7 |
| 19:50 | 16.8 | 13.8 | 14.5 | 15.7 | 13.6 | 14.5 | 15.4 | 14.6 | 14.3 | 15.6 | 14.5 | 14.5 | 14.5 | 13.8 | 14.6 | 13.8 | 14.5 | 14.5 | 13.7 | 13.7 | 13.8 | 14.6 | 14.5 | 13.7 |
| 20:00 | 16.6 | 14.0 | 13.9 | 15.2 | 14.0 | 15.0 | 15.7 | 14.0 | 13.8 | 16.2 | 14.0 | 14.0 | 14.9 | 14.0 | 14.8 | 14.0 | 14.7 | 14.8 | 13.2 | 14.0 | 14.0 | 14.0 | 14.0 | 13.2 |
| 20:10 | 16.1 | 13.8 | 14.5 | 15.7 | 13.6 | 14.5 | 15.4 | 14.6 | 14.3 | 15.6 | 14.5 | 13.7 | 14.5 | 13.8 | 14.6 | 13.8 | 14.5 | 14.5 | 13.7 | 13.7 | 13.8 | 13.7 | 14.5 | 13.7 |
| 20:20 | 15.8 | 13.8 | 14.5 | 14.5 | 13.6 | 14.5 | 15.4 | 14.6 | 13.4 | 15.6 | 13.6 | 13.7 | 14.5 | 13.8 | 14.6 | 13.8 | 14.5 | 14.5 | 13.7 | 13.7 | 13.8 | 13.7 | 14.5 | 13.7 |
| 20:30 | 15.6 | 13.3 | 14.0 | 15.2 | 14.0 | 14.0 | 14.9 | 14.1 | 13.8 | 15.1 | 14.0 | 14.0 | 14.0 | 14.1 | 14.9 | 13.3 | 14.9 | 14.8 | 13.2 | 13.2 | 14.1 | 14.1 | 14.0 | 13.2 |

| 05/10/2017, spring | | | | | | | | $T_{iB,LE}$ has been corrected for View Factors over turf grass and PP | | | | | | | | | | | | | | | | |
|---|---|---|---|---|---|---|---|---|---|---|---|---|---|---|---|---|---|---|---|---|---|---|---|---|
| $T_{iB,LE}$ (°C) | VF of LE | 0.71 | 0.61 | 0.43 | 0.56 | 0.52 | 0.58 | 0.58 | 0.56 | 0.45 | 0.54 | 0.59 | 0.54 | 0.66 | 0.62 | 0.56 | 0.56 | 0.63 | 0.61 | 0.61 | 0.62 | 0.59 | 0.92 | 0.61 |
| Time | Murdoch $T_{ambient}$ | A | B | C | D | E | F | G | H | I | J | K | L | M | N | O | P | Q | R | S | T | U | V | W |
| 5:30 | 9.3 | 7.8 | 8.4 | 8.5 | 7.6 | 7.5 | 8.5 | 8.6 | 8.2 | 8.5 | 7.6 | 7.7 | 7.6 | 7.8 | 8.6 | 6.9 | 8.4 | 8.5 | 6.9 | 7.7 | 7.8 | 7.7 | 8.5 | 7.7 |
| 5:40 | 9.5 | 7.3 | 7.9 | 8.0 | 8.0 | 8.0 | 8.9 | 8.9 | 7.7 | 9.1 | 8.0 | 8.0 | 8.0 | 7.2 | 8.0 | 7.1 | 7.7 | 8.0 | 7.2 | 8.0 | 7.2 | 8.0 | 8.0 | 7.2 |
| 5:50 | 9.5 | 7.3 | 7.9 | 8.0 | 7.1 | 8.0 | 8.0 | 8.1 | 7.7 | 9.1 | 7.1 | 7.2 | 8.0 | 7.3 | 8.1 | 7.3 | 7.9 | 8.0 | 7.2 | 7.2 | 7.3 | 7.2 | 8.0 | 7.2 |
| 6:00 | 9.5 | 7.5 | 8.2 | 8.7 | 7.5 | 7.5 | 8.4 | 8.4 | 8.1 | 8.6 | 7.5 | 7.5 | 7.5 | 7.5 | 7.5 | 6.6 | 8.1 | 8.3 | 7.5 | 7.5 | 7.5 | 7.5 | 7.5 | 7.5 |
| 6:10 | 9.7 | 7.8 | 8.4 | 8.5 | 7.6 | 7.5 | 8.5 | 8.6 | 8.2 | 8.5 | 7.6 | 7.7 | 7.6 | 7.8 | 8.6 | 6.9 | 8.4 | 8.5 | 7.7 | 7.7 | 7.8 | 7.7 | 8.5 | 7.7 |
| 6:20 | 10.0 | 8.3 | 8.9 | 9.0 | 8.1 | 9.0 | 9.0 | 9.1 | 8.7 | 10.1 | 8.1 | 8.2 | 8.1 | 8.3 | 9.1 | 8.3 | 8.9 | 9.0 | 8.2 | 8.2 | 8.3 | 8.2 | 9.0 | 8.2 |
| 6:30 | 10.5 | 9.0 | 8.9 | 10.2 | 9.0 | 9.0 | 9.9 | 9.9 | 8.7 | 10.1 | 9.0 | 9.0 | 9.0 | 9.0 | 9.0 | 8.1 | 9.6 | 9.8 | 9.0 | 9.0 | 9.0 | 9.0 | 9.0 | 9.0 |
| 6:40 | 10.6 | 9.0 | 8.9 | 10.2 | 9.0 | 9.0 | 9.9 | 9.9 | 9.6 | 10.1 | 9.0 | 9.0 | 9.0 | 9.0 | 9.0 | 8.1 | 9.6 | 9.8 | 9.0 | 9.0 | 9.0 | 9.0 | 9.0 | 9.0 |
| 6:50 | 10.8 | 8.6 | 9.1 | 8.8 | 9.1 | 9.0 | 10.0 | 10.1 | 8.8 | 10.0 | 8.1 | 8.3 | 9.1 | 8.5 | 9.3 | 8.4 | 9.0 | 9.2 | 8.4 | 9.2 | 8.5 | 8.4 | 10.0 | 9.2 |
| 7:00 | 11.1 | 9.3 | 9.9 | 10.0 | 10.0 | 10.0 | 10.0 | 10.9 | 9.7 | 10.0 | 9.1 | 10.0 | 10.0 | 9.2 | 10.0 | 9.1 | 9.7 | 10.0 | 9.2 | 10.0 | 9.2 | 9.2 | 10.0 | 10.0 |
| 7:10 | 11.5 | 9.8 | 10.4 | 9.3 | 9.6 | 10.5 | 10.5 | 10.5 | 11.1 | 10.5 | 9.6 | 9.7 | 9.6 | 9.0 | 9.7 | 9.6 | 11.1 | 10.5 | 9.7 | 10.5 | 9.7 | 9.7 | 10.5 | 9.7 |
| 7:20 | 11.8 | 9.8 | 10.4 | 11.7 | 10.5 | 10.5 | 10.5 | 11.4 | 10.2 | 11.6 | 9.6 | 10.5 | 10.5 | 9.7 | 10.5 | 9.6 | 11.1 | 10.5 | 9.7 | 10.5 | 10.5 | 10.5 | 10.5 | 10.5 |
| 7:30 | 11.9 | 9.9 | 12.0 | 8.5 | 10.2 | 9.1 | 11.1 | 10.4 | 9.1 | 14.2 | 9.2 | 9.5 | 9.2 | 9.8 | 9.8 | 8.9 | 10.3 | 10.4 | 9.5 | 10.4 | 9.7 | 9.6 | 12.0 | 10.4 |
| 7:40 | 12.3 | 8.7 | 13.6 | 5.7 | 13.2 | 13.1 | 15.0 | 17.3 | 14.8 | 16.1 | 8.5 | 10.8 | 8.5 | 8.3 | 12.4 | 9.5 | 13.4 | 8.7 | 8.4 | 9.3 | 12.2 | 12.1 | 15.0 | 18.3 |
| 7:50 | 12.6 | 12.5 | 15.0 | 10.2 | 15.1 | 15.0 | 16.0 | 19.3 | 16.7 | 17.1 | 12.3 | 15.2 | 15.1 | 12.2 | 13.4 | 12.9 | 17.3 | 12.8 | 15.2 | 17.6 | 13.4 | 13.3 | 16.0 | 20.9 |

**Table A1.** *Cont.*

| 05/10/2017, spring | | $T_{iB,LE}$ has been corrected for View Factors over turf grass and PP | | | | | | | | | | | | | | | | | | | | | | |
| $T_{iB,LE}$ (°C) | VF of LE | 0.71 | 0.61 | 0.43 | 0.56 | 0.52 | 0.58 | 0.58 | 0.56 | 0.45 | 0.54 | 0.59 | 0.54 | 0.66 | 0.62 | 0.56 | 0.56 | 0.63 | 0.61 | 0.61 | 0.62 | 0.59 | 0.92 | 0.61 |
| Time | Murdoch $T_{ambient}$ | A | B | C | D | E | F | G | H | I | J | K | L | M | N | O | P | Q | R | S | T | U | V | W |
|---|---|---|---|---|---|---|---|---|---|---|---|---|---|---|---|---|---|---|---|---|---|---|---|---|---|
| 8:00 | 13.4 | 14.4 | 14.6 | 13.0 | 15.6 | 15.5 | 16.5 | 19.7 | 16.3 | 16.5 | 13.7 | 15.7 | 16.5 | 13.4 | 13.8 | 14.0 | 17.6 | 14.1 | 16.5 | 19.0 | 14.7 | 13.7 | 16.5 | 20.6 |
| 8:10 | 13.8 | 15.9 | 15.2 | 14.5 | 17.1 | 17.0 | 18.0 | 21.1 | 19.7 | 19.1 | 15.2 | 17.2 | 18.0 | 14.9 | 15.2 | 16.2 | 20.7 | 16.4 | 18.8 | 21.3 | 16.9 | 15.2 | 18.0 | 22.9 |
| 8:20 | 14.1 | 16.6 | 16.0 | 15.7 | 17.1 | 17.0 | 18.0 | 21.8 | 18.8 | 19.1 | 15.2 | 18.0 | 18.9 | 15.7 | 15.0 | 16.6 | 20.3 | 17.2 | 19.6 | 22.9 | 16.0 | 15.0 | 18.0 | 23.7 |
| 8:30 | 14.0 | 16.9 | 17.0 | 15.5 | 17.2 | 18.0 | 19.0 | 22.0 | 19.8 | 19.0 | 15.3 | 18.2 | 19.0 | 15.9 | 16.1 | 16.9 | 20.6 | 17.4 | 19.8 | 23.1 | 17.1 | 16.1 | 19.0 | 24.7 |
| 8:40 | 14.4 | 18.3 | 17.8 | 17.8 | 18.1 | 19.0 | 19.9 | 22.8 | 20.7 | 20.1 | 17.1 | 19.0 | 20.9 | 16.7 | 16.8 | 18.5 | 22.3 | 19.0 | 21.5 | 23.9 | 17.8 | 16.0 | 19.0 | 25.6 |
| 8:50 | 14.7 | 19.3 | 17.9 | 17.7 | 19.1 | 20.0 | 20.9 | 24.6 | 21.7 | 21.1 | 18.1 | 20.0 | 21.9 | 17.6 | 16.9 | 19.3 | 23.1 | 20.0 | 23.3 | 25.7 | 18.7 | 17.8 | 20.0 | 28.2 |
| 9:00 | 14.9 | 20.3 | 18.9 | 18.7 | 20.1 | 21.0 | 21.9 | 25.5 | 23.6 | 22.1 | 18.2 | 21.0 | 22.9 | 18.6 | 17.8 | 20.0 | 23.9 | 20.2 | 23.5 | 27.6 | 19.7 | 17.9 | 21.0 | 29.2 |
| 9:10 | 14.9 | 20.8 | 20.2 | 19.2 | 20.6 | 22.5 | 22.4 | 26.0 | 24.1 | 23.7 | 19.6 | 22.3 | 24.3 | 19.1 | 18.3 | 20.5 | 24.4 | 21.5 | 24.8 | 28.1 | 20.2 | 18.4 | 21.5 | 28.9 |
| 9:20 | 15.6 | 21.1 | 20.4 | 19.0 | 21.6 | 22.5 | 23.4 | 27.9 | 24.2 | 22.5 | 19.7 | 22.5 | 24.4 | 19.4 | 18.5 | 21.5 | 25.4 | 21.7 | 25.0 | 29.1 | 21.2 | 18.6 | 22.5 | 31.5 |
| 9:30 | 15.4 | 22.3 | 20.0 | 19.5 | 23.0 | 23.0 | 24.7 | 29.1 | 25.6 | 24.1 | 21.1 | 23.8 | 24.9 | 19.9 | 19.7 | 21.8 | 26.6 | 22.2 | 26.3 | 30.4 | 22.4 | 19.0 | 23.0 | 32.8 |
| 9:40 | 16.1 | 22.8 | 20.5 | 20.0 | 23.5 | 23.5 | 24.4 | 29.6 | 26.1 | 24.6 | 21.6 | 24.3 | 26.3 | 20.4 | 20.2 | 22.3 | 27.1 | 23.5 | 26.8 | 30.9 | 22.1 | 19.5 | 23.5 | 34.2 |
| 9:50 | 15.8 | 22.8 | 21.3 | 21.2 | 22.6 | 24.5 | 24.4 | 27.9 | 26.1 | 24.6 | 20.7 | 24.3 | 26.3 | 20.4 | 20.2 | 22.3 | 27.1 | 22.7 | 26.8 | 30.9 | 22.9 | 19.5 | 23.5 | 31.7 |
| 10:00 | 16.7 | 22.8 | 22.1 | 21.2 | 23.5 | 24.5 | 25.2 | 29.5 | 26.1 | 25.7 | 22.6 | 25.2 | 27.2 | 21.1 | 21.0 | 22.1 | 27.0 | 24.3 | 27.6 | 31.7 | 22.8 | 19.4 | 23.5 | 33.3 |
| 10:10 | 16.4 | 23.3 | 21.8 | 22.8 | 25.8 | 25.9 | 26.6 | 31.8 | 27.5 | 26.2 | 22.1 | 24.8 | 27.7 | 22.4 | 20.7 | 23.4 | 28.4 | 25.6 | 29.7 | 32.2 | 25.0 | 21.6 | 24.0 | 36.3 |
| 10:20 | 16.9 | 22.3 | 20.8 | 21.8 | 23.0 | 24.9 | 24.7 | 29.1 | 25.6 | 25.2 | 21.1 | 23.8 | 25.8 | 20.6 | 19.7 | 22.7 | 26.6 | 23.0 | 26.3 | 30.4 | 23.2 | 19.8 | 23.0 | 33.7 |
| 10:30 | 17.3 | 24.0 | 23.4 | 22.8 | 25.8 | 25.9 | 26.6 | 31.8 | 27.5 | 27.3 | 23.1 | 25.7 | 27.7 | 21.6 | 21.5 | 23.4 | 28.4 | 24.8 | 28.9 | 32.2 | 23.3 | 20.8 | 24.0 | 37.1 |
| 10:40 | 17.0 | 23.3 | 21.7 | 22.8 | 24.9 | 25.9 | 26.6 | 30.8 | 26.6 | 26.2 | 23.1 | 25.7 | 27.7 | 21.6 | 21.4 | 23.2 | 28.2 | 24.8 | 28.9 | 33.0 | 23.3 | 21.5 | 24.0 | 36.3 |
| 10:50 | 17.4 | 25.0 | 23.5 | 23.8 | 25.9 | 26.9 | 27.6 | 32.6 | 28.6 | 27.2 | 24.1 | 26.7 | 28.7 | 22.6 | 22.3 | 25.8 | 29.0 | 25.8 | 29.9 | 34.8 | 25.0 | 21.6 | 25.0 | 37.3 |
| 11:00 | 17.7 | 25.7 | 23.5 | 25.0 | 25.9 | 27.9 | 27.6 | 32.6 | 29.5 | 28.3 | 24.1 | 26.7 | 29.6 | 24.1 | 22.3 | 25.8 | 30.8 | 26.6 | 30.7 | 34.8 | 25.8 | 22.5 | 25.0 | 38.1 |
| 11:10 | 18.1 | 28.3 | 24.7 | 26.7 | 26.4 | 27.4 | 28.1 | 33.7 | 29.1 | 27.7 | 26.4 | 28.9 | 32.0 | 25.3 | 24.1 | 27.3 | 31.6 | 27.1 | 31.2 | 37.8 | 25.3 | 21.9 | 25.5 | 38.6 |
| 11:20 | 17.8 | 27.0 | 26.3 | 25.8 | 27.9 | 28.9 | 29.6 | 35.5 | 30.6 | 30.3 | 25.1 | 28.7 | 30.7 | 25.3 | 23.5 | 26.9 | 31.9 | 28.6 | 32.7 | 36.8 | 27.8 | 23.6 | 27.0 | 40.9 |
| 11:30 | 18.3 | 27.2 | 26.6 | 25.3 | 28.3 | 28.4 | 29.1 | 35.0 | 29.2 | 28.7 | 25.6 | 28.2 | 31.1 | 24.8 | 23.8 | 27.3 | 31.4 | 28.1 | 32.2 | 36.3 | 27.3 | 23.1 | 26.5 | 40.4 |
| 11:40 | 18.4 | 27.7 | 26.3 | 25.8 | 27.0 | 29.9 | 30.4 | 34.5 | 30.6 | 30.3 | 26.1 | 29.5 | 31.6 | 25.3 | 24.2 | 27.5 | 32.6 | 27.8 | 31.9 | 37.7 | 26.9 | 22.7 | 27.0 | 39.3 |
| 11:50 | 18.3 | 27.5 | 26.8 | 25.2 | 29.3 | 29.4 | 30.1 | 35.1 | 30.2 | 30.8 | 26.6 | 28.3 | 31.2 | 25.1 | 24.0 | 27.4 | 32.4 | 29.1 | 33.2 | 37.3 | 29.1 | 23.3 | 27.5 | 39.8 |
| 12:00 | 18.8 | 27.7 | 25.4 | 25.8 | 29.7 | 28.9 | 29.6 | 36.2 | 29.7 | 29.2 | 27.0 | 29.5 | 31.6 | 24.6 | 24.2 | 27.5 | 31.7 | 28.6 | 32.7 | 37.7 | 27.8 | 24.4 | 27.0 | 41.8 |
| 12:10 | 19.0 | 29.1 | 27.1 | 27.0 | 27.9 | 29.9 | 30.4 | 34.5 | 30.6 | 30.3 | 27.0 | 29.5 | 32.6 | 25.3 | 24.2 | 28.4 | 32.6 | 29.4 | 33.6 | 37.7 | 28.6 | 24.4 | 27.0 | 40.9 |

**Table A1.** *Cont.*

| 05/10/2017, spring | | $T_{iB,LE}$ has been corrected for View Factors over turf grass and PP | | | | | | | | | | | | | | | | | | | | | |
|---|---|---|---|---|---|---|---|---|---|---|---|---|---|---|---|---|---|---|---|---|---|---|---|
| $T_{iB,LE}$ (°C) | VF of LE | 0.71 | 0.61 | 0.43 | 0.56 | 0.52 | 0.58 | 0.58 | 0.56 | 0.45 | 0.54 | 0.59 | 0.54 | 0.66 | 0.62 | 0.56 | 0.56 | 0.63 | 0.61 | 0.61 | 0.62 | 0.59 | 0.92 | 0.61 |
| Time | Murdoch $T_{ambient}$ | A | B | C | D | E | F | G | H | I | J | K | L | M | N | O | P | Q | R | S | T | U | V | W |
| 12:20 | 19.7 | 27.5 | 26.1 | 26.3 | 28.4 | 29.4 | 30.1 | 35.2 | 29.3 | 29.7 | 26.6 | 28.3 | 31.2 | 25.1 | 24.1 | 27.6 | 31.7 | 28.3 | 31.6 | 36.5 | 28.4 | 23.3 | 27.5 | 39.8 |
| 12:30 | 19.8 | 28.2 | 26.8 | 27.5 | 29.3 | 31.3 | 30.9 | 35.0 | 31.1 | 30.8 | 27.5 | 30.0 | 33.1 | 26.6 | 25.5 | 28.0 | 32.2 | 28.3 | 31.6 | 38.2 | 29.1 | 24.1 | 27.5 | 39.0 |
| 12:40 | 19.4 | 29.2 | 28.6 | 27.3 | 29.4 | 32.3 | 31.9 | 36.9 | 33.0 | 32.9 | 28.5 | 31.0 | 34.1 | 26.8 | 25.7 | 29.9 | 34.1 | 29.3 | 33.4 | 39.2 | 30.1 | 25.1 | 28.5 | 40.8 |
| 12:50 | 19.7 | 28.0 | 27.3 | 26.8 | 28.0 | 30.9 | 31.4 | 35.5 | 30.7 | 30.2 | 28.0 | 30.5 | 32.6 | 26.3 | 25.2 | 29.4 | 32.7 | 28.8 | 32.9 | 38.7 | 27.9 | 25.4 | 28.0 | 40.3 |
| 13:00 | 19.7 | 26.5 | 29.2 | 26.5 | 27.4 | 30.3 | 29.1 | 33.4 | 30.1 | 29.8 | 26.5 | 28.2 | 31.1 | 24.9 | 24.8 | 26.8 | 30.9 | 27.3 | 30.6 | 34.7 | 27.5 | 24.1 | 26.5 | 38.0 |
| 13:10 | 20.5 | 29.5 | 30.5 | 27.2 | 29.5 | 33.3 | 32.9 | 35.4 | 34.0 | 33.9 | 28.6 | 31.2 | 35.1 | 27.8 | 26.8 | 30.3 | 35.3 | 28.7 | 32.0 | 39.3 | 30.3 | 24.4 | 29.5 | 41.0 |
| 13:20 | 20.4 | 28.0 | 26.5 | 28.0 | 28.9 | 30.9 | 31.4 | 34.7 | 31.6 | 30.2 | 28.9 | 31.4 | 33.6 | 26.3 | 26.0 | 28.5 | 32.7 | 28.0 | 32.1 | 38.7 | 28.8 | 25.4 | 28.0 | 40.3 |
| 13:30 | 20.5 | 31.1 | 27.5 | 30.2 | 31.7 | 31.9 | 33.3 | 38.2 | 34.4 | 33.4 | 29.0 | 32.4 | 35.5 | 28.8 | 27.8 | 31.3 | 34.6 | 33.0 | 35.6 | 39.7 | 32.2 | 27.2 | 29.0 | 42.1 |
| 13:40 | 20.6 | 29.2 | 28.7 | 28.5 | 29.4 | 31.4 | 31.9 | 35.3 | 32.1 | 30.7 | 28.5 | 30.2 | 33.1 | 27.6 | 25.9 | 29.5 | 32.7 | 29.3 | 32.6 | 37.5 | 29.4 | 26.0 | 28.5 | 39.2 |
| 13:50 | 20.5 | 27.4 | 29.1 | 26.0 | 29.5 | 32.4 | 32.9 | 35.7 | 32.2 | 32.8 | 27.6 | 29.5 | 32.3 | 26.4 | 26.3 | 29.4 | 32.5 | 28.7 | 32.0 | 36.1 | 29.8 | 25.6 | 29.5 | 38.5 |
| 14:00 | 21.0 | 30.2 | 28.1 | 29.5 | 30.4 | 32.4 | 32.9 | 35.5 | 33.1 | 32.8 | 29.5 | 32.0 | 34.1 | 28.6 | 26.9 | 30.5 | 33.7 | 30.3 | 33.6 | 38.5 | 31.2 | 26.2 | 29.5 | 40.2 |
| 14:10 | 20.7 | 28.5 | 29.6 | 27.3 | 29.4 | 31.4 | 31.9 | 34.5 | 31.2 | 30.7 | 28.5 | 30.2 | 32.2 | 26.9 | 26.0 | 29.7 | 32.0 | 28.5 | 31.8 | 36.7 | 28.6 | 25.3 | 28.5 | 38.3 |
| 14:20 | 21.1 | 29.0 | 27.6 | 27.8 | 30.8 | 31.9 | 32.4 | 35.9 | 32.6 | 32.3 | 29.9 | 31.5 | 33.6 | 27.4 | 27.3 | 29.3 | 32.5 | 29.8 | 32.3 | 37.2 | 30.0 | 26.6 | 29.0 | 38.8 |
| 14:30 | 21.5 | 28.5 | 27.1 | 28.5 | 29.4 | 31.4 | 31.9 | 33.7 | 32.1 | 30.7 | 29.4 | 31.0 | 33.1 | 26.9 | 26.8 | 29.7 | 32.0 | 29.3 | 31.8 | 36.7 | 28.6 | 26.1 | 28.5 | 37.5 |
| 14:40 | 21.2 | 27.8 | 27.3 | 27.3 | 28.5 | 31.4 | 31.9 | 33.9 | 31.2 | 29.6 | 28.5 | 30.2 | 31.3 | 26.1 | 26.2 | 28.6 | 30.8 | 28.5 | 31.0 | 34.2 | 28.8 | 25.5 | 28.5 | 35.9 |
| 14:50 | 21.6 | 27.5 | 28.7 | 28.7 | 28.4 | 31.3 | 30.9 | 32.9 | 31.1 | 30.8 | 28.4 | 29.2 | 32.1 | 26.7 | 26.0 | 28.5 | 30.7 | 27.5 | 30.0 | 33.2 | 28.7 | 25.3 | 27.5 | 34.9 |
| 15:00 | 21.5 | 27.5 | 27.9 | 27.5 | 28.4 | 30.4 | 30.9 | 32.1 | 30.2 | 30.8 | 28.4 | 29.2 | 31.2 | 25.9 | 26.0 | 28.5 | 30.7 | 27.5 | 29.1 | 33.2 | 28.7 | 25.3 | 27.5 | 34.9 |
| 15:10 | 21.3 | 27.0 | 27.4 | 28.2 | 27.9 | 29.9 | 30.4 | 31.6 | 29.7 | 29.2 | 27.9 | 28.7 | 30.7 | 26.2 | 25.5 | 28.0 | 30.2 | 27.0 | 29.5 | 32.7 | 27.3 | 24.8 | 27.0 | 34.4 |
| 15:20 | 21.5 | 28.2 | 26.2 | 27.5 | 28.4 | 30.4 | 30.9 | 32.8 | 30.2 | 29.7 | 28.4 | 30.0 | 31.2 | 26.6 | 25.9 | 29.0 | 30.3 | 28.3 | 30.0 | 34.9 | 27.7 | 26.0 | 27.5 | 34.9 |
| 15:30 | 21.4 | 27.0 | 26.6 | 28.2 | 27.0 | 29.9 | 29.6 | 31.6 | 28.8 | 28.1 | 27.0 | 28.7 | 29.8 | 26.2 | 26.4 | 28.0 | 29.3 | 27.0 | 28.6 | 32.7 | 26.5 | 24.8 | 27.0 | 33.6 |
| 15:40 | 21.1 | 26.5 | 26.2 | 27.7 | 26.5 | 28.4 | 29.1 | 30.4 | 28.3 | 27.6 | 26.5 | 27.3 | 29.3 | 25.7 | 25.2 | 28.0 | 29.1 | 27.3 | 28.1 | 30.6 | 27.0 | 24.5 | 26.5 | 31.4 |
| 15:50 | 21.4 | 25.5 | 25.1 | 26.7 | 25.5 | 28.4 | 28.9 | 28.4 | 27.3 | 26.6 | 26.4 | 27.2 | 28.3 | 24.7 | 24.9 | 26.8 | 27.9 | 26.3 | 27.1 | 30.4 | 25.9 | 23.4 | 25.5 | 30.4 |
| 16:00 | 21.0 | 25.3 | 26.5 | 26.0 | 26.0 | 28.9 | 28.6 | 29.0 | 27.8 | 28.2 | 26.9 | 26.8 | 29.7 | 24.4 | 25.5 | 25.7 | 27.7 | 26.0 | 26.8 | 30.1 | 26.5 | 24.0 | 26.0 | 30.1 |
| 16:10 | 21.3 | 24.3 | 24.8 | 25.0 | 24.1 | 26.9 | 27.6 | 27.3 | 25.9 | 26.1 | 25.9 | 25.8 | 26.9 | 23.4 | 24.7 | 25.2 | 26.2 | 24.2 | 25.0 | 27.5 | 24.8 | 23.1 | 25.0 | 27.5 |
| 16:20 | 21.2 | 22.8 | 23.3 | 24.7 | 23.5 | 26.4 | 26.1 | 25.8 | 25.2 | 24.6 | 25.4 | 25.2 | 26.3 | 22.7 | 23.2 | 24.6 | 25.6 | 24.3 | 24.3 | 26.0 | 24.1 | 22.4 | 23.5 | 26.0 |
| 16:30 | 20.9 | 22.8 | 23.3 | 24.7 | 22.6 | 25.4 | 26.1 | 25.1 | 24.3 | 24.6 | 24.4 | 24.3 | 25.4 | 22.7 | 23.3 | 23.9 | 24.9 | 23.5 | 23.5 | 25.1 | 24.2 | 21.7 | 23.5 | 25.1 |

**Table A1.** *Cont.*

| 05/10/2017, spring $T_{iB,LE}$ (°C) | VF of LE Murdoch $T_{ambient}$ | 0.71 | 0.61 | 0.43 | 0.56 | 0.52 | 0.58 | 0.58 | 0.56 | 0.45 | 0.54 | 0.59 | 0.54 | 0.66 | 0.62 | 0.56 | 0.56 | 0.63 | 0.61 | 0.61 | 0.62 | 0.59 | 0.92 | 0.61 |
|---|---|---|---|---|---|---|---|---|---|---|---|---|---|---|---|---|---|---|---|---|---|---|---|---|
| **Time** | | **A** | **B** | **C** | **D** | **E** | **F** | **G** | **H** | **I** | **J** | **K** | **L** | **M** | **N** | **O** | **P** | **Q** | **R** | **S** | **T** | **U** | **V** | **W** |
| 16:40 | 20.9 | 22.3 | 22.8 | 24.2 | 22.1 | 24.9 | 25.6 | 24.6 | 23.8 | 24.1 | 23.9 | 23.8 | 24.9 | 22.2 | 22.8 | 23.4 | 24.4 | 23.0 | 23.0 | 24.6 | 22.9 | 21.2 | 23.0 | 24.6 |
| 16:50 | 20.6 | 22.3 | 22.9 | 24.2 | 22.1 | 24.9 | 25.6 | 23.8 | 23.8 | 24.1 | 23.9 | 23.0 | 24.9 | 22.2 | 22.9 | 22.8 | 23.7 | 22.2 | 22.2 | 23.8 | 22.9 | 21.2 | 23.0 | 23.8 |
| 17:00 | 20.6 | 21.3 | 21.9 | 23.2 | 21.1 | 23.9 | 23.7 | 22.8 | 22.8 | 23.1 | 22.9 | 22.0 | 23.9 | 21.2 | 21.9 | 22.7 | 22.6 | 22.0 | 22.0 | 22.8 | 21.9 | 20.2 | 22.0 | 22.0 |
| 17:10 | 20.6 | 20.8 | 21.4 | 22.7 | 20.6 | 23.4 | 23.2 | 22.3 | 22.3 | 22.6 | 22.4 | 21.5 | 23.4 | 20.7 | 21.4 | 21.3 | 22.1 | 21.5 | 20.7 | 22.3 | 21.4 | 19.7 | 21.5 | 21.5 |
| 17:20 | 20.5 | 19.8 | 21.3 | 21.7 | 19.6 | 22.4 | 23.1 | 20.5 | 21.3 | 21.6 | 21.4 | 20.5 | 22.4 | 19.7 | 21.3 | 21.4 | 21.3 | 20.5 | 19.7 | 20.5 | 20.5 | 19.7 | 20.5 | 19.7 |
| 17:30 | 20.2 | 18.8 | 20.3 | 21.8 | 19.5 | 21.4 | 22.1 | 20.4 | 20.3 | 21.7 | 21.4 | 20.3 | 21.4 | 19.5 | 20.3 | 19.5 | 20.3 | 19.5 | 18.7 | 19.5 | 19.5 | 18.7 | 19.5 | 19.5 |
| 17:40 | 20.1 | 18.5 | 20.0 | 22.0 | 19.4 | 21.4 | 21.9 | 20.1 | 20.2 | 21.8 | 21.3 | 20.2 | 21.3 | 19.2 | 20.8 | 19.2 | 20.0 | 19.3 | 18.5 | 19.3 | 19.2 | 19.3 | 18.5 | 19.3 |
| 17:50 | 19.6 | 18.0 | 18.7 | 20.3 | 18.0 | 19.9 | 20.6 | 19.7 | 18.8 | 20.2 | 19.9 | 18.8 | 19.9 | 18.0 | 19.6 | 18.9 | 18.8 | 18.0 | 17.2 | 18.0 | 18.0 | 18.0 | 18.0 | 18.0 |
| 18:00 | 19.5 | 17.0 | 17.7 | 19.3 | 17.0 | 18.9 | 19.6 | 18.7 | 16.9 | 19.2 | 19.8 | 18.7 | 18.9 | 17.0 | 19.4 | 17.0 | 17.8 | 17.0 | 16.2 | 17.0 | 17.0 | 17.0 | 17.0 | 17.0 |
| 18:10 | 19.0 | 16.0 | 17.6 | 18.3 | 16.0 | 17.9 | 19.4 | 17.7 | 16.7 | 19.3 | 18.8 | 17.7 | 18.8 | 16.0 | 18.4 | 16.0 | 17.6 | 16.8 | 16.0 | 16.0 | 16.8 | 16.8 | 16.0 | 16.0 |
| 18:20 | 18.8 | 14.3 | 16.5 | 18.5 | 15.0 | 16.9 | 18.4 | 16.7 | 15.7 | 18.3 | 17.8 | 16.7 | 16.9 | 15.0 | 17.4 | 15.0 | 15.7 | 15.0 | 14.2 | 15.0 | 15.0 | 15.0 | 15.0 | 14.2 |
| 18:30 | 18.3 | 13.8 | 15.3 | 18.0 | 14.5 | 16.4 | 17.1 | 16.3 | 15.2 | 17.8 | 17.3 | 16.2 | 16.4 | 14.5 | 17.0 | 14.7 | 15.4 | 14.5 | 13.7 | 13.7 | 14.6 | 14.6 | 14.5 | 13.7 |
| 18:40 | 17.6 | 13.5 | 15.0 | 18.2 | 12.6 | 16.4 | 16.9 | 14.4 | 14.2 | 16.8 | 16.3 | 15.2 | 16.3 | 14.3 | 15.9 | 14.4 | 15.1 | 14.3 | 12.7 | 13.5 | 14.3 | 14.3 | 13.5 | 12.7 |
| 18:50 | 17.3 | 11.1 | 14.1 | 14.8 | 11.6 | 15.4 | 16.8 | 13.5 | 13.2 | 16.9 | 15.3 | 14.2 | 14.4 | 12.5 | 15.1 | 13.0 | 13.6 | 11.7 | 10.0 | 10.9 | 12.6 | 13.5 | 12.5 | 10.9 |
| 19:00 | 16.8 | 11.3 | 13.6 | 14.3 | 11.1 | 14.9 | 16.3 | 12.9 | 12.7 | 15.3 | 14.8 | 13.7 | 13.9 | 11.3 | 14.5 | 12.2 | 12.9 | 12.0 | 10.4 | 11.2 | 12.1 | 12.9 | 12.0 | 11.2 |
| 19:10 | 16.7 | 10.1 | 12.3 | 13.8 | 10.6 | 13.4 | 14.9 | 11.7 | 11.3 | 14.8 | 13.4 | 12.3 | 12.4 | 10.0 | 13.3 | 11.1 | 12.5 | 10.7 | 9.0 | 9.9 | 10.8 | 11.6 | 11.5 | 9.9 |
| 19:20 | 16.2 | 9.6 | 11.8 | 12.2 | 10.1 | 12.9 | 14.4 | 11.2 | 10.8 | 14.3 | 12.9 | 11.8 | 11.9 | 9.5 | 12.8 | 10.6 | 11.1 | 10.2 | 8.5 | 9.4 | 9.5 | 11.1 | 11.0 | 9.4 |
| 19:30 | 15.8 | 9.1 | 11.3 | 12.8 | 9.6 | 12.4 | 13.9 | 11.5 | 10.2 | 13.8 | 12.4 | 11.3 | 11.4 | 9.0 | 12.3 | 10.1 | 10.6 | 9.7 | 8.9 | 8.9 | 9.8 | 10.6 | 10.5 | 8.9 |
| 19:40 | 16.0 | 9.1 | 11.3 | 11.7 | 9.6 | 12.4 | 13.1 | 10.7 | 10.2 | 13.8 | 11.4 | 11.3 | 11.4 | 9.0 | 12.3 | 9.2 | 10.6 | 9.7 | 8.0 | 8.9 | 9.0 | 10.6 | 10.5 | 8.9 |
| 19:50 | 15.8 | 8.6 | 10.7 | 11.2 | 9.1 | 11.9 | 13.4 | 10.9 | 9.7 | 13.3 | 11.9 | 10.8 | 10.9 | 8.5 | 11.7 | 9.3 | 9.9 | 9.2 | 8.4 | 9.2 | 9.3 | 10.1 | 10.0 | 9.2 |
| 20:00 | 15.6 | 8.6 | 10.8 | 11.2 | 9.1 | 11.0 | 12.6 | 10.2 | 9.7 | 12.2 | 10.9 | 10.8 | 10.9 | 8.5 | 11.8 | 8.7 | 10.1 | 9.2 | 8.4 | 8.4 | 8.5 | 10.1 | 10.0 | 8.4 |
| 20:10 | 14.9 | 8.8 | 10.2 | 10.7 | 8.6 | 11.4 | 12.9 | 10.4 | 9.2 | 12.8 | 11.4 | 10.3 | 10.4 | 8.8 | 11.2 | 8.8 | 10.3 | 9.5 | 7.9 | 8.7 | 8.8 | 9.6 | 9.5 | 8.7 |
| 20:20 | 13.7 | 8.8 | 10.2 | 10.7 | 8.6 | 11.4 | 12.9 | 9.6 | 9.2 | 12.8 | 11.4 | 10.3 | 10.4 | 8.8 | 11.2 | 8.8 | 10.3 | 9.5 | 7.9 | 8.7 | 8.8 | 9.6 | 9.5 | 7.9 |
| 20:30 | 13.4 | 8.3 | 9.7 | 10.2 | 8.1 | 10.9 | 11.6 | 9.9 | 8.7 | 12.3 | 9.9 | 9.8 | 9.9 | 8.3 | 10.7 | 8.3 | 9.8 | 9.0 | 7.4 | 8.2 | 8.3 | 9.1 | 9.0 | 8.2 |

**Table A1.** *Cont.*

| 09/01/2018, summer | | | | | | $T_{iB,LE}$ has been corrected for View Factors over turf grass and PP | | | | | | | | | | | | | | | | | |
| $T_{iB,LE}$ (°C) | **VF of LE** | 0.71 | 0.61 | 0.43 | 0.56 | 0.52 | 0.58 | 0.58 | 0.56 | 0.45 | 0.54 | 0.59 | 0.54 | 0.66 | 0.62 | 0.56 | 0.56 | 0.63 | 0.61 | 0.61 | 0.62 | 0.59 | 0.98 | 0.61 |
| **Time** | **Murdoch** $T_{ambient}$ | **A** | **B** | **C** | **D** | **E** | **F** | **G** | **H** | **I** | **J** | **K** | **L** | **M** | **N** | **O** | **P** | **Q** | **R** | **S** | **T** | **U** | **V** | **W** |
| 5:00 | 18.6 | 16.3 | 17.0 | 17.0 | 17.0 | 17.0 | 17.9 | 17.1 | 16.9 | 18.1 | 17.0 | 17.0 | 17.0 | 16.3 | 17.1 | 16.3 | 17.9 | 17.8 | 16.2 | 16.2 | 17.1 | 17.0 | 17.0 | 16.2 |
| 5:10 | 18.6 | 16.3 | 17.0 | 17.0 | 16.1 | 17.0 | 17.0 | 17.1 | 16.9 | 18.1 | 17.0 | 16.2 | 17.0 | 16.3 | 17.1 | 16.3 | 17.0 | 17.0 | 16.2 | 16.2 | 16.3 | 16.2 | 17.0 | 16.2 |
| 5:20 | 18.1 | 16.3 | 17.0 | 17.0 | 16.1 | 17.0 | 17.0 | 17.1 | 16.9 | 17.0 | 17.0 | 16.2 | 16.1 | 16.3 | 17.1 | 16.3 | 17.0 | 17.0 | 16.2 | 16.2 | 16.3 | 16.2 | 17.0 | 16.2 |
| 5:30 | 18.1 | 16.3 | 17.0 | 17.0 | 16.1 | 17.0 | 17.0 | 17.1 | 16.9 | 17.0 | 17.0 | 16.2 | 16.1 | 16.3 | 17.1 | 16.3 | 17.0 | 17.0 | 16.2 | 16.2 | 16.3 | 16.2 | 17.0 | 16.2 |
| 5:40 | 18.2 | 16.3 | 17.0 | 17.0 | 16.1 | 17.0 | 17.0 | 17.1 | 16.9 | 17.0 | 17.0 | 16.2 | 16.1 | 16.3 | 17.1 | 16.3 | 17.0 | 17.0 | 16.2 | 16.2 | 16.3 | 16.2 | 17.0 | 16.2 |
| 5:50 | 18.2 | 16.1 | 16.6 | 16.3 | 16.6 | 16.5 | 17.5 | 17.6 | 16.6 | 17.5 | 16.6 | 16.7 | 16.6 | 16.0 | 16.8 | 15.9 | 17.6 | 17.5 | 16.7 | 16.7 | 16.8 | 16.7 | 17.5 | 16.7 |
| 6:00 | 18.3 | 16.8 | 17.4 | 17.5 | 16.6 | 17.5 | 18.4 | 17.5 | 17.4 | 17.5 | 17.5 | 16.7 | 17.5 | 16.7 | 17.5 | 16.6 | 18.3 | 17.5 | 16.7 | 17.5 | 16.7 | 16.7 | 17.5 | 16.7 |
| 6:10 | 18.3 | 16.8 | 17.4 | 17.5 | 16.6 | 17.5 | 18.4 | 18.4 | 17.4 | 18.6 | 17.5 | 16.7 | 17.5 | 16.7 | 17.5 | 16.6 | 18.3 | 17.5 | 16.7 | 17.5 | 16.7 | 16.7 | 17.5 | 17.5 |
| 6:20 | 18.5 | 17.1 | 17.7 | 16.2 | 17.6 | 17.5 | 18.5 | 18.6 | 17.6 | 18.5 | 16.6 | 16.8 | 16.6 | 17.0 | 17.8 | 16.9 | 17.7 | 17.7 | 16.9 | 17.7 | 17.0 | 16.8 | 18.5 | 19.3 |
| 6:30 | 19.0 | 18.1 | 19.4 | 18.3 | 19.5 | 19.5 | 20.4 | 21.1 | 19.4 | 20.6 | 18.6 | 18.7 | 19.5 | 18.0 | 18.6 | 18.4 | 20.1 | 18.7 | 18.7 | 20.3 | 18.6 | 17.8 | 19.5 | 20.3 |
| 6:40 | 19.2 | 19.8 | 19.6 | 19.3 | 19.6 | 19.5 | 20.5 | 22.1 | 20.4 | 20.5 | 19.6 | 19.7 | 20.5 | 19.0 | 19.6 | 19.4 | 21.1 | 19.7 | 20.5 | 21.3 | 18.8 | 18.8 | 20.5 | 21.3 |
| 6:50 | 19.4 | 20.3 | 20.0 | 19.8 | 21.0 | 21.0 | 21.0 | 22.6 | 21.7 | 22.1 | 20.1 | 20.2 | 21.0 | 19.5 | 20.0 | 19.6 | 22.3 | 21.0 | 21.8 | 22.6 | 19.3 | 19.3 | 21.0 | 22.6 |
| 7:00 | 19.5 | 21.5 | 21.3 | 20.3 | 21.5 | 21.5 | 22.4 | 23.8 | 22.2 | 22.6 | 20.6 | 21.5 | 22.4 | 19.9 | 20.4 | 20.8 | 22.7 | 21.5 | 22.3 | 24.0 | 20.5 | 19.8 | 21.5 | 24.0 |
| 7:10 | 20.0 | 22.3 | 22.0 | 20.7 | 22.1 | 23.0 | 23.0 | 25.3 | 23.7 | 24.1 | 21.1 | 22.2 | 23.9 | 21.4 | 21.1 | 21.4 | 24.2 | 23.0 | 23.8 | 25.5 | 21.2 | 20.5 | 23.0 | 25.5 |
| 7:20 | 20.2 | 22.8 | 22.5 | 21.2 | 22.6 | 23.5 | 23.5 | 25.8 | 24.2 | 24.6 | 21.6 | 22.7 | 24.4 | 21.9 | 21.6 | 21.9 | 25.6 | 23.5 | 25.1 | 26.0 | 21.7 | 21.0 | 23.5 | 26.8 |
| 7:30 | 20.7 | 24.0 | 22.9 | 22.8 | 24.0 | 24.0 | 24.9 | 27.1 | 24.7 | 25.1 | 23.1 | 24.0 | 25.9 | 22.4 | 22.0 | 23.1 | 25.9 | 24.0 | 25.6 | 27.3 | 22.1 | 21.5 | 24.0 | 28.1 |
| 7:40 | 21.0 | 25.0 | 23.9 | 23.8 | 25.0 | 25.0 | 25.9 | 29.0 | 26.5 | 27.2 | 24.1 | 25.0 | 26.9 | 23.4 | 23.0 | 24.1 | 27.8 | 25.0 | 27.5 | 28.3 | 23.1 | 22.5 | 25.0 | 29.9 |
| 7:50 | 21.2 | 25.5 | 24.4 | 24.3 | 25.5 | 25.5 | 26.4 | 29.4 | 27.0 | 26.6 | 24.6 | 25.5 | 27.4 | 23.9 | 23.4 | 24.3 | 28.1 | 25.5 | 28.0 | 29.6 | 23.6 | 23.0 | 25.5 | 30.4 |
| 8:00 | 21.7 | 26.5 | 25.4 | 25.3 | 25.6 | 27.5 | 27.4 | 30.4 | 28.1 | 27.6 | 25.6 | 26.5 | 28.4 | 24.9 | 24.4 | 25.3 | 29.1 | 26.5 | 28.1 | 30.6 | 24.6 | 24.0 | 26.5 | 31.4 |
| 8:10 | 22.6 | 28.0 | 26.9 | 25.7 | 28.0 | 28.0 | 28.9 | 31.9 | 29.6 | 29.1 | 26.1 | 28.0 | 29.9 | 26.4 | 25.1 | 26.8 | 31.6 | 28.0 | 30.5 | 32.1 | 26.1 | 25.5 | 28.0 | 34.6 |
| 8:20 | 23.1 | 28.5 | 26.5 | 26.2 | 29.4 | 29.5 | 30.2 | 33.2 | 30.1 | 30.7 | 27.6 | 29.3 | 31.3 | 26.9 | 26.3 | 27.1 | 31.9 | 28.5 | 31.8 | 33.4 | 27.3 | 26.0 | 28.5 | 36.7 |
| 8:30 | 23.0 | 29.7 | 28.6 | 27.8 | 29.0 | 30.9 | 30.7 | 34.4 | 32.1 | 31.2 | 28.1 | 30.7 | 32.7 | 28.2 | 27.5 | 29.1 | 34.0 | 29.8 | 33.1 | 34.7 | 27.7 | 27.3 | 29.0 | 35.6 |
| 8:40 | 23.4 | 30.2 | 28.3 | 28.3 | 29.5 | 31.4 | 31.2 | 34.1 | 31.9 | 31.7 | 28.6 | 30.3 | 33.2 | 27.9 | 27.2 | 28.8 | 34.5 | 30.3 | 33.6 | 35.2 | 28.2 | 27.0 | 29.5 | 37.7 |
| 8:50 | 24.2 | 31.2 | 29.2 | 29.3 | 31.4 | 32.4 | 32.2 | 36.7 | 32.9 | 32.7 | 29.6 | 32.2 | 34.2 | 28.9 | 28.1 | 30.4 | 34.4 | 31.3 | 34.6 | 37.1 | 29.2 | 28.0 | 30.5 | 39.5 |
| 9:00 | 24.5 | 31.7 | 29.8 | 29.8 | 31.9 | 32.9 | 32.7 | 36.4 | 34.2 | 33.2 | 30.1 | 31.8 | 34.7 | 29.4 | 28.7 | 31.1 | 36.0 | 31.0 | 34.3 | 36.7 | 29.7 | 28.5 | 31.0 | 40.0 |
| 9:10 | 25.1 | 33.4 | 30.7 | 30.8 | 33.8 | 33.9 | 34.6 | 38.1 | 35.2 | 35.3 | 31.1 | 33.7 | 36.6 | 31.1 | 29.6 | 31.7 | 37.5 | 33.6 | 36.9 | 39.4 | 31.4 | 29.5 | 32.0 | 42.7 |

**Table A1.** *Cont.*

| 09/01/2018, summer | | | | | | $T_{iB,LE}$ has been corrected for View Factors over turf grass and PP | | | | | | | | | | | | | | | | | | |
| $T_{iB,LE}$ (°C) | VF of LE | 0.71 | 0.61 | 0.43 | 0.56 | 0.52 | 0.58 | 0.58 | 0.56 | 0.45 | 0.54 | 0.59 | 0.54 | 0.66 | 0.62 | 0.56 | 0.56 | 0.63 | 0.61 | 0.61 | 0.62 | 0.59 | 0.98 | 0.61 |
| Time | Murdoch $T_{ambient}$ | A | B | C | D | E | F | G | H | I | J | K | L | M | N | O | P | Q | R | S | T | U | V | W |
| 9:20 | 25.1 | 33.2 | 30.4 | 30.2 | 33.4 | 34.4 | 34.2 | 40.4 | 34.9 | 35.8 | 31.6 | 33.3 | 36.2 | 30.1 | 30.1 | 31.5 | 36.4 | 32.5 | 36.6 | 39.1 | 31.2 | 30.0 | 32.5 | 42.3 |
| 9:30 | 25.6 | 34.9 | 32.2 | 32.3 | 34.4 | 35.4 | 35.2 | 40.5 | 36.7 | 35.7 | 32.6 | 35.2 | 38.1 | 31.9 | 31.1 | 33.2 | 38.2 | 34.3 | 37.6 | 40.9 | 32.1 | 31.0 | 33.5 | 43.3 |
| 9:40 | 26.4 | 34.2 | 33.0 | 31.2 | 35.3 | 35.4 | 35.2 | 41.4 | 36.7 | 36.8 | 32.6 | 36.0 | 38.1 | 31.9 | 31.1 | 33.2 | 39.1 | 34.3 | 38.4 | 40.9 | 32.1 | 31.0 | 33.5 | 45.0 |
| 9:50 | 26.6 | 36.4 | 33.7 | 32.7 | 36.8 | 36.9 | 36.7 | 42.0 | 38.2 | 38.3 | 34.1 | 37.5 | 39.6 | 33.4 | 31.7 | 34.7 | 40.6 | 35.8 | 39.9 | 42.4 | 33.6 | 32.5 | 35.0 | 48.1 |
| 10:00 | 27.8 | 36.4 | 34.5 | 33.8 | 37.7 | 37.9 | 37.6 | 43.6 | 39.8 | 38.3 | 35.0 | 37.5 | 40.6 | 33.4 | 33.3 | 35.3 | 41.3 | 35.8 | 40.7 | 43.2 | 34.3 | 33.3 | 35.0 | 48.9 |
| 10:10 | 27.9 | 38.1 | 36.2 | 36.0 | 39.6 | 38.9 | 38.6 | 45.4 | 40.8 | 40.4 | 36.0 | 39.4 | 41.6 | 35.1 | 34.2 | 37.0 | 43.0 | 36.8 | 41.7 | 45.0 | 36.1 | 34.3 | 36.0 | 49.9 |
| 10:20 | 28.5 | 38.1 | 36.2 | 36.0 | 38.7 | 38.9 | 38.6 | 45.4 | 40.8 | 40.4 | 36.9 | 39.4 | 41.6 | 35.1 | 34.2 | 37.0 | 43.0 | 37.6 | 41.7 | 45.0 | 36.1 | 34.3 | 36.0 | 50.8 |
| 10:30 | 28.5 | 38.8 | 34.6 | 37.2 | 36.9 | 39.8 | 39.4 | 45.4 | 40.0 | 39.3 | 36.9 | 39.4 | 41.6 | 35.9 | 34.2 | 37.9 | 42.1 | 37.6 | 41.7 | 45.0 | 36.1 | 34.3 | 36.0 | 47.5 |
| 10:40 | 28.5 | 38.6 | 35.9 | 36.5 | 39.2 | 40.3 | 39.9 | 45.9 | 40.5 | 40.9 | 37.4 | 40.7 | 42.1 | 35.6 | 34.7 | 37.5 | 43.5 | 38.1 | 43.1 | 45.5 | 36.6 | 34.8 | 36.5 | 50.4 |
| 10:50 | 29.9 | 40.1 | 39.0 | 36.8 | 42.5 | 41.8 | 41.4 | 49.9 | 42.0 | 42.4 | 38.9 | 42.2 | 43.6 | 37.1 | 36.9 | 38.8 | 44.8 | 38.8 | 44.6 | 47.8 | 37.2 | 36.3 | 38.0 | 52.8 |
| 11:00 | 29.9 | 40.6 | 37.9 | 37.3 | 41.2 | 42.3 | 41.1 | 49.6 | 42.5 | 42.9 | 39.4 | 42.7 | 44.1 | 36.8 | 36.6 | 39.3 | 44.4 | 39.3 | 44.2 | 48.3 | 38.5 | 36.8 | 38.5 | 52.4 |
| 11:10 | 29.9 | 41.3 | 38.6 | 37.3 | 42.1 | 42.3 | 41.9 | 49.5 | 44.1 | 42.9 | 39.4 | 43.6 | 45.0 | 37.6 | 37.3 | 39.9 | 46.0 | 40.1 | 45.1 | 49.2 | 39.3 | 36.8 | 38.5 | 53.3 |
| 11:20 | 30.9 | 43.5 | 40.1 | 38.8 | 45.4 | 43.8 | 43.4 | 51.7 | 44.8 | 45.6 | 41.9 | 45.1 | 47.4 | 39.1 | 38.7 | 41.2 | 47.3 | 42.4 | 47.4 | 51.5 | 41.5 | 39.2 | 40.0 | 57.2 |
| 11:30 | 30.8 | 43.3 | 40.6 | 39.3 | 45.0 | 44.3 | 44.8 | 52.2 | 46.1 | 46.1 | 42.4 | 45.6 | 47.9 | 39.6 | 39.2 | 42.6 | 47.8 | 42.1 | 47.1 | 52.0 | 42.8 | 39.7 | 40.5 | 56.9 |
| 11:40 | 31.5 | 42.3 | 41.3 | 39.5 | 45.8 | 43.3 | 43.8 | 51.3 | 45.1 | 45.1 | 41.4 | 45.4 | 46.9 | 39.3 | 39.1 | 41.8 | 47.9 | 41.9 | 46.1 | 50.2 | 41.1 | 38.7 | 39.5 | 56.7 |
| 11:50 | 31.2 | 41.6 | 39.6 | 39.5 | 44.0 | 43.3 | 44.7 | 51.3 | 42.7 | 45.1 | 41.4 | 45.4 | 46.9 | 38.6 | 38.3 | 40.9 | 45.2 | 40.3 | 45.2 | 50.2 | 41.1 | 38.7 | 39.5 | 55.9 |
| 12:00 | 32.4 | 42.1 | 40.9 | 40.0 | 44.5 | 45.8 | 44.3 | 52.6 | 44.8 | 45.6 | 42.8 | 45.9 | 47.4 | 39.1 | 39.5 | 41.2 | 46.4 | 41.6 | 46.6 | 51.5 | 40.7 | 39.2 | 40.0 | 55.6 |
| 12:10 | 31.9 | 44.3 | 42.4 | 41.5 | 46.9 | 46.3 | 45.8 | 54.1 | 46.3 | 47.1 | 43.4 | 46.6 | 48.9 | 40.6 | 41.0 | 43.6 | 49.7 | 43.1 | 48.9 | 53.0 | 43.0 | 40.7 | 41.5 | 58.7 |
| 12:20 | 32.9 | 44.1 | 43.7 | 42.0 | 47.4 | 45.8 | 45.4 | 54.6 | 46.8 | 47.6 | 42.9 | 47.1 | 49.4 | 41.1 | 41.5 | 43.2 | 49.3 | 43.6 | 48.6 | 53.5 | 42.7 | 40.3 | 42.0 | 58.4 |
| 12:30 | 33.7 | 43.2 | 43.5 | 40.2 | 46.1 | 46.3 | 45.1 | 55.3 | 45.7 | 46.9 | 42.5 | 46.7 | 47.1 | 39.3 | 41.4 | 42.4 | 47.5 | 41.7 | 47.4 | 52.3 | 41.7 | 40.0 | 42.5 | 57.3 |
| 12:40 | 32.7 | 43.6 | 43.3 | 41.5 | 46.9 | 46.3 | 45.8 | 54.3 | 47.1 | 47.1 | 43.4 | 47.4 | 48.9 | 40.6 | 41.2 | 43.2 | 50.1 | 43.1 | 48.1 | 51.3 | 42.3 | 40.7 | 41.5 | 58.7 |
| 12:50 | 33.4 | 43.6 | 43.3 | 41.5 | 46.9 | 45.3 | 45.8 | 55.1 | 46.3 | 48.2 | 43.4 | 46.6 | 48.9 | 40.6 | 42.0 | 43.2 | 49.2 | 43.1 | 48.1 | 51.3 | 41.5 | 40.7 | 41.5 | 57.9 |
| 13:00 | 34.2 | 42.9 | 44.2 | 40.3 | 46.9 | 46.3 | 45.8 | 54.3 | 47.1 | 47.1 | 44.3 | 47.4 | 48.9 | 39.8 | 42.0 | 42.3 | 49.2 | 41.5 | 47.2 | 51.3 | 41.5 | 40.7 | 41.5 | 57.9 |
| 13:10 | 34.3 | 42.9 | 42.5 | 40.3 | 46.0 | 46.3 | 45.8 | 53.4 | 46.3 | 47.1 | 43.4 | 47.4 | 48.9 | 39.8 | 41.2 | 42.3 | 48.3 | 42.3 | 47.2 | 51.3 | 41.5 | 40.7 | 41.5 | 57.9 |
| 13:20 | 35.0 | 43.2 | 43.6 | 41.3 | 46.1 | 46.3 | 45.9 | 54.6 | 46.5 | 48.1 | 43.4 | 46.7 | 48.1 | 40.1 | 42.4 | 42.8 | 47.9 | 42.5 | 46.6 | 50.7 | 41.8 | 40.8 | 42.5 | 56.4 |
| 13:30 | 35.0 | 42.9 | 43.3 | 41.5 | 44.2 | 46.3 | 45.8 | 52.5 | 47.1 | 45.9 | 44.3 | 47.4 | 49.8 | 39.8 | 42.7 | 42.9 | 49.0 | 42.3 | 47.2 | 52.2 | 41.4 | 41.5 | 41.5 | 54.6 |

**Table A1.** *Cont.*

| | 09/01/2018, summer | | $T_{iB,LE}$ has been corrected for View Factors over turf grass and PP | | | | | | | | | | | | | | | | | | | | | |
|---|---|---|---|---|---|---|---|---|---|---|---|---|---|---|---|---|---|---|---|---|---|---|---|---|
| $T_{iB,LE}$ (°C) | VF of LE | 0.71 | 0.61 | 0.43 | 0.56 | 0.52 | 0.58 | 0.58 | 0.56 | 0.45 | 0.54 | 0.59 | 0.54 | 0.66 | 0.62 | 0.56 | 0.56 | 0.63 | 0.61 | 0.61 | 0.62 | 0.59 | 0.98 | 0.61 |
| Time | Murdoch $T_{ambient}$ | A | B | C | D | E | F | G | H | I | J | K | L | M | N | O | P | Q | R | S | T | U | V | W |
| 13:40 | 33.3 | 39.0 | 37.5 | 39.0 | 39.9 | 42.8 | 42.4 | 48.2 | 41.4 | 43.4 | 39.9 | 43.2 | 45.5 | 36.6 | 39.4 | 37.7 | 42.9 | 38.2 | 43.1 | 49.7 | 36.5 | 37.3 | 39.0 | 49.7 |
| 13:50 | 33.0 | 38.3 | 39.2 | 37.8 | 39.9 | 42.8 | 42.4 | 48.3 | 42.2 | 43.4 | 39.9 | 42.4 | 45.5 | 36.6 | 39.5 | 37.1 | 43.1 | 37.4 | 42.3 | 48.8 | 36.6 | 37.3 | 39.0 | 49.7 |
| 14:00 | 33.7 | 39.7 | 37.4 | 40.2 | 40.8 | 42.8 | 42.4 | 49.0 | 41.4 | 43.4 | 41.8 | 44.1 | 46.4 | 38.1 | 40.1 | 39.3 | 43.6 | 39.0 | 43.9 | 50.5 | 37.3 | 38.2 | 39.0 | 50.5 |
| 14:10 | 32.9 | 39.9 | 38.5 | 39.7 | 40.3 | 42.3 | 41.9 | 47.6 | 42.5 | 42.9 | 40.4 | 42.7 | 46.8 | 37.6 | 39.5 | 38.6 | 43.8 | 38.5 | 44.2 | 50.8 | 36.7 | 37.7 | 38.5 | 50.0 |
| 14:20 | 31.5 | 38.2 | 36.9 | 38.7 | 38.4 | 41.3 | 40.9 | 46.8 | 40.7 | 40.8 | 38.4 | 40.9 | 44.0 | 35.8 | 38.0 | 37.4 | 43.4 | 37.5 | 41.6 | 47.3 | 35.9 | 36.7 | 37.5 | 47.3 |
| 14:30 | 31.5 | 35.6 | 35.6 | 34.7 | 37.0 | 38.9 | 39.6 | 43.8 | 38.6 | 40.3 | 37.0 | 39.5 | 42.6 | 33.8 | 36.8 | 34.4 | 40.4 | 34.6 | 39.5 | 46.0 | 33.9 | 34.5 | 37.0 | 46.0 |
| 14:40 | 30.5 | 35.8 | 35.0 | 36.5 | 37.4 | 39.4 | 39.9 | 44.1 | 38.9 | 39.8 | 37.4 | 39.9 | 43.0 | 34.1 | 36.2 | 35.5 | 40.6 | 35.7 | 39.8 | 46.3 | 34.1 | 34.8 | 36.5 | 45.5 |
| 14:50 | 29.8 | 38.0 | 35.8 | 36.8 | 38.0 | 41.8 | 41.4 | 46.6 | 40.4 | 40.2 | 38.9 | 41.4 | 43.6 | 34.8 | 37.9 | 36.6 | 41.6 | 37.2 | 41.3 | 46.2 | 35.7 | 36.3 | 38.0 | 47.0 |
| 15:00 | 30.5 | 36.0 | 34.6 | 36.0 | 36.9 | 38.9 | 39.4 | 43.7 | 38.4 | 39.3 | 36.9 | 39.4 | 42.5 | 33.6 | 36.6 | 35.2 | 39.4 | 35.2 | 39.3 | 45.0 | 33.7 | 35.2 | 36.0 | 45.0 |
| 15:10 | 30.8 | 37.0 | 36.5 | 37.0 | 37.0 | 39.9 | 40.4 | 44.8 | 41.0 | 39.2 | 37.9 | 40.4 | 42.6 | 34.6 | 36.9 | 36.4 | 41.5 | 37.0 | 41.1 | 45.2 | 35.5 | 36.2 | 37.0 | 45.2 |
| 15:20 | 30.8 | 35.5 | 35.0 | 36.7 | 36.4 | 38.4 | 38.9 | 43.3 | 37.9 | 38.8 | 37.4 | 38.9 | 42.0 | 33.9 | 36.2 | 34.9 | 39.1 | 35.5 | 38.8 | 43.7 | 34.0 | 34.7 | 35.5 | 44.5 |
| 15:30 | 30.8 | 36.5 | 36.0 | 36.5 | 36.5 | 38.4 | 39.1 | 43.5 | 38.9 | 39.8 | 37.4 | 39.0 | 42.1 | 34.1 | 36.5 | 35.3 | 40.3 | 35.7 | 39.8 | 43.9 | 34.3 | 34.8 | 36.5 | 44.7 |
| 15:40 | 30.7 | 35.3 | 33.9 | 34.8 | 36.0 | 38.9 | 38.6 | 43.1 | 36.8 | 38.2 | 36.9 | 38.5 | 40.6 | 33.6 | 36.1 | 35.0 | 37.3 | 34.4 | 38.5 | 42.6 | 33.9 | 34.3 | 36.0 | 43.4 |
| 15:50 | 29.8 | 33.8 | 33.3 | 34.5 | 33.6 | 36.4 | 37.1 | 40.8 | 35.3 | 36.7 | 35.4 | 37.0 | 39.1 | 32.1 | 34.7 | 33.8 | 36.9 | 33.7 | 37.0 | 40.2 | 32.4 | 32.8 | 34.5 | 40.2 |
| 16:00 | 29.0 | 34.3 | 34.6 | 33.8 | 34.1 | 37.9 | 37.6 | 41.3 | 38.2 | 37.2 | 35.9 | 37.5 | 38.7 | 32.6 | 35.2 | 33.4 | 38.3 | 33.4 | 36.6 | 40.7 | 32.9 | 33.3 | 35.0 | 40.7 |
| 16:10 | 29.0 | 32.8 | 33.1 | 34.7 | 32.6 | 36.4 | 36.1 | 39.8 | 35.9 | 35.7 | 35.4 | 36.0 | 38.1 | 31.9 | 33.7 | 32.8 | 36.7 | 32.7 | 36.0 | 39.2 | 32.2 | 32.7 | 33.5 | 39.2 |
| 16:20 | 28.5 | 32.6 | 32.9 | 32.8 | 33.1 | 35.9 | 36.6 | 40.5 | 35.6 | 36.2 | 34.9 | 35.7 | 36.8 | 30.9 | 33.5 | 32.8 | 36.7 | 32.4 | 35.6 | 38.1 | 32.1 | 32.3 | 34.0 | 38.9 |
| 16:30 | 28.8 | 31.3 | 30.8 | 32.0 | 31.1 | 33.9 | 34.6 | 36.7 | 33.6 | 34.2 | 32.9 | 33.7 | 35.7 | 29.7 | 32.3 | 30.6 | 34.5 | 31.2 | 33.6 | 36.9 | 30.0 | 30.3 | 32.0 | 36.9 |
| 16:40 | 27.7 | 31.3 | 30.9 | 32.0 | 31.1 | 33.9 | 34.6 | 36.7 | 33.6 | 34.2 | 32.9 | 33.7 | 35.7 | 29.7 | 32.3 | 31.7 | 34.7 | 31.2 | 33.6 | 36.1 | 30.9 | 31.2 | 32.0 | 36.9 |
| 16:50 | 27.3 | 31.3 | 30.1 | 32.0 | 31.1 | 33.9 | 34.6 | 36.7 | 32.8 | 34.2 | 32.9 | 33.7 | 34.8 | 29.7 | 31.5 | 30.8 | 33.8 | 31.2 | 33.6 | 36.1 | 30.1 | 30.3 | 32.0 | 36.1 |
| 17:00 | 27.2 | 30.3 | 29.9 | 31.0 | 30.1 | 32.9 | 32.7 | 34.9 | 32.6 | 33.2 | 31.9 | 32.7 | 33.8 | 29.4 | 31.3 | 30.7 | 33.7 | 30.2 | 32.6 | 35.1 | 29.9 | 30.2 | 31.0 | 34.3 |
| 17:10 | 26.7 | 29.8 | 28.6 | 31.7 | 28.7 | 32.4 | 32.2 | 33.5 | 31.3 | 31.6 | 31.4 | 32.2 | 33.3 | 28.9 | 30.0 | 30.2 | 32.3 | 30.5 | 32.1 | 34.6 | 29.4 | 29.7 | 30.5 | 33.0 |
| 17:20 | 26.6 | 29.3 | 29.8 | 30.0 | 29.1 | 31.9 | 31.7 | 34.1 | 30.8 | 32.2 | 30.9 | 30.8 | 31.9 | 27.7 | 30.5 | 29.3 | 31.3 | 29.2 | 30.8 | 32.5 | 28.2 | 28.3 | 30.0 | 33.3 |
| 17:30 | 26.8 | 29.3 | 28.9 | 31.2 | 29.1 | 31.0 | 31.7 | 32.2 | 30.8 | 31.1 | 30.9 | 30.8 | 32.8 | 28.4 | 29.6 | 29.1 | 32.0 | 30.0 | 31.6 | 33.3 | 28.9 | 29.2 | 30.0 | 32.5 |
| 17:40 | 26.3 | 28.8 | 28.5 | 30.7 | 28.6 | 31.4 | 31.2 | 32.7 | 30.3 | 30.6 | 30.4 | 30.3 | 32.3 | 27.9 | 30.0 | 28.8 | 30.8 | 29.5 | 30.3 | 32.0 | 28.5 | 28.7 | 29.5 | 32.0 |
| 17:50 | 26.4 | 28.3 | 28.8 | 30.2 | 28.1 | 30.0 | 30.7 | 31.3 | 29.8 | 30.1 | 29.9 | 29.8 | 30.9 | 27.4 | 28.7 | 28.3 | 30.3 | 29.0 | 29.8 | 31.5 | 28.0 | 28.2 | 29.0 | 30.6 |

**Table A1.** *Cont.*

| $T_{iB,LE}$ (°C) | VF of LE | 0.71 | 0.61 | 0.43 | 0.56 | 0.52 | 0.58 | 0.58 | 0.56 | 0.45 | 0.54 | 0.59 | 0.54 | 0.66 | 0.62 | 0.56 | 0.56 | 0.63 | 0.61 | 0.61 | 0.62 | 0.59 | 0.98 | 0.61 |
|---|---|---|---|---|---|---|---|---|---|---|---|---|---|---|---|---|---|---|---|---|---|---|---|---|
| Time | Murdoch $T_{ambient}$ | A | B | C | D | E | F | G | H | I | J | K | L | M | N | O | P | Q | R | S | T | U | V | W |
| 18:00 | 26.0 | 27.8 | 27.5 | 29.7 | 26.7 | 29.5 | 30.2 | 30.1 | 29.3 | 28.5 | 29.4 | 29.3 | 30.4 | 27.0 | 28.3 | 28.0 | 29.9 | 28.5 | 29.3 | 30.1 | 27.6 | 27.7 | 28.5 | 29.3 |
| 18:10 | 25.8 | 27.0 | 26.9 | 28.2 | 26.1 | 28.9 | 28.7 | 29.4 | 27.8 | 28.1 | 28.9 | 27.8 | 28.9 | 26.2 | 27.6 | 26.5 | 28.4 | 27.8 | 27.8 | 28.6 | 26.9 | 27.0 | 27.0 | 27.8 |
| 18:20 | 25.5 | 25.8 | 25.6 | 27.7 | 25.6 | 27.5 | 27.4 | 28.1 | 26.5 | 26.5 | 27.4 | 26.5 | 27.4 | 25.7 | 26.4 | 26.3 | 27.2 | 26.5 | 26.5 | 27.3 | 25.6 | 25.7 | 26.5 | 26.5 |
| 18:30 | 25.4 | 25.3 | 26.0 | 27.2 | 25.1 | 27.0 | 27.7 | 27.7 | 26.0 | 27.1 | 26.9 | 25.2 | 26.9 | 25.2 | 26.0 | 25.1 | 26.9 | 26.0 | 26.0 | 26.0 | 25.2 | 25.2 | 26.0 | 26.0 |
| 18:40 | 25.3 | 24.8 | 25.4 | 26.7 | 25.5 | 26.5 | 27.2 | 27.1 | 26.3 | 26.6 | 26.4 | 25.5 | 26.4 | 24.7 | 26.2 | 25.3 | 26.2 | 26.3 | 25.5 | 26.3 | 25.4 | 25.5 | 25.5 | 26.3 |
| 18:50 | 25.5 | 24.8 | 25.5 | 26.7 | 24.6 | 26.5 | 27.2 | 26.4 | 25.5 | 26.6 | 26.4 | 25.5 | 25.5 | 24.7 | 25.5 | 24.6 | 25.5 | 25.5 | 24.7 | 25.5 | 24.7 | 24.7 | 25.5 | 25.5 |
| 19:00 | 25.7 | 23.8 | 25.3 | 25.7 | 24.5 | 25.5 | 26.2 | 25.4 | 25.2 | 25.6 | 25.4 | 25.3 | 25.4 | 23.7 | 25.3 | 24.5 | 25.4 | 24.5 | 23.7 | 24.5 | 24.5 | 24.5 | 24.5 | 24.5 |
| 19:10 | 25.0 | 23.3 | 24.8 | 25.2 | 24.0 | 25.0 | 25.7 | 24.9 | 24.0 | 25.1 | 24.9 | 24.0 | 24.9 | 23.3 | 24.9 | 24.2 | 25.1 | 24.0 | 23.2 | 23.2 | 24.1 | 24.0 | 24.0 | 23.2 |
| 19:20 | 24.9 | 22.8 | 24.3 | 24.7 | 23.5 | 24.5 | 25.2 | 24.4 | 23.4 | 24.6 | 25.4 | 24.3 | 24.4 | 23.5 | 24.3 | 23.5 | 24.4 | 24.3 | 22.7 | 23.5 | 23.5 | 23.5 | 23.5 | 22.7 |
| 19:30 | 24.7 | 22.8 | 23.5 | 24.7 | 22.6 | 23.5 | 24.4 | 23.6 | 23.4 | 24.6 | 24.4 | 23.5 | 23.5 | 22.8 | 23.6 | 22.8 | 23.7 | 23.5 | 22.7 | 22.7 | 23.6 | 22.7 | 23.5 | 22.7 |
| 19:40 | 24.9 | 22.8 | 24.3 | 24.7 | 23.5 | 24.5 | 25.2 | 24.4 | 23.4 | 24.6 | 24.4 | 23.5 | 24.4 | 22.8 | 24.4 | 22.8 | 24.6 | 23.5 | 22.7 | 22.7 | 23.6 | 23.5 | 23.5 | 22.7 |
| 19:50 | 24.7 | 22.8 | 24.3 | 24.7 | 23.5 | 23.5 | 24.4 | 23.6 | 23.4 | 24.6 | 24.4 | 23.5 | 23.5 | 22.8 | 23.6 | 22.8 | 23.7 | 24.3 | 22.7 | 22.7 | 23.6 | 23.5 | 23.5 | 22.7 |
| 20:00 | 24.6 | 23.5 | 24.3 | 24.7 | 23.5 | 24.5 | 25.2 | 23.5 | 23.4 | 24.6 | 24.4 | 23.5 | 24.4 | 23.5 | 24.3 | 23.5 | 24.4 | 24.3 | 23.5 | 23.5 | 23.5 | 23.5 | 23.5 | 22.7 |
| 20:10 | 24.4 | 22.8 | 24.3 | 24.7 | 23.5 | 23.5 | 24.4 | 23.6 | 23.4 | 24.6 | 24.4 | 23.5 | 23.5 | 23.5 | 23.6 | 22.8 | 24.6 | 23.5 | 22.7 | 22.7 | 23.6 | 23.5 | 23.5 | 22.7 |
| 20:20 | 24.3 | 22.3 | 23.8 | 24.2 | 23.0 | 24.0 | 24.7 | 23.9 | 22.9 | 24.1 | 23.9 | 23.0 | 23.0 | 23.0 | 23.9 | 22.3 | 24.0 | 23.8 | 22.2 | 22.2 | 23.1 | 23.0 | 23.0 | 22.2 |
| 20:30 | 24.4 | 22.3 | 23.0 | 24.2 | 22.1 | 23.0 | 23.9 | 23.1 | 22.9 | 24.1 | 23.9 | 23.0 | 23.0 | 23.0 | 23.1 | 22.3 | 23.1 | 23.8 | 22.2 | 22.2 | 23.1 | 23.0 | 23.0 | 22.2 |
| 20:40 | 24.5 | 22.3 | 23.0 | 23.0 | 22.1 | 23.0 | 23.9 | 23.1 | 22.2 | 24.1 | 23.0 | 22.2 | 23.0 | 22.3 | 23.1 | 22.3 | 23.1 | 23.0 | 22.2 | 22.2 | 22.3 | 22.2 | 23.0 | 22.2 |
| 20:50 | 24.0 | 23.0 | 23.0 | 24.2 | 22.1 | 23.0 | 23.9 | 23.0 | 22.9 | 24.1 | 23.9 | 23.0 | 23.0 | 23.0 | 23.0 | 22.1 | 23.9 | 23.8 | 22.2 | 23.0 | 23.0 | 23.0 | 23.0 | 22.2 |
| 21:00 | 24.1 | 22.5 | 23.3 | 24.8 | 22.5 | 23.5 | 24.2 | 23.4 | 23.2 | 23.6 | 23.4 | 23.3 | 23.4 | 23.3 | 23.3 | 22.5 | 23.3 | 24.1 | 22.5 | 22.5 | 23.3 | 23.3 | 22.5 | 21.7 |
| 21:10 | 23.8 | 22.5 | 23.3 | 23.7 | 22.5 | 23.5 | 24.2 | 22.5 | 22.4 | 23.6 | 23.4 | 22.5 | 23.4 | 22.5 | 23.3 | 22.5 | 23.3 | 23.3 | 22.5 | 22.5 | 22.5 | 22.5 | 22.5 | 21.7 |
| 21:20 | 23.7 | 22.3 | 23.0 | 23.0 | 22.1 | 23.0 | 23.9 | 23.1 | 22.9 | 23.0 | 23.0 | 23.0 | 23.0 | 23.0 | 23.1 | 22.3 | 24.0 | 23.8 | 22.2 | 22.2 | 23.1 | 23.0 | 23.0 | 21.4 |
| 21:30 | 23.5 | 22.3 | 23.0 | 23.0 | 22.1 | 23.0 | 23.9 | 23.0 | 22.9 | 23.0 | 23.9 | 23.0 | 23.0 | 22.2 | 23.0 | 22.1 | 23.0 | 23.8 | 22.2 | 23.0 | 23.0 | 23.0 | 23.0 | 22.2 |

**Table A1.** *Cont.*

| 21/01/2019, summer | | \multicolumn | | | $T_{iB,LE}$ has been corrected for View Factor over turf grass | | | | | | | | | | | | |
|---|---|---|---|---|---|---|---|---|---|---|---|---|---|---|---|---|---|
| $T_{iB,LE}$ (°C) | VF of LE | 0.92 | 0.92 | 0.92 | 0.92 | 0.92 | 0.92 | 0.92 | 0.92 | 0.92 | 0.92 | 0.92 | 0.92 | 1.00 | 0.92 | 0.92 | 0.92 |
| Time | Murdoch $T_{ambient}$ | A | C | D | E | G | H | J | K | L | M | T | U | V | W | X | Y |
| 5:00 | 19.3 | 17.5 | 18.6 | 17.0 | 18.0 | 18.0 | 17.4 | 18.6 | 18.0 | 18.0 | 17.5 | 17.5 | 17.5 | 17.5 | 17.0 | 1:02 | 18.6 |
| 5:10 | 19.2 | 17.5 | 18.5 | 17.5 | 18.5 | 18.5 | 17.9 | 18.5 | 18.5 | 18.5 | 17.5 | 17.5 | 17.5 | 18.0 | 16.9 | 0:00 | 18.5 |
| 5:20 | 19.3 | 18.0 | 19.1 | 18.0 | 18.5 | 18.5 | 17.9 | 19.1 | 18.5 | 18.5 | 17.5 | 17.5 | 18.0 | 18.0 | 17.5 | 2:05 | 19.6 |
| 5:30 | 19.1 | 17.5 | 18.6 | 17.5 | 18.0 | 18.0 | 18.0 | 18.6 | 18.0 | 18.0 | 17.0 | 17.5 | 18.6 | 17.5 | 17.0 | 14:05 | 18.6 |
| 5:40 | 19.1 | 18.0 | 19.1 | 17.5 | 18.5 | 19.1 | 17.9 | 19.1 | 18.5 | 18.5 | 17.5 | 18.0 | 18.5 | 18.0 | 17.5 | 13:02 | 19.1 |
| 5:50 | 19.0 | 18.0 | 19.0 | 18.0 | 18.5 | 18.5 | 17.9 | 18.5 | 18.5 | 18.5 | 17.4 | 18.0 | 18.5 | 18.5 | 17.4 | 12:00 | 19.0 |
| 6:00 | 18.8 | 18.5 | 19.0 | 18.5 | 19.0 | 19.0 | 18.4 | 19.0 | 19.0 | 19.0 | 18.0 | 18.0 | 18.5 | 18.5 | 18.0 | 1:02 | 19.6 |
| 6:10 | 19.2 | 19.0 | 19.5 | 19.0 | 19.5 | 19.5 | 18.9 | 19.5 | 19.0 | 19.5 | 18.5 | 19.0 | 19.0 | 19.0 | 18.5 | 13:02 | 19.5 |
| 6:20 | 19.6 | 19.0 | 20.0 | 19.0 | 19.5 | 20.0 | 19.4 | 19.5 | 19.5 | 20.0 | 19.0 | 19.0 | 19.0 | 19.5 | 19.0 | 1:02 | 20.0 |
| 6:30 | 20.1 | 19.4 | 19.9 | 19.9 | 20.5 | 20.5 | 19.3 | 20.5 | 19.9 | 19.9 | 18.8 | 19.4 | 19.4 | 21.0 | 20.5 | 0:00 | 21.5 |
| 6:40 | 20.7 | 20.4 | 20.9 | 20.9 | 22.5 | 22.0 | 20.9 | 22.0 | 21.5 | 22.0 | 20.4 | 20.9 | 20.9 | 22.0 | 22.0 | 13:02 | 23.1 |
| 6:50 | 21.1 | 20.9 | 22.5 | 21.4 | 24.1 | 24.1 | 21.9 | 23.0 | 23.6 | 23.6 | 20.9 | 21.4 | 21.4 | 22.5 | 22.5 | 3:07 | 24.1 |
| 7:00 | 21.4 | 23.5 | 24.0 | 21.9 | 25.1 | 25.7 | 22.4 | 24.0 | 25.1 | 25.1 | 22.4 | 22.4 | 22.4 | 23.5 | 24.0 | 14:05 | 24.6 |
| 7:10 | 22.1 | 24.0 | 25.0 | 22.3 | 25.6 | 26.1 | 23.9 | 24.5 | 25.6 | 25.6 | 24.0 | 23.4 | 23.4 | 24.5 | 26.1 | 12:00 | 25.0 |
| 7:20 | 22.7 | 25.5 | 26.0 | 22.8 | 26.6 | 27.7 | 25.5 | 25.0 | 26.6 | 27.1 | 25.0 | 25.0 | 24.4 | 25.5 | 27.7 | 22:57 | 26.0 |
| 7:30 | 23.1 | 27.5 | 27.0 | 25.9 | 28.6 | 29.7 | 28.1 | 25.9 | 28.1 | 28.6 | 26.5 | 26.5 | 25.4 | 27.0 | 30.8 | 10:57 | 27.5 |
| 7:40 | 23.7 | 29.0 | 28.5 | 30.1 | 30.1 | 31.7 | 30.6 | 27.9 | 29.5 | 30.6 | 28.5 | 29.0 | 26.8 | 29.0 | 33.9 | 21:54 | 29.0 |
| 7:50 | 23.9 | 29.0 | 28.4 | 30.6 | 29.5 | 31.7 | 30.1 | 27.3 | 29.5 | 30.6 | 28.4 | 29.0 | 26.8 | 29.5 | 34.4 | 20:52 | 28.4 |
| 8:00 | 24.3 | 31.0 | 29.4 | 33.2 | 31.5 | 33.7 | 33.2 | 28.8 | 31.5 | 32.1 | 29.9 | 31.5 | 28.3 | 31.0 | 36.4 | 21:54 | 30.5 |
| 8:10 | 23.7 | 31.5 | 30.4 | 33.7 | 32.0 | 34.8 | 33.7 | 29.3 | 32.0 | 33.1 | 30.4 | 32.0 | 28.2 | 31.5 | 38.0 | 20:52 | 30.4 |
| 8:20 | 24.9 | 32.5 | 30.9 | 35.2 | 33.6 | 36.3 | 35.3 | 30.3 | 33.0 | 34.1 | 31.4 | 33.6 | 29.8 | 32.5 | 40.1 | 7:49 | 31.4 |
| 8:30 | 25.2 | 33.5 | 31.9 | 36.3 | 34.1 | 37.3 | 35.8 | 30.8 | 34.1 | 35.2 | 32.5 | 34.6 | 30.3 | 33.0 | 41.2 | 19:49 | 31.4 |
| 8:40 | 25.6 | 34.0 | 32.4 | 37.3 | 34.6 | 37.8 | 37.4 | 31.9 | 34.6 | 35.7 | 32.4 | 35.1 | 30.2 | 33.5 | 41.7 | 7:49 | 31.9 |
| 8:50 | 26.1 | 35.0 | 33.4 | 38.3 | 36.1 | 38.8 | 37.2 | 32.3 | 35.5 | 36.6 | 33.4 | 36.1 | 31.2 | 35.0 | 43.7 | 17:44 | 32.8 |
| 9:00 | 26.1 | 34.5 | 33.4 | 38.3 | 36.6 | 39.9 | 37.8 | 33.4 | 35.5 | 36.6 | 33.9 | 36.6 | 31.2 | 35.0 | 44.8 | 6:46 | 32.8 |
| 9:10 | 25.9 | 37.1 | 34.4 | 39.8 | 37.7 | 41.5 | 40.5 | 33.9 | 37.1 | 38.2 | 35.5 | 38.2 | 32.2 | 35.5 | 46.9 | 7:49 | 33.9 |

**Table A1.** *Cont.*

| 21/01/2019, summer | | $T_{iB,LE}$ has been corrected for View Factor over turf grass | | | | | | | | | | | | | | | |
| Time | $T_{iB,LE}$ (°C) VF of LE | 0.92 | 0.92 | 0.92 | 0.92 | 0.92 | 0.92 | 0.92 | 0.92 | 0.92 | 0.92 | 0.92 | 0.92 | 1.00 | 0.92 | 0.92 | 0.92 |
| Time | Murdoch $T_{ambient}$ | A | C | D | E | G | H | J | K | L | M | T | U | V | W | X | Y |
| 9:20 | 27.2 | 38.1 | 35.4 | 40.8 | 38.7 | 42.5 | 41.5 | 34.3 | 38.1 | 39.8 | 36.0 | 39.8 | 32.7 | 36.5 | 48.5 | 18:46 | 33.8 |
| 9:30 | 26.8 | 35.0 | 33.4 | 37.7 | 36.6 | 39.9 | 37.8 | 33.4 | 35.5 | 36.6 | 33.9 | 36.6 | 31.2 | 35.0 | 44.8 | 6:46 | 32.8 |
| 9:40 | 27.1 | 36.0 | 34.3 | 38.7 | 37.6 | 40.8 | 38.2 | 33.8 | 37.0 | 37.6 | 34.3 | 37.6 | 31.6 | 36.5 | 46.3 | 16:41 | 33.2 |
| 9:50 | 26.8 | 33.9 | 33.4 | 38.3 | 36.6 | 39.9 | 38.3 | 33.4 | 36.1 | 36.6 | 33.4 | 36.1 | 31.2 | 35.0 | 45.3 | 6:46 | 32.8 |
| 10:00 | 26.5 | 35.0 | 33.9 | 38.2 | 37.7 | 40.9 | 38.3 | 34.4 | 36.6 | 37.7 | 33.9 | 37.1 | 31.7 | 35.5 | 46.4 | 18:46 | 32.8 |
| 10:10 | 26.5 | 35.0 | 33.3 | 37.7 | 37.1 | 40.4 | 38.3 | 34.4 | 36.6 | 37.1 | 33.3 | 36.6 | 30.6 | 35.5 | 46.4 | 18:46 | 32.2 |
| 10:20 | 26.9 | 35.0 | 33.3 | 38.2 | 37.7 | 40.9 | 37.7 | 35.0 | 36.6 | 37.7 | 33.9 | 37.1 | 30.6 | 35.5 | 46.9 | 18:46 | 32.8 |
| 10:30 | 27.0 | 37.0 | 34.8 | 39.7 | 39.2 | 43.0 | 40.3 | 35.9 | 38.6 | 39.2 | 34.8 | 39.2 | 32.1 | 37.0 | 49.0 | 17:44 | 33.7 |
| 10:40 | 27.5 | 37.0 | 35.4 | 40.3 | 39.2 | 43.0 | 40.4 | 36.0 | 38.7 | 39.8 | 35.4 | 39.2 | 32.7 | 36.5 | 49.0 | 18:46 | 33.8 |
| 10:50 | 27.0 | 38.1 | 35.9 | 40.3 | 39.2 | 43.5 | 40.3 | 35.9 | 39.2 | 40.8 | 35.9 | 39.7 | 32.1 | 37.0 | 49.0 | 17:44 | 33.7 |
| 11:00 | 27.4 | 38.2 | 36.0 | 39.3 | 38.8 | 43.7 | 39.4 | 35.5 | 39.3 | 40.4 | 35.5 | 39.8 | 32.2 | 35.5 | 49.1 | 7:49 | 33.3 |
| 11:10 | 26.7 | 36.1 | 34.5 | 38.3 | 37.8 | 41.6 | 38.9 | 35.0 | 37.8 | 38.8 | 34.0 | 37.8 | 31.2 | 34.5 | 47.5 | 7:49 | 32.3 |
| 11:20 | 26.4 | 35.5 | 33.9 | 38.2 | 38.2 | 41.5 | 38.8 | 35.5 | 37.1 | 38.2 | 33.9 | 37.1 | 30.6 | 35.5 | 48.0 | 18:46 | 32.8 |
| 11:30 | 26.0 | 35.5 | 33.9 | 36.6 | 37.7 | 41.0 | 37.8 | 35.0 | 37.2 | 38.3 | 33.4 | 36.1 | 30.7 | 35.0 | 46.4 | 6:46 | 31.7 |
| 11:40 | 26.1 | 38.1 | 36.0 | 40.3 | 39.8 | 44.1 | 40.4 | 37.0 | 39.8 | 40.8 | 35.4 | 39.8 | 32.2 | 36.5 | 50.1 | 18:46 | 33.8 |
| 11:50 | 26.9 | 37.6 | 34.9 | 39.3 | 39.3 | 43.1 | 39.9 | 36.0 | 38.7 | 40.3 | 34.9 | 38.7 | 31.7 | 36.0 | 49.0 | 6:46 | 33.3 |
| 12:00 | 25.9 | 36.0 | 34.4 | 38.7 | 38.7 | 42.5 | 39.3 | 36.0 | 38.2 | 39.3 | 34.4 | 37.6 | 31.1 | 36.0 | 48.0 | 6:46 | 32.7 |
| 12:10 | 26.3 | 36.6 | 35.5 | 38.8 | 38.8 | 42.6 | 39.4 | 35.0 | 38.8 | 39.8 | 35.0 | 38.2 | 30.6 | 35.5 | 48.0 | 18:46 | 32.8 |
| 12:20 | 26.7 | 38.8 | 36.0 | 39.8 | 39.8 | 44.7 | 41.5 | 36.6 | 40.4 | 41.5 | 36.0 | 40.4 | 32.2 | 35.5 | 50.2 | 7:49 | 33.3 |
| 12:30 | 25.4 | 36.6 | 34.4 | 38.2 | 38.8 | 42.6 | 38.3 | 35.5 | 38.2 | 39.8 | 34.4 | 37.7 | 30.6 | 35.5 | 48.5 | 18:46 | 32.2 |
| 12:40 | 26.1 | 35.6 | 34.0 | 37.2 | 37.8 | 41.6 | 37.8 | 35.0 | 37.2 | 38.3 | 33.4 | 36.7 | 30.7 | 34.5 | 47.5 | 7:49 | 31.8 |
| 12:50 | 26.2 | 35.5 | 33.9 | 36.6 | 37.7 | 41.0 | 37.2 | 35.0 | 37.2 | 38.3 | 33.4 | 36.6 | 30.1 | 35.0 | 46.4 | 17:44 | 31.7 |
| 13:00 | 25.9 | 36.1 | 34.5 | 37.2 | 38.3 | 42.1 | 37.8 | 35.0 | 38.3 | 39.9 | 34.5 | 37.2 | 30.7 | 35.0 | 47.5 | 6:46 | 32.3 |
| 13:10 | 26.9 | 34.9 | 33.8 | 36.5 | 38.7 | 40.9 | 37.7 | 35.5 | 37.1 | 38.2 | 33.3 | 36.5 | 30.6 | 36.0 | 45.8 | 4:41 | 32.2 |
| 13:20 | 26.3 | 35.1 | 33.5 | 36.2 | 37.8 | 41.1 | 38.4 | 35.1 | 37.3 | 38.4 | 33.0 | 36.8 | 30.2 | 33.5 | 46.0 | 20:52 | 31.3 |
| 13:30 | 25.7 | 38.6 | 35.9 | 39.7 | 40.8 | 44.1 | 39.8 | 37.0 | 39.7 | 41.3 | 35.4 | 39.7 | 31.6 | 37.0 | 49.5 | 4:41 | 33.7 |

**Table A1.** *Cont.*

| 21/01/2019, summer | | | | | | | | | | | | | | | | | |
|---|---|---|---|---|---|---|---|---|---|---|---|---|---|---|---|---|---|
| $T_{iB,LE}$ (°C) | | \multicolumn | | | $T_{iB,LE}$ has been corrected for View Factor over turf grass | | | | | | | | | | | | |
| | VF of LE | 0.92 | 0.92 | 0.92 | 0.92 | 0.92 | 0.92 | 0.92 | 0.92 | 0.92 | 0.92 | 0.92 | 0.92 | 1.00 | 0.92 | 0.92 | 0.92 |
| Time | Murdoch $T_{ambient}$ | A | C | D | E | G | H | J | K | L | M | T | U | V | W | X | Y |
| 13:40 | 27.8 | 39.8 | 37.0 | 39.8 | 40.8 | 44.1 | 39.8 | 37.6 | 40.8 | 41.9 | 36.0 | 40.8 | 32.2 | 36.5 | 50.1 | 5:44 | 33.2 |
| 13:50 | 27.4 | 40.2 | 37.5 | 39.7 | 41.3 | 45.1 | 40.8 | 38.0 | 41.8 | 44.0 | 37.0 | 41.8 | 32.6 | 37.5 | 49.5 | 16:41 | 33.7 |
| 14:00 | 26.8 | 38.2 | 36.0 | 38.7 | 39.8 | 43.1 | 39.9 | 37.1 | 39.3 | 40.9 | 35.5 | 39.3 | 32.2 | 36.0 | 48.0 | 17:44 | 33.3 |
| 14:10 | 26.8 | 38.6 | 37.0 | 39.7 | 41.3 | 44.1 | 41.4 | 38.1 | 40.8 | 42.4 | 35.9 | 40.8 | 31.6 | 37.0 | 49.0 | 17:44 | 33.7 |
| 14:20 | 26.9 | 38.2 | 36.5 | 39.3 | 39.8 | 44.2 | 40.4 | 37.1 | 39.8 | 41.4 | 35.5 | 39.3 | 31.7 | 36.0 | 48.0 | 6:46 | 33.8 |
| 14:30 | 26.9 | 39.8 | 37.6 | 39.8 | 40.8 | 44.1 | 40.9 | 37.6 | 40.3 | 42.5 | 36.0 | 39.8 | 32.2 | 36.5 | 48.5 | 18:46 | 33.8 |
| 14:40 | 27.3 | 36.0 | 35.0 | 37.7 | 38.8 | 42.0 | 38.8 | 36.0 | 38.2 | 39.3 | 34.4 | 37.7 | 31.2 | 35.5 | 46.4 | 18:46 | 32.8 |
| 14:50 | 27.3 | 36.0 | 35.0 | 37.1 | 38.8 | 41.5 | 37.7 | 36.0 | 38.2 | 39.3 | 33.9 | 37.1 | 30.6 | 35.5 | 45.8 | 5:44 | 32.8 |
| 15:00 | 27.5 | 39.3 | 37.1 | 38.7 | 40.3 | 43.6 | 39.9 | 37.6 | 40.3 | 42.0 | 36.0 | 39.8 | 32.2 | 36.0 | 47.4 | 6:46 | 33.8 |
| 15:10 | 26.5 | 38.6 | 37.0 | 38.6 | 40.8 | 43.0 | 39.8 | 37.5 | 40.3 | 41.9 | 35.9 | 40.3 | 32.1 | 37.0 | 46.8 | 4:41 | 33.7 |
| 15:20 | 26.6 | 38.3 | 36.1 | 36.1 | 39.3 | 42.1 | 38.3 | 36.1 | 39.3 | 41.0 | 35.0 | 38.8 | 31.7 | 35.0 | 45.3 | 6:46 | 32.8 |
| 15:30 | 26.2 | 38.3 | 36.2 | 35.6 | 38.9 | 41.1 | 36.2 | 35.6 | 38.9 | 40.5 | 34.5 | 37.8 | 31.3 | 34.0 | 43.8 | 6:46 | 31.8 |
| 15:40 | 26.0 | 35.6 | 34.5 | 35.0 | 37.8 | 39.9 | 36.2 | 35.0 | 37.8 | 38.8 | 33.4 | 36.7 | 30.7 | 34.5 | 42.7 | 5:44 | 31.8 |
| 15:50 | 26.5 | 36.2 | 34.6 | 34.0 | 37.3 | 40.0 | 36.3 | 34.6 | 37.8 | 39.5 | 33.5 | 37.3 | 30.8 | 33.5 | 42.2 | 18:46 | 31.3 |
| 16:00 | 25.9 | 34.5 | 34.0 | 36.1 | 38.3 | 39.9 | 36.7 | 35.6 | 37.2 | 38.3 | 32.3 | 36.7 | 30.7 | 34.5 | 42.7 | 18:46 | 32.3 |
| 16:10 | 26.0 | 34.6 | 33.5 | 34.6 | 37.3 | 38.9 | 35.7 | 34.6 | 36.8 | 37.8 | 32.4 | 35.7 | 30.2 | 33.5 | 41.1 | 18:46 | 31.3 |
| 16:20 | 26.2 | 33.0 | 32.5 | 33.0 | 35.8 | 37.4 | 34.2 | 33.0 | 34.7 | 35.8 | 30.9 | 34.1 | 29.2 | 32.5 | 39.6 | 18:46 | 30.3 |
| 16:30 | 25.8 | 33.0 | 32.5 | 33.0 | 35.8 | 37.4 | 34.7 | 33.6 | 35.2 | 36.3 | 30.9 | 34.1 | 29.2 | 32.5 | 39.0 | 18:46 | 30.3 |
| 16:40 | 25.1 | 32.1 | 31.6 | 31.6 | 34.3 | 35.9 | 33.8 | 32.7 | 33.8 | 34.3 | 30.0 | 32.7 | 28.9 | 30.5 | 37.0 | 20:52 | 29.4 |
| 16:50 | 25.1 | 30.5 | 30.0 | 30.0 | 33.2 | 33.8 | 31.6 | 31.6 | 32.1 | 32.7 | 28.3 | 31.0 | 27.2 | 30.5 | 35.4 | 7:49 | 28.9 |
| 17:00 | 24.6 | 31.6 | 31.0 | 30.0 | 33.2 | 34.3 | 31.6 | 31.0 | 32.7 | 33.8 | 29.4 | 31.6 | 27.8 | 30.5 | 35.4 | 7:49 | 28.9 |
| 17:10 | 24.6 | 31.0 | 31.0 | 30.5 | 33.2 | 34.3 | 32.2 | 32.1 | 32.7 | 33.2 | 29.4 | 31.6 | 28.3 | 30.5 | 35.4 | 20:52 | 29.4 |
| 17:20 | 25.4 | 30.5 | 30.5 | 29.4 | 32.7 | 33.8 | 31.1 | 31.0 | 32.7 | 33.2 | 28.9 | 31.0 | 27.8 | 30.5 | 34.3 | 7:49 | 28.9 |
| 17:30 | 24.7 | 30.6 | 30.6 | 29.5 | 32.2 | 33.3 | 30.6 | 31.1 | 32.2 | 32.8 | 28.4 | 30.6 | 27.3 | 29.5 | 33.3 | 20:52 | 28.4 |
| 17:40 | 24.4 | 29.0 | 29.6 | 28.5 | 31.2 | 32.3 | 29.6 | 29.6 | 31.2 | 31.8 | 28.0 | 29.6 | 26.9 | 28.5 | 31.8 | 20:52 | 28.0 |
| 17:50 | 24.5 | 28.6 | 29.1 | 27.5 | 30.2 | 30.8 | 28.6 | 29.1 | 30.2 | 30.8 | 27.0 | 28.6 | 26.4 | 27.5 | 30.2 | 9:54 | 27.0 |

**Table A1.** *Cont.*

| 21/01/2019, summer | | $T_{iB,LE}$ has been corrected for View Factor over turf grass | | | | | | | | | | | | | | | |
| $T_{iB,LE}$ (°C) | VF of LE | 0.92 | 0.92 | 0.92 | 0.92 | 0.92 | 0.92 | 0.92 | 0.92 | 0.92 | 0.92 | 0.92 | 0.92 | 1.00 | 0.92 | 0.92 | 0.92 |
| Time | Murdoch $T_{ambient}$ | A | C | D | E | G | H | J | K | L | M | T | U | V | W | X | Y |
| 18:00 | 24.4 | 28.0 | 28.6 | 27.0 | 29.7 | 30.2 | 28.0 | 28.6 | 29.7 | 30.2 | 26.4 | 28.0 | 25.9 | 27.5 | 29.1 | 9:54 | 27.0 |
| 18:10 | 23.9 | 27.5 | 28.1 | 26.5 | 29.2 | 29.7 | 28.1 | 28.1 | 29.2 | 29.7 | 25.9 | 27.5 | 25.9 | 27.0 | 28.1 | 21:54 | 26.5 |
| 18:20 | 24.0 | 27.1 | 27.7 | 25.5 | 28.2 | 28.8 | 26.6 | 27.1 | 28.2 | 28.8 | 25.5 | 26.6 | 25.0 | 25.5 | 26.6 | 22:57 | 26.0 |
| 18:30 | 23.7 | 26.0 | 27.1 | 25.0 | 27.7 | 27.7 | 26.0 | 27.7 | 27.7 | 28.2 | 25.0 | 26.0 | 24.4 | 25.5 | 25.5 | 22:57 | 25.5 |
| 18:40 | 23.3 | 25.6 | 26.7 | 24.0 | 27.2 | 27.2 | 25.6 | 26.7 | 27.2 | 27.2 | 24.5 | 25.6 | 24.5 | 24.5 | 25.0 | 12:00 | 25.0 |
| 18:50 | 23.0 | 24.5 | 26.2 | 22.9 | 25.6 | 25.6 | 24.5 | 25.6 | 25.6 | 26.2 | 23.5 | 24.5 | 23.5 | 24.0 | 24.0 | 10:57 | 24.0 |
| 19:00 | 22.9 | 23.6 | 24.7 | 22.0 | 24.1 | 24.1 | 23.0 | 24.7 | 24.7 | 24.7 | 22.5 | 23.0 | 22.5 | 22.5 | 22.5 | 12:00 | 23.0 |
| 19:10 | 22.3 | 22.5 | 24.2 | 20.9 | 23.6 | 23.1 | 22.0 | 24.2 | 23.6 | 23.6 | 21.5 | 22.0 | 21.5 | 22.0 | 20.9 | 0:00 | 22.5 |
| 19:20 | 22.1 | 22.0 | 23.7 | 21.0 | 23.1 | 22.6 | 21.4 | 24.2 | 23.1 | 23.1 | 21.5 | 21.5 | 21.5 | 21.5 | 21.0 | 12:00 | 22.0 |
| 19:30 | 21.8 | 21.5 | 23.2 | 20.5 | 22.6 | 22.1 | 21.5 | 23.7 | 22.6 | 23.2 | 21.0 | 21.0 | 21.0 | 21.0 | 20.5 | 13:02 | 22.1 |
| 19:40 | 21.7 | 21.5 | 23.2 | 20.5 | 22.6 | 21.5 | 20.9 | 23.2 | 22.1 | 22.6 | 20.5 | 20.5 | 20.5 | 21.0 | 19.9 | 0:00 | 21.5 |
| 19:50 | 21.6 | 21.0 | 22.7 | 20.0 | 22.7 | 21.6 | 20.4 | 23.2 | 22.1 | 22.7 | 20.5 | 20.5 | 20.5 | 20.5 | 19.4 | 1:02 | 21.6 |
| 20:00 | 21.7 | 21.0 | 22.7 | 20.0 | 22.1 | 21.6 | 20.4 | 23.2 | 22.1 | 22.1 | 20.5 | 20.5 | 20.5 | 20.5 | 20.0 | 1:02 | 21.6 |
| 20:10 | 21.4 | 21.0 | 22.7 | 20.0 | 22.1 | 21.0 | 20.4 | 22.7 | 21.6 | 22.1 | 20.5 | 20.5 | 20.5 | 20.5 | 20.0 | 1:02 | 21.6 |
| 20:20 | 21.3 | 21.0 | 22.1 | 20.0 | 22.1 | 21.0 | 20.4 | 22.7 | 21.6 | 22.1 | 20.5 | 20.0 | 20.0 | 20.5 | 19.4 | 1:02 | 21.6 |
| 20:30 | 21.4 | 21.0 | 22.1 | 20.0 | 21.6 | 21.0 | 20.4 | 22.7 | 21.6 | 21.6 | 20.5 | 20.0 | 20.0 | 20.5 | 19.4 | 1:02 | 21.0 |
| 20:40 | 21.2 | 20.5 | 22.1 | 20.0 | 21.6 | 21.0 | 20.4 | 22.7 | 21.6 | 21.6 | 20.0 | 20.0 | 20.0 | 20.5 | 19.4 | 1:02 | 21.0 |
| 20:50 | 21.2 | 20.5 | 22.1 | 20.0 | 21.6 | 21.0 | 20.4 | 22.1 | 21.6 | 21.6 | 20.0 | 20.0 | 20.0 | 20.5 | 19.4 | 1:02 | 21.0 |
| 21:00 | 21.8 | 20.5 | 22.1 | 20.0 | 21.6 | 21.0 | 20.4 | 22.1 | 21.0 | 21.6 | 20.0 | 20.0 | 20.0 | 20.5 | 19.4 | 1:02 | 21.0 |

Key: A—Artificial grass; B—Artificial grass (laid); C—Asphalt; D—Black card on polystyrene; E—Concrete (x 40mm); F—Concrete (x 80mm); G—Crushed rock (white); H—Decking; I—Limestone block; J—Pavers (grey); K—Pavers (red); L—Pavers (sandstone); M—Pine bark mulch; N—Polished stones (black); O—Sand (grey); P—Sand (white); Q—Shade cloth (black); R—Shade cloth (cream); S—Shade cloth (white); T—Soil (dry); U—Soil (moist); V—Turf grass; W—White painted polystyrene; X—Ground cover (non-native, petunia); Y—Seedlings (native, saltbush).

## Appendix B

**Table A2.** Average daytime ratio $DR_{av}$, (the ratio of the black iButton temperature for each LE minus the ambient temperature, over the black iButton temperature of the WPP minus the ambient temperature). SE is the standard error. Data are ordered by cross-seasonal average.

| Landscape Elements | Autumn 2017 Average 09:30–15:00, SE 34 Data Points | | Winter 2017 Average 09:20–15:00, SE 35 Data Points | | Spring 2017 Average 09:00–15:00, SE 37 Data Points | | Summer 2018 Average 09:00–15:40, SE 41 Data Points | | Cross-Seasonal Average, SE 4 Data Points | |
|---|---|---|---|---|---|---|---|---|---|---|
| | $DR_{av}$ | SE | $DR_{av}$ | SE | $DR_{av}$ | SE | $DR_{av}$ | SE | $DR_{av}$ | SE |
| Soil (moist) | 0.33 | 0.00 | 0.20 | 0.01 | 0.24 | 0.04 | 0.29 | 0.01 | 0.26 | 0.03 |
| Pine bark mulch | 0.33 | 0.00 | 0.26 | 0.01 | 0.31 | 0.05 | 0.28 | 0.01 | 0.30 | 0.01 |
| Polished stones (black) | 0.27 | 0.00 | 0.36 | 0.01 | 0.27 | 0.05 | 0.33 | 0.01 | 0.31 | 0.02 |
| Asphalt | 0.23 | 0.01 | 0.36 | 0.01 | 0.35 | 0.07 | 0.34 | 0.01 | 0.35 | 0.03 |
| Artificial grass (laid) | | 0.00 | 0.36 | 0.01 | 0.37 | 0.08 | 0.36 | 0.01 | 0.37 | 0.00 |
| Soil (dry) | 0.40 | 0.01 | 0.34 | 0.01 | 0.43 | 0.07 | 0.32 | 0.01 | 0.37 | 0.03 |
| Pavers (grey) | 0.40 | 0.01 | 0.36 | 0.01 | 0.37 | 0.07 | 0.42 | 0.01 | 0.39 | 0.01 |
| Turf grass | 0.43 | 0.00 | 0.40 | 0.00 | 0.41 | 0.04 | 0.37 | 0.01 | 0.40 | 0.01 |
| Artificial grass | 0.43 | 0.00 | 0.39 | 0.01 | 0.41 | 0.05 | 0.42 | 0.01 | 0.42 | 0.01 |
| Sand (grey) | 0.50 | 0.01 | 0.47 | 0.01 | 0.43 | 0.06 | 0.37 | 0.01 | 0.44 | 0.03 |
| Shade cloth (black) | 0.50 | 0.00 | 0.46 | 0.01 | 0.44 | 0.04 | 0.38 | 0.01 | 0.45 | 0.02 |
| Black card on polystyrene | 0.47 | 0.01 | 0.42 | 0.01 | 0.45 | 0.05 | 0.48 | 0.01 | 0.46 | 0.01 |
| Pavers (red) | 0.54 | 0.01 | 0.41 | 0.01 | 0.50 | 0.06 | 0.56 | 0.01 | 0.50 | 0.03 |
| Concrete (x 40mm) | 0.57 | 0.01 | 0.51 | 0.01 | 0.53 | 0.08 | 0.54 | 0.01 | 0.54 | 0.01 |
| Concrete (x 80mm) | 0.56 | 0.01 | 0.54 | 0.01 | 0.56 | 0.08 | 0.54 | 0.01 | 0.55 | 0.01 |
| Limestone block | 0.84 | 0.00 | 0.56 | 0.01 | 0.55 | 0.07 | 0.58 | 0.00 | 0.56 | 0.07 |
| Decking | 0.64 | 0.01 | 0.59 | 0.01 | 0.59 | 0.06 | 0.55 | 0.01 | 0.59 | 0.02 |
| Shade cloth (cream) | 0.69 | 0.00 | 0.60 | 0.01 | 0.63 | 0.04 | 0.62 | 0.01 | 0.63 | 0.02 |
| Pavers (sandstone) | 0.66 | 0.01 | 0.64 | 0.01 | 0.63 | 0.07 | 0.68 | 0.01 | 0.65 | 0.01 |
| Sand (white) | 0.77 | 0.00 | 0.73 | 0.01 | 0.64 | 0.04 | 0.65 | 0.01 | 0.70 | 0.03 |
| Crushed rock (white) | 0.81 | 0.00 | 0.82 | 0.01 | 0.77 | 0.04 | 0.86 | 0.01 | 0.81 | 0.02 |
| Shade cloth (white) | 0.83 | 0.00 | 0.74 | 0.01 | 0.88 | 0.05 | 0.85 | 0.02 | 0.82 | 0.03 |
| White painted polystyrene | 1.00 | 0.00 | 1.00 | 0.00 | 1.00 | 0.00 | 1.00 | 0.00 | 1.00 | 0.00 |

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
