# Peer review of "Radiosity from Individual Urban Landscape Elements Measured Using a Modified Low-Cost Temperature Sensor"

_urbansci, doi:10.3390/urbansci4010014_

Round 1
Reviewer 1 Report
On December 2019 it was published in sustainability a really similar article form the same authors:
Sustainability 2019, 11(24), 6896; https://doi.org/10.3390/su11246896
Article Modified iButtons: A Low-Cost Instrument to Measure the Albedo of Landscape Elements Please clarify the difference between the two articles and the reserach they are describing.Author Response
Please see attachment.

Reviewer 2 Report
The manuscript is well written, of great interest and the method presented opens the doors to other researcher with limited budgets to study the behaviour of landscape elements to radiosity. The manuscript however needs to be improved.
The authors present a lot of details, mainly technical regarding the materials and measurements however miss to emphasise some key points regarding the methodology that need to be addressed. Also in the results and discussion area many of the arguments stated are assumptions and should be presented as such. A general question raised is how is your method validated? There seems to be no statistical analysis undertaken and no discussion of your work in relation to similar work of other researchers using conventional methods. Furthermore, the conclusions section reads more like a discussion, it should only summarise on the main research findings.
I have noted below in more detail some of the points raised above, in:
page 4 lines 137 and 138 before the brackets please add the author names.
page 5, lines 176-177 please replace author names with Loveday et al.
lines 172 -189 are too detailed and should be moved to the methodology.
page 6, line 201 you state the use of "crumpled" aluminium foil please state why crumpled.
line 202 you state 19+2 different LEs -Table 1 shows 23 total, please correct.
line 208 you mention "shade cloths", please state more details of the materials, there are so many types in the market, also please how were they maintained in place i.e. not blown away.
lines 202-217 concerning the "sub-base" i.e. turf, PP bag, etc. based on your comment 89-93 but also our research it may affect the surface temperatures of the LEs. Ideally all LEs to allow for comparisons between the materials should be "mounted" the same way to minimise variation. These research implications should be mentioned in the discussion section.
page 11 Table 2 there is no paint specification provided for the white painted polystyrene, please state.
Section 2.3 Data analysis
There is no mention of any statistical analysis undertaken. There seems that there were also no replicates of the LEs. How was significance (in the discussion determine). It needs to be clear if the results are derived after a statistical analysis or are a summary of the measurements taken and also state the research implications that are related to the latter.
page 13 lines 339-340 you state " Temperature results were not
affected by the use of PP bags and plastic crates to contain samples" how do you confirm this? please see comment above (lines 202-217).
lines 344-351 What LEs needed to be circular that couldn't be rectangular? Please state the necessity. Why not make all the same shape to reduce variation? Otherwise please state the research implications in the results and discussion section.
lines 351-352 and equation 1 are confusing: You state that
Le, grass, and PP were termed "other" yet the equation shows them independently, please correct.
lines 355-358 please rephrase or provide more detail how was the energy fraction removed?
page 15, figure 6 please provide reference of data as time period.
table 3 please provide dates related to the seasons in the legend and clarify that they are daily measurements.
page 17 Figure 7: you mention "comparison" In relation to the comment made above for section 2.3, Was there a test applied to assess differences? What was the P value? Please state in the legend the averaged time period, please also show SE bars above graphs.
page 18, figures 8 and 9 same comments as for figure 7
page 20 figure 10 please enlarge graph or change the representation or colours of the different seasons to be more clear. Also please replace the comments with outlier and state in the legend that outliers were not included.
lines 499-500 you measured wind speed, did you correlate the Drav values with the wind speed? Please provide a reference related to similar findings, link your work with other researchers, this is considered an assumption as it is not based on statistical findings
lines 503-513 the comments are similar to lines 499-500, with no statistical findings these are assumptions, you need to clarify these. Furthermore was the substrate moisture measured? The moisture may have an effect i.e. between LEs placed directly on the turf, or the PP bags. What was the moisture content of the materials i.e. soil, pine and how was this correlated to the DRav ? These "findings" without being determined statistically create more questions on the validity of the results therefore you need to support your results cautiously with examples of other researchers and make clear that they are assumptions.
page 22 lines 541-546 Your justification here contradicts your methodology that tended to this. In lines 355-359 you stated that the particular fractions were removed (TiB,LE) hence one would assume it was removed from the corresponding calculated DRav. Is this not the case?
page 23 line 568 please state the comfortable range for pedestrians and provide reference to emphasise the use of pedestrian friendly materials.
line 573, where would one specify the use of moist soil? What was the moisture content of the soil (it would affect your results), Furthermore, high temperature causes rapid evaporation on non vegetated soil surface. You need to state more clearly the research implications.
Reviewer 3 Report
The manuscript is an interesting experiment on the performances of a low cost sensor. I like the idea but from an experimental point of view it rose me up some questions:
The experiment was prevalently conducted in open field. Do you have conducted calibration experiment in controlled chambers? From my previous experiments conducted with low cost sensors one of the manin problem was related to deviations produced by changing in temperatures. Probably this it is not the case, thus this mine it is only a question/suggestion. The Figures 7,8, and 9 present strong variation on the tails of the graphs. Does it is the recults of the explanation reported in lines 445-462? I think that one more devoted line of explanation can help the reader.In my opinion the manuscript is of interest and can be published once some few improvements in the description of the methodology have been made.
Round 2
Reviewer 2 Report
I am happy to see the authors have improved their manuscript. There a few oversights that need to be addressed and some minor additional information needed. Please find my comments below:
Lines 194-200 are repeated in lines 226-233 and lines 242-247. I think this is perhaps an oversight. It seems to me that lines 194-200 and 226-230 do not make sense. Please check
Also Line 201 and line 234 Table) ..... something is missing here. I do not understand.
Line 201 ....were positioned in the centre.... Something in the sentence is missing. Does not make sense. Please correct
Line 283 Please state plant and pot size (planting density i.e. the distance between pots and number of pots per m2 or surface of LE)
Line 289 Please replace "10 " at the beginning of the sentence with "Ten"
line 303 Table 2 legend: please describe in more detail. Legends need to stand on their own without referring to the main text ie. Installation details of Landscape Elements (LE) used in Study Design Phase 2, during the summer 2019. In table please add asterisk in Sub-layer* or the first row sand* to link with the Table footnote (* missing in table)
line 455 missing "Table? "
line 463 Table 3 ??? I presume it was meant for line 455? Please check
line 485 please provide a more detailed legend for the table.
Line 573 ...shows Something is missing, please correct.
Line 581 missing "Table? " Please add n value (number of measurements of mean value calculated)
Line 695 Table 7 ?? Is there a table?
